# DOMAIN-ADJUSTED REGRESSION OR: ERM MAY ALREADY LEARN FEATURES SUFFICIENT FOR OUT-OF-DISTRIBUTION GENERALIZATION

## ABSTRACT

A common explanation for the failure of deep networks to generalize out-of-distribution is that they fail to recover the "correct" features. We challenge this notion with a simple experiment which suggests that ERM already learns sufficient features and that the current bottleneck is not feature learning, but *robust regression*. Our findings also imply that given a small amount of data from the target distribution, retraining only the last linear layer will give excellent performance. We therefore argue that devising simpler methods for learning predictors on existing features is a promising direction for future research. Towards this end, we introduce *Domain-Adjusted Regression* (DARE), a convex objective for learning a linear predictor that is provably robust under a new model of distribution shift. Rather than learning one function, DARE performs a domain-specific adjustment to unify the domains in a canonical latent space and learns to predict in this space. Under a natural model, we prove that the DARE solution is the minimax-optimal predictor for a constrained set of test distributions. Further, we provide the first finite-environment convergence guarantee to the minimax risk, improving over existing analyses which only yield minimax predictors after an environment threshold. Evaluated on finetuned features, we find that DARE compares favorably to prior methods, consistently achieving equal or better performance.

## 1 INTRODUCTION

The historical motivation for deep learning focuses on the ability of deep neural networks to automatically learn rich, hierarchical features of complex data (LeCun et al., 2015; Goodfellow et al., 2016). Simple Empirical Risk Minimization (ERM), with appropriate regularization, results in high-quality representations which surpass carefully hand-selected features on a wide variety of downstream tasks. Despite these successes, or perhaps because of them, the dominant focus of late is on the shortcomings of this approach: recent work points to the failure of networks trained with ERM to generalize under even moderate distribution shift (Recht et al., 2019; Miller et al., 2020). A common explanation for this phenomenon is reliance on "spurious correlations" or "shortcuts", where a network makes predictions based on structure in the data which generalizes on average in the training set but may not persist in future test distributions (Poliak et al., 2018; Geirhos et al., 2019; Xiao et al., 2021).

Many proposed solutions *implicitly assume* that this problem is due to the entire neural network: they suggest an alternate objective to be minimized over a deep network in an end-to-end fashion (Sun & Saenko, 2016; Ganin et al., 2016; Arjovsky et al., 2019). These objectives are complex, poorly understood, and difficult to optimize. Indeed, the efficacy of many such objectives was recently called into serious question (Zhao et al., 2019; Rosenfeld et al., 2021; Gulrajani & Lopez-Paz, 2021). Though a neural network is often viewed as a deep feature embedder with a final linear predictor applied to the features, it is still unclear—and to our knowledge *has not been directly asked or tested*—whether these issues are primarily because of (i) learning the wrong features or (ii) learning good features but failing to find the best-generalizing linear predictor on top of them.

We begin with a simple experiment (Figure 1) to try to distinguish between these two possibilities: we train a deep network with ERM on several domain generalization benchmarks, where on task is to learn a predictor using a collection of distinct training domains and then perform well on a new,

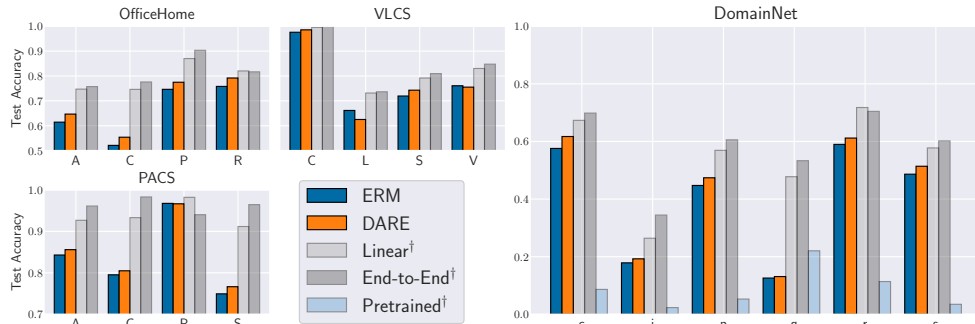

Figure 1: Accuracy via "cheating": dagger (†) denotes access to test domain at train-time. Each letter is a domain. Dark blue is approximate SOTA, orange is our proposed DARE objective, light grey represents cheating while retraining the linear classifier only. All three methods use the *same features*, attained without cheating. Dark grey is "ideal" accuracy, cheating while training the entire deep network. Surprisingly, cheating only for the linear classifier rivals cheating for the whole network. Cheating accuracy on pretrained features (light blue) makes clear that this effect is due to finetuning on the train domains, and not simply overparameterization (i.e., a very large number of features).

unseen domain. After training, we freeze the features and separately learn a linear classifier on top of them. Crucially, when training this classifier (i.e., retraining just the last linear layer), we give it an unreasonable advantage by optimizing on both the train and test domains—henceforth we refer to this as "cheating". Since we use just a linear classifier, this process establishes a lower bound on what performance we could plausibly achieve using standard ERM features. We then separately cheat while training the full network end-to-end, simulating the idealized setting with no distribution shift. Note that in neither case do we train on the test *points*; our cheating entails training on (different) samples from the test *domain*, which are assumed unavailable in domain generalization.

Notably, we find that simple (cheating) logistic regression on frozen deep features learned via ERM results in enormous improvements over current state of the art, on the order of 10-15%. In fact, it usually performs comparably to the full cheating method—which learns both features and classifier end-to-end with test domain access—sometimes even outperforming it. Put another way, **cheating while training the entire network rarely does significantly better than cheating while training just the last linear layer.** One possible explanation for this is that the pretrained model is so overparametrized as to effectively be a kernel with universal approximation power; in this case, the outstanding performance of a cheating linear classifier on top of these features would be unsurprising. However, we find that this cheating method does *not* ensure good performance on pretrained features, which implies that we are not yet in such a regime and that the effect we observe is indeed due to finetuning via ERM. Collectively, these results suggest that **training modern deep architectures with ERM and established in-distribution training and regularization practices may be "good enough" for out-of-distribution generalization** and that the current bottleneck lies primarily in learning a simple, robust predictor.

Motivated by these findings, we propose a new objective, which we call Domain-Adjusted Regression (DARE). The DARE objective is convex and it learns a linear predictor on frozen features. Unlike invariant prediction (Peters et al., 2016), which projects out feature variation such that a single predictor performs acceptably on very different domains, DARE performs a domain-specific adjustment to unify the environmental features in a canonical latent space. Based on the presumption that standard ERM features are good enough (made formal in Section 4), DARE enjoys strong theoretical guarantees: under a new model of distribution shift which captures ideas from invariant/non-invariant latent variable models, we precisely characterize the adversarial risk of the DARE solution against a natural perturbation set, and we prove that this risk is *minimax*. We further provide the first *finite-environment* convergence guarantee to the minimax risk, improving over existing results which merely demonstrate a threshold in the number of observed environments at which the solution is discovered (Rosenfeld et al., 2021; Chen et al., 2021; Wang et al., 2022). Finally, we show how our objective can be modified to leverage access to unlabeled samples at test-time. We use this to derive a method for provably effective "just-in-time" unsupervised domain adaptation, for which we provide a finite-sample excess risk bound.

Evaluated on finetuned features, we find that DARE compares favorably to existing methods, consistently achieving equal or better performance. We also find that methods which previously underperformed on these benchmarks do much better in this frozen feature setting, often besting ERM. This suggests that these approaches *are* beneficial for linear prediction (the setting in which they are understood and justified) but using them to train deep networks may result in worse features.

## 2 ERM LEARNS SURPRISINGLY USEFUL FEATURES

Our experiments are motivated by a simple question: **is the observed failure of deep neural networks to generalize out-of-distribution more appropriately attributed to inadequate** *feature learning* **or inadequate** *robust prediction***?** Both could undoubtedly be improved, but we are concerned with which currently serves as the primary bottleneck. It's typical to train the entire network end-to-end and then evaluate on a test distribution; in reality, this measures the quality of the interaction of the features and the classifier, not of either one individually.

Using datasets and methodology from DOMAINBED (Gulrajani & Lopez-Paz, 2021), we finetune a ResNet-50 (He et al., 2016) with ERM on the training domains to extract features. Next, we cheat while learning a linear classifier on top of these frozen features by optimizing on both the train and test domains. We compare this cheating linear classifier to a full cheating network trained on all domains end-to-end. If it is the case that ERM learns features which do not generalize, we should expect that the cheating linear classifier will not substantially improve over the current state of the art and will perform significantly worse than cheating end-to-end, since the latter method can adapt the features to better suit the test domain.

Instead, we find that simply giving the linear predictor access to all domains while training makes up the vast majority of the gap between current state of the art (ERM with heavy regularization) and the ideal setting where we train a network on all domains. In other words, ERM produces features which are informative enough that a linear classifier on top of these frozen features is—in principle—capable of generalizing *almost as well as if we had access to the test domain when training the entire network.*[1] Figure 2 in the Appendix depicts the evaluation methodology described above, along with a more detailed explanation. We conjecture that this phenomenon occurs more broadly: our findings suggest that for many applications, existing features learned via ERM may be "good enough" for out-of-distribution generalization in deep learning. Based on this idea, we posit that future work would benefit from modularity, working to improve representation learning and robust classification/regression separately. There are several distinct advantages to this approach:

- **More robust comparisons and better reproducibility:** Current methods have myriad degrees of freedom which makes informative comparisons difficult; evaluating feature learning and robust regression separately eliminates many of these sources of experimental variation.
- **Less compute overhead:** Training large networks to learn both features and a classifier is expensive. Benchmarks could include files with the weights for ready-to-use deep feature embedders trained with various objectives—these models can be much larger and trained on much more data than would be feasible for many. Using these features, academic researchers could thus make faster progress on better classifiers with less compute.
- **Modular theoretical guarantees:** Conditioning on the frozen features, we can use more classical analyses to provide guarantees for the simpler parametric classifiers learned on top of these features. For example, Bansal et al. (2021) derive a generalization bound for simpler classifiers which is agnostic to the complexity of the features on which they are trained.

We conclude by emphasizing that while the predictor in our experiments is linear, future methods need not be. Rather, we are highlighting that **there may be no need for complex, expensive, highly variable regularization of a deep network when much simpler approaches suffice.**

## 3 THE DOMAIN-ADJUSTED REGRESSION OBJECTIVE

The goal of many prior methods in deep domain adaptation or generalization is to learn a single network which does well on all environments simultaneously, often by throwing away non-invariant

---

[1]On a few domains the linear method sees a gap of $\sim$5% accuracy from the idealized setting. We emphasize that our use of a simple linear predictor serves as a *lower bound* on the achievable error using ERM features.

components (Peng et al., 2019; Arjovsky et al., 2019). While invariance is a powerful framework, there are some clear drawbacks , such as the need to throw away possibly informative features. In settings where we expect the distribution shift to be less than worst-case, this would be unnecessarily conservative. Instead, we reframe the problem by thinking of each training domain as a distinct transformation from a shared canonical representation space. In this framing, we can "adjust" each domain in order to undo these transformations, aligning the representations to learn a single robust predictor in this unified space. Specifically, we propose to whiten each domain's features to have zero mean and identity covariance, which can be thought of as a "domain-specific batchnorm" but using the full feature covariance. This idea is not totally new: some prior works align the moments between domains to improve generalization. The difference is that DARE does not learn a single featurizer which aligns the moments, but rather *aligns the moments of already learned features*; the latter approach maintains useful variation between domains which would be eliminated by the former. DARE is closer to methods which learn separate batchnorm parameters per domain over a deep network, possibly adjusting at test-time (Seo et al., 2019; Chang et al., 2019; Segù et al., 2020)—these methods perform well, but they are entirely heuristic based, difficult to optimize, and come with no formal guarantees. Our theoretical analysis thus serves as preliminary justification for the observed benefits of such methods, which have so far lacked serious grounding.

To begin, define $\mathcal{E}$ as the set of observed training environments, each of which is defined by a distribution $p^e(x, y)$ over features $x \in \mathbb{R}^d$ and class labels $y \in [k]$. For each $e \in \mathcal{E}$, denote the mean and covariance of the features as $\mu_e, \Sigma_e$. Our first step is to adjust the features via the whitening transformation $\Sigma_e^{-1/2}(x - \mu_e)$. Unfortunately, we cannot undo the test transformation because we have no way of knowing what the test mean will be. Interestingly, this is not a problem provided the predictor satisfies a simple constraint. Suppose that the predictor's output on the mean representation of each environment is a multiple of the all-ones vector. As softmax is invariant to a constant offset, this enforces that the environment mean has *no effect* on the final probability distribution, and thus there is no need to adjust for the test-time mean. We therefore enforce this constraint during training, with the hope that the same invariance will approximately hold at test-time. Formally, the DARE objective finds a matrix $\beta \in \mathbb{R}^{d \times k}$ which solves

$$\min_\beta \sum_{e \in \mathcal{E}} \mathbb{E}_{p^e}[\ell(\beta^T \Sigma_e^{-1/2} x, \ y)] \quad \text{subject to softmax} \left( \beta^T \Sigma_e^{-1/2} \mu_e \right) = \frac{1}{k} \mathbf{1}. \ \ \forall e \in \mathcal{E}, \quad (1)$$

where $\ell$ is the multinomial logistic loss and we omit the bias $\beta_0$ for brevity. For binary classification, $\beta$ is a vector and the softmax is replaced with the logistic function—the constraint is then equivalent to requiring the mean of the logits to be 0. Thus, the DARE objective explicitly regresses on the adjusted features, while the constraint enforces that each environment mean has no effect on the output distribution to encourage predictions to also be invariant to test-time transformations. The astute reader will point out that we also do not know the correct whitening matrix for the test data—instead, we adjust using our best guess for the test covariance: denoting this estimate as $\bar{\Sigma}$, our prediction on a new sample $x$ is $f(x; \beta) = \text{softmax} \left( \beta^T \bar{\Sigma}^{-1/2} x \right)$. We prove that this prediction is minimax so long as our guess is "sufficiently close", and in practice we find that simply averaging the training domain adjustments performs well. Table 2 in the Appendix shows this average is actually quite close to the sample covariance of the (unseen) test domain, explaining the good performance.

Unlike many prior methods, DARE does not enforce invariance of the features themselves. Rather, it *aligns the representations* such that different domains share similar optimal predictors. To support this claim, Table 3 displays the cosine similarity between optimal linear classifiers for individual domains—we observe a large increase in average similarity as a result of the feature adjustment. Further, in Section 5 we demonstrate that the DARE objective is in fact minimizing the worst-case risk under a constrained set of possible distribution shifts, making it less conservative than methods which require complete feature invariance. Though the objective (1) assumes a particular form of invariance, **this specific choice is by no means a requirement for the approach we take.** We justify this by noting that the minimax predictor can always be written as the solution to an objective of this form, depending only on what we choose to assume holds constant:

**Proposition 3.1** (informal). *Assume the minimax-optimal predictor $f^*$ lies in our hypothesis class $\mathcal{F}$. Let $S : \mathcal{F} \times \mathcal{P}(\mathcal{X} \times \mathcal{Y}) \to \mathbb{R}^d$ be any "sufficient statistic" which encodes all invariances, i.e. $S(f^*, p_e) = c \ \forall e$, for some constant $c$, and $\forall f \in \mathcal{F}$ with lower risk than $f^*$ on at least one environment, $\exists e, e'. \ S(f, p_e) \neq S(f, p_{e'})$. Then as $|\mathcal{E}| \to \infty$, $f^* = \arg\min_{f \in \mathcal{F}} \frac{1}{|\mathcal{E}|} \sum_{e \in \mathcal{E}} \ell(f(x), y) \ s.t. \ S(f, p_e) = c \ \forall e \in \mathcal{E}.$*

This generic objective recovers IRM by defining $S(f, p_e) = \mathbb{E}_{p_e}[\nabla \ell(f(x), y))]$, $c = \mathbf{0}$ (the "sufficient statistic" is the environment loss gradient), or CORAL with $S(f, p_e) = \left[ \mathbb{E}_{p_e}[x], \mathbb{E}_{p_e}[(x - \mu)(x - \mu)^T]] \right]$ (the feature mean and covariance) and letting $c$ be arbitrary. DARE instead allows for domain-specific transformations $T_e$, and we define $S(f, p_e) = \text{softmax} \left( \mathbb{E}_{p_e}[f(T_e^{-1}(x))] \right)$—here $T_e^{-1}$ is the whitening operation detailed in the next section.

**Implementation in practice.** Due to its convexity, the DARE objective is extremely simple to optimize. In practice, we finetune a deep network over the training data with ERM and then extract the features. Next, treating the frozen features of the training data as direct observations $x$, we minimize the empirical Lagrangian form:

$$\hat{\mathcal{L}}_{\text{cls}}^\lambda(\beta) := \frac{1}{|\mathcal{E}|} \sum_{e \in \mathcal{E}} \left[ \frac{1}{n_e} \sum_{i=1}^{n_e} \ell(\beta^T \hat{\Sigma}_e^{-1/2} x_i, \ y_i) + \lambda \ell(\beta^T \hat{\Sigma}_e^{-1/2} \hat{\mu}_e, k^{-1} \mathbf{1}) \right],$$

where $\hat{\mu}_e, \hat{\Sigma}_e$ are the usual sample estimates and $n_e$ is the number of samples in environment $e$. We find that the solution is incredibly robust to the choice of $\lambda$, but it is natural to wonder whether each of the components above is necessary for the performance gains we observe. We ablate both the whitening operation and the use of the constraint (Appendix E) and see performance drops in both cases. We also consider estimating the test-time mean rather than enforcing invariance, but this results in substantially worse accuracy—the environment feature means vary quite a bit, so the estimate is usually inaccurate. For regression, we consider the same setup but minimize mean squared error on targets $y \in \mathbb{R}$. Here, the DARE solution is constrained such that the mean output has no effect on the prediction, meaning each domain's mean prediction should be zero. We discuss this in more detail in Appendix A.1, along with an interesting connection to anchor regression (Rothenhäusler et al., 2021).

Another benefit to our approach is that the adjustments for each domain do not depend on labels, so given unlabeled samples from the test domain we can often do even better. Unlike methods which use those samples while training (or even for test-time training), this adjustment can be done *just-in-time*, without updating any parameters! We name this task *Just-in-Time Unsupervised Domain Adaptation* (JIT-UDA), and we provide finite-sample risk bounds for the DARE objective in this setting.

## 4  A NEW MODEL OF DISTRIBUTION SHIFT

The DARE objective is based on the intuition that all domains jointly share a representation space and that they arise as unique transformations from this space. To capture this notion mathematically, we model the joint distribution $p^e(\epsilon, y)$ over latents $\epsilon \in \mathbb{R}^d$ and label $y \in \{0, 1\}$, along with an environment-specific transformation to observations $x \in \mathbb{R}^d$ for each domain. In the fully general case, this transformation can take an arbitrary form and can be written as $x = T_e(\epsilon, y)$.

Our primary assumption is that $p^e(y \mid \epsilon)$ is constant for all domains. Our goal is thus to invert each transformation $T_e$ such that we are learning an invariant conditional $p(y \mid T_e^{-1}(x))$ (throughout we assume $T_e$ is invertible). One can also view this model as a generalization of covariate shift: where the usual assumption is constancy of $p(y \mid x)$, the inverse transformation gives a richer model which can more realistically capture real-world variation across domains. It is important to note that this model generalizes (and has *strictly weaker* requirements than) both IRM and domain-invariant representation learning, which can be recovered by assuming $T_e$ is the same for all environments.

For a typical deep learning analysis we would model $T_e$ as a non-linear generative map from latents to high-dimensional observations. However, our finding that ERM features are good enough suggests that modeling the learned features as a simple function of the "true latents" $\epsilon$ is not unreasonable. In other words, we now consider $x$ to represent the frozen features of the trained network, and we expect this network to have already "undone" the majority of the complexity, resulting in observations which are a simple function of the ground truth. Accordingly, we consider the following model:

$$\epsilon = \epsilon_0 + b_e, \ y = \mathbf{1}\{\beta^{*T} \epsilon + \eta \geq 0\}, \ x = A_e \epsilon. \tag{2}$$

Here, $\epsilon_0 \sim p^e(\epsilon_0)$, which we allow to be *any domain-specific distribution*; we assume only that its mean is zero (such that $\mathbb{E}[\epsilon] = b_e$) and that its covariance exists. We fix $\beta^* \in \mathbb{R}^d$ for all domains and model $\eta$ as logistic noise. Finally, $A_e \in \mathbb{R}^{d \times d}, b_e \in \mathbb{R}^d$ are domain-specific. For regression, we model the same generative process for $x, \epsilon$, while for the response, we have $y \in \mathbb{R}$ and $\eta$ is zero-mean independent noise: $y = \beta^{*T} \epsilon + \eta$. We remark that the reason for separate definitions of $p^e(\epsilon_0)$ and

$b_e$ is that our robustness guarantees are agnostic to the distribution of latent residuals $p^e(\epsilon_0)$—the *only* aspect of the latent distribution $p(\epsilon)$ which affects our bounds is the environment mean $b_e$.

**Connection to Invariant Prediction.** Statistical models of varying and invariant (latent) features have recently become a popular tool for analyzing the behavior of deep representation learning algorithms (Rosenfeld et al., 2021; Chen et al., 2021; Wald et al., 2021). We see that Equation (2) can model such a setting by assuming, e.g., that a subspace of the columnspan of $A_e$ is constant for all $e$, while the remaining subspace can vary. In such a case, the features $x$ are expressed as the sum of a varying and an invariant component, and any minimax representation must remove the varying component, throwing away potentially useful information. Instead, DARE *realigns* these components so that we can use them at test-time. We illustrate this with the following running example:

**Example 1** (Invariant Latent Subspace Recovery). Consider the model (2) with $\epsilon \sim \mathcal{N}(0, I_{d_1+d_2})$. Define $\Pi := \text{blockdiag}(I_{d_1}, \mathbf{0}_{d_2})$ and assume $A_e = \text{blockdiag}(\Sigma^{1/2}, \Sigma_e^{1/2})$ for all $e$, where $\Sigma \in \mathbb{R}^{d_1 \times d_1}$ is constant for all domains but $\Sigma_e \in \mathbb{R}^{d_2 \times d_2}$ varies.

Here we have a simple latent variable model: the features have the decomposition $x = A_e\epsilon = \Sigma^{1/2}\Pi\epsilon + \Sigma_e^{1/2}(I - \Pi)\epsilon$, where the first component is constant across environments and the second varies. It will be instructive at this point to analyze what an invariant prediction algorithm such as IRM would do in this setting. Here, the IRM constraint would enforce learning a featurizer $\Phi$ such that $\Phi(x)$ has an invariant relationship with the target. Under the above model, the solution is $\Phi(x) = \Pi A_e^{-1}x = \Sigma^{1/2}\Pi\epsilon$, retainining only the component lying in span($\Pi$). Crucially, removing these features actually results in *worse* performance on the training environments—IRM only does so because using these features could harm test performance in the worst case. However, projecting out the varying component in this manner is unnecessarily conservative, as we may expect that for a future distribution, $\Sigma_{e'}$ will not be *too* different from what we have seen before. Instead of projecting out this component, DARE performs a more nuanced alignment of environment subspaces, and it is thus able to take advantage of this additional informaton. We will see shortly the resulting benefits.

## 5 THEORETICAL ANALYSIS

Before we can analyze the DARE objective, we observe that there is a possible degeneracy in Equation (2), since two identical observational distributions $p(x, y)$ can have different regression vectors $\beta^*$. We therefore begin with a simple non-degeneracy assumption:

**Assumption 5.1.** Write the SVD of $A_e$ as $U_e S_e V_e^T$. We assume $V_e = V \; \forall e$.

One special case where this holds is when $A_e$ is constant for all domains; this is very similar to the "additive intervention" setting of Rothenhäusler et al. (2021), since only $p^e, b_e$ can vary. We assume WLOG that $V = I$, as any other value can be subsumed by the other parameters. We further let $\mathbb{E}[\epsilon_0\epsilon_0^T] = I$ WLOG by the same reasoning (see Appendix A.2 for a discussion on these conditions). With Assumption 5.1, we can uniquely recover $A_e = U_e S_e$ via the eigendecomposition of the covariance $\Sigma_e = A_e A_e^T = U_e S_e^2 U_e^T$. We therefore use the notation $\Sigma_e^{1/2}$ to refer to $A_e$ recovered in this way. We allow covariances to have zero eigenvalues, in which case we write the matrix inverse to implicitly refer to the pseudoinverse. As is standard in domain generalization analysis, unless stated otherwise we assume full distribution access to the training domains, though standard concentration inequalities could easily be applied.

**A remark on our assumptions.** Assumption 5.1 is not trivial. Domain generalization is an exceptionally difficult problem, and showing anything meaningful requires *some* assumption of consistency between train and test. Our assumptions are only as strong as necessary to prove our results, but future work could relax them, resulting in weaker performance guarantees. Experiments in Appendix E demonstrate that our covariance estimation is indeed accurate, and the fact that our method exceeds state of the art even with this strong constraint (and does worse without the constraint, see Figure 4) is further evidence that our assumptions are reasonable.

We begin by deriving the solution to Equations (1) and (3) . Recall that the DARE constraint requires that the mean representation of each domain has no effect on our prediction. To enforce this, the DARE solution must project out the subspace in which the means vary. Given a set of $E$ training environments, define $\mathbb{B}$ as the $d \times E$ matrix whose columns are the environmental mean parameters $b_e$.

Throughout this section, we make use of the matrix $\hat{\Pi}$, which is defined as the orthogonal projection onto the nullspace of $\mathbb{B}^T$: $\hat{\Pi} := I - \mathbb{B}\mathbb{B}^\dagger = U_{\hat{\Pi}} S_{\hat{\Pi}} U_{\hat{\Pi}}^T \in \mathbb{R}^{d \times d}$. This matrix projects onto the DARE constraint set, and it turns out to be all that is necessary to state the solution:

**Theorem 5.2.** (Closed-form solution to the DARE population objective). *Under model (2), the solution to the* DARE *population objective (3) for linear regression is* $\hat{\Pi}\beta^*$. *If $\epsilon$ is Gaussian, then the solution for logistic regression (1) is* $\alpha\hat{\Pi}\beta^*$ *for some* $\alpha \in (0, 1]$.

In Example 1, we saw how invariant prediction will discard a large subspace of the representation and why this is undesirable. Instead, DARE undoes the environment transformation and regresses directly on $\epsilon = T_e^{-1}(x)$. Because we are performing a separate transformation for each environment, we are aligning the varying subspaces rather than throwing them away, allowing us to make use of additional information. Though the DARE solution also recovers $\beta^*$ up to a projection, it is using the adjusted features; DARE therefore only removes what cannot be aligned. In particular, whereas $\Pi$ has rank $d_1$ in Example 1, $\hat{\Pi} = I$ would have full rank—this retains strictly more information. Indeed, in the ideal setting where we have a good estimate of $\Sigma_{e'}$ (e.g., under mild distribution shift or when solving JIT-UDA), we can make the Bayes-optimal prediction as $\mathbb{E}[y \mid x] = \beta^{*T} A_{e'}^{-1} x$. Thus we see a clear advantage that DARE enjoys over invariant prediction.

### 5.1 THE ADVERSARIAL RISK OF DARE

Moving forward, we denote the DARE solution $\beta_{\hat{\Pi}}^* := \hat{\Pi}\beta^*$, with $\beta_{I-\hat{\Pi}}^*$ defined analogously. We next study the behavior of the DARE solution under worst-case distribution shift. We consider the setting where an adversary directly observes our choices of $\bar{\Sigma}, \hat{\beta}$ and chooses new environmental parameters $A_{e'}, b_{e'}$ so as to cause the greatest possible loss. Specifically, we study the square loss, defining the *excess test risk* of a predictor as $\mathcal{R}_{e'}(\hat{\beta}) := \mathbb{E}_{p_{e'}}[(\hat{\beta}^T \bar{\Sigma}^{-1/2} x - \beta^{*T} \epsilon)^2]$ (we leave the dependence on $\bar{\Sigma}$ implicit). For logistic regression we therefore analyze the squared error with respect to the log-odds. With some abuse of notation, we also reference excess risk when only using a particular subspace, i.e. $\mathcal{R}_{e'}^\Pi(\hat{\beta}) := \mathbb{E}_{p_{e'}}[(\hat{\beta}_{\hat{\Pi}}^T \bar{\Sigma}^{-1/2} x - \beta_{\hat{\Pi}}^{*T} \epsilon)^2]$.

Because we guess $\bar{\Sigma}$ before observing any data from this new distribution, ensuring success is impossible in the general case. Instead, we consider a set of restrictions on the adversary which will make the problem tractable. Define the error in our test domain adjustment as $\Delta := \Sigma_{e'}^{1/2} \bar{\Sigma}^{-1/2} - I$; observe that if $\bar{\Sigma} = \Sigma_{e'}$, then $\Delta = \mathbf{0}$. Our first assumption[2] says that the effect of our adjustment error with respect to the *interaction between subspaces* $\hat{\Pi}$ and $(I - \hat{\Pi})$ is bounded:

**Assumption 5.3.** For a fixed constant $B \geq 0$, $\|(I - \hat{\Pi})\Delta\hat{\Pi}\hat{\beta}\| \leq B\|\hat{\Pi}\beta^*\|$.

*Remark* 5.4. To see specific cases when this would hold, consider the decomposition of $\Delta$ according to its components in the subspaces $\hat{\Pi}$ and $I - \hat{\Pi}$: $U_{\hat{\Pi}}^T \Delta U_{\hat{\Pi}} = \begin{bmatrix} \Delta_1 & \Delta_{12} \\ \Delta_{21} & \Delta_2 \end{bmatrix}$. A few settings automatically satisfy Assumption 5.3 with $B = 0$, since $U_{\hat{\Pi}}^T \Delta U_{\hat{\Pi}}$ will be block-diagonal. In particular, this will be the case if all domains share an invariant subspace—e.g., if $\Pi A_e \Pi$ is constant as in Example 1. Below, we show that exact recovery of this subspace occurs once we observe $\text{rank}(I - \Pi)$ environments—this matches (actually, it is one less than) the linear environment complexity of most invariant predictors (Rosenfeld et al., 2021), and therefore Assumption 5.3 with $B = 0$ is no stronger than assuming a linear number of environments.

**Assumption 5.5.** Using only covariates in the non-varying subspace, the risk of the ground truth regressor $\beta^*$ is less than that of the trivial zero predictor: $\mathcal{R}_{p_{e'}}^{\hat{\Pi}}(\beta^*) < \mathcal{R}_{p_{e'}}^{\hat{\Pi}}(\mathbf{0})$.

The need for this restriction should be immediate—if our adjustment error were so large that this did not hold, even the oracle regression vector would do worse than simply always predicting $\hat{y} = 0$. Assumption 5.5 is satisfied for example if $\|\Delta\hat{\Pi}\| < 1$, which again is guaranteed if there is an invariant subspace. Note that we make *no restriction* on the risk in the subspace $I - \hat{\Pi}$—the adversary is allowed any amount of variation in directions where we have *already seen* variation in the mean terms $b_e$, but introduction of *new* variation is assumed bounded. **This is a no-free-lunch necessity:** if we have never seen a particular type of variation, we cannot possibly know how to use it at test-time.

---

[2]Though we label these as assumptions, they are properly interpreted as *restrictions* on an adversary—we consider an "uncertainty set" comprising all possible domains subject to these requirements.

With these restrictions on the adversary, our main result derives the supremum of the excess test risk of the DARE solution under adversarial distribution shift. Furthermore, we prove that this risk is *minimax*: making no more restrictions on the adversary other than a global bound on the mean, the DARE solution achieves the best performance we could possibly hope for at test-time:

**Theorem 5.6** (DARE risk and minimaxity). *Denote the set of possible test environments $\mathcal{A}_\rho$ which contains all parameters $(A_{e'}, b_{e'})$ subject to Assumptions 5.3 and 5.5 and a bound on the mean: $\|b_{e'}\| \leq \rho$. For logistic or linear regression, let $\hat{\beta}$ be the minimizer of the corresponding DARE objective as in Theorem 5.2. Then, $\sup_{(A_{e'}, b_{e'}) \in \mathcal{A}_\rho} \mathcal{R}_{e'}(\hat{\beta}) = (1 + \rho^2)(\|\beta^*\|^2 + 2B\|\beta^*_{\hat{\Pi}}\|\|\beta^*_{I-\hat{\Pi}}\|)$. Furthermore, the DARE solution is* minimax: $\hat{\beta} \in \arg\min_{\beta \in \mathbb{R}^d} \sup_{(A_{e'}, b_{e'}) \in \mathcal{A}_\rho} \mathcal{R}_{e'}(\beta)$.

A special case when our assumptions hold is when all domains share an invariant subspace and we only predict using that subspace, but this is often too conservative. There are settings where allowing for (limited) new variation can improve our predictions, and Theorem 5.6 shows that DARE should outperform invariant prediction in such settings.

## 5.2 THE ENVIRONMENT COMPLEXITY OF DARE

An important new measure of domain generalization algorithms is their *environment complexity*, which describes how the test risk behaves as a function of the number of (possibly random) domains we observe. In contrast to Example 1, for this analysis we assume an invariant subspace $\Pi$ outside of which *both* $A_e$ and $b_e$ can vary arbitrarily—we formalize this as a prior over the $b_e$ whose covariance has the same span as $I - \Pi$. Our next result demonstrates that DARE achieves the same threshold as prior methods, but we also prove the first *finite-environment convergence guarantee*, quantifying how quickly the risk of the DARE predictor approaches that of the minimax-optimal predictor.

**Theorem 5.7** (Environment complexity of DARE). *Define the smallest gap between consecutive eigenvalues: $\xi(\Sigma) := \min_{i \in [d-1]} \lambda_i - \lambda_{i+1}$. Fix test parameters $A_{e'}, b_{e'}$ and guess $\bar{\Sigma}$. Suppose we minimize the* DARE *regression objective (3) on environments whose means $b_e$ are Gaussian vectors with covariance $\Sigma_b$, with $span(\Sigma_b) = span(I - \Pi)$. After seeing $E$ training domains:*

*1. If $E \geq \text{rank}(\Sigma_b)$ then* DARE *recovers the minimax-optimal predictor almost surely: $\hat{\beta} = \beta^*_\Pi$.*

*2. Otherwise, if $E \geq r(\Sigma_b)$ then with probability $\geq 1 - \delta$,*

$$\mathcal{R}_{e'}(\hat{\beta}) \leq \mathcal{R}_{e'}(\beta^*_\Pi) + \mathcal{O}\left(\frac{\|\Sigma_b\|}{\xi(\Sigma_b)}\left(\sqrt{\frac{r(\Sigma_b)}{E}} + \max\left\{\sqrt{\frac{\log 1/\delta}{E}}, \frac{\log 1/\delta}{E}\right\}\right)\right),$$

*where $\mathcal{O}(\cdot)$ hides dependence on $\|\Delta\|$ and $r(\Sigma) := \text{Tr}(\Sigma)/\|\Sigma\|$ is the effective rank.*

For coherence we present the first item as a probabilistic statement, but it holds deterministically so long as there are $\text{rank}(\Sigma_b)$ linearly independent observations of $b_e$.

*Remark* 5.8. Prior analyses of invariant prediction methods only show a discontinuous threshold where the minimax predictor is discovered after seeing a fixed number of environments—usually linear in the non-invariant latent dimension But one should expect that if the variation is not *too* large then we can do better, and indeed Theorem 5.7 shows that if the *effective rank* of $\Sigma_b$ is sufficiently small, the risk of the DARE predictor will approach that of the minimax predictor as $\mathcal{O}(E^{-1/2})$.

## 5.3 APPLYING DARE TO JIT-UDA

So far, we have only considered a setting with no knowledge of the test domain. As discussed in Section 3, we'd expect that estimating the adjustment via unlabeled samples will improve performance. Prior works have extensively explored how to leverage access to unlabeled test samples for improved generalization—but while some suggest ways of using unlabeled samples at test-time, they are not truly "just-in time", nor have the advantages been formally quantified. Our final theorem investigates the provable benefits of using the empirical moments instead of enforcing invariance, giving a finite-sample convergence guarantee for the unconstrained DARE objective in the JIT-UDA setting:

**Theorem 5.9** (JIT-UDA, shortened). *Let $\Sigma_S, \Sigma_T$ be the covariances of the source and target distributions, respectively. Define $m(\Sigma) := \frac{\lambda_{\max}(\Sigma)}{\lambda^3_{\min}(\Sigma)}$. Assume we observe $n_s = \Omega(m(\Sigma_S)d^2)$ source samples and analogous $n_T$ target samples. Then after solving the* DARE *objective, with probability at least $1 - 3d^{-1}$, the excess squared risk of our predictor on the new environment is bounded as $\mathcal{R}_T(\hat{\beta}) = \mathcal{O}\left(d^2\|\mu_T\|^2\left(\frac{m(\Sigma_S)}{n_S} + \frac{m(\Sigma_T)}{n_T}\right)\right)$.*

Experimentally, we found that DARE does not outperform methods specifically intended for UDA, possibly because $n \ll d^2$—but we believe this is a promising direction for future research, since it doesn't require unlabeled samples at train-time and it can incorporate new data on-the-fly.

# 6 EXPERIMENTS

| Dataset / Algorithm | Mean Accuracy by Domain ($\pm$ 90% CI) | | | | Dataset / Algorithm | Mean Accuracy by Domain ($\pm$ 90% CI) | | | | | |
|---|---|---|---|---|---|---|---|---|---|---|---|
| Office-Home | A | C | P | R | VLCS | C | L | S | V | | |
| ERM | $61.4 \pm 1.9$ | $52.1 \pm 1.6$ | $74.6 \pm 0.7$ | $75.8 \pm 1.9$ | ERM | $97.6 \pm 0.7$ | $\mathbf{66.1 \pm 0.6}$ | $71.9 \pm 1.9$ | $76.1 \pm 2.3$ | | |
| IRM | $62.1 \pm 1.5$ | $53.2 \pm 1.7$ | $75.2 \pm 0.9$ | $77.3 \pm 0.8$ | IRM | $\underline{98.6 \pm 0.6}$ | $64.9 \pm 0.9$ | $72.9 \pm 0.7$ | $74.1 \pm 2.1$ | | |
| GroupDRO | $62.5 \pm 0.5$ | $53.1 \pm 1.7$ | $\underline{75.7 \pm 0.5}$ | $77.7 \pm 1.1$ | GroupDRO | $\underline{98.5 \pm 0.7}$ | $65.0 \pm 0.5$ | $73.9 \pm 1.0$ | $75.1 \pm 1.8$ | | |
| DARE | $\mathbf{64.7 \pm 0.8}$ | $\mathbf{55.4 \pm 1.1}$ | $\mathbf{77.5 \pm 0.3}$ | $\mathbf{79.2 \pm 0.5}$ | DARE | $\underline{98.5 \pm 0.2}$ | $62.5 \pm 1.6$ | $74.3 \pm 1.4$ | $75.5 \pm 1.4$ | | |
| PACS | A | C | P | S | DomainNet | c | i | p | q | r | s |
| ERM | $84.3 \pm 1.5$ | $79.5 \pm 1.9$ | $96.7 \pm 0.5$ | $74.9 \pm 3.7$ | ERM | $57.6 \pm 0.6$ | $17.9 \pm 0.6$ | $44.7 \pm 0.7$ | $12.6 \pm 0.9$ | $59.0 \pm 0.7$ | $48.6 \pm 0.3$ |
| IRM | $83.6 \pm 0.4$ | $79.2 \pm 0.5$ | $97.1 \pm 0.2$ | $76.4 \pm 2.3$ | IRM | $\underline{60.9 \pm 0.4}$ | $\underline{19.3 \pm 0.2}$ | $\underline{47.6 \pm 0.4}$ | $12.4 \pm 0.4$ | $\underline{61.9 \pm 1.0}$ | $\underline{49.8 \pm 0.6}$ |
| GroupDRO | $83.6 \pm 1.0$ | $79.1 \pm 0.2$ | $96.9 \pm 0.3$ | $\underline{76.8 \pm 2.8}$ | GroupDRO | $57.8 \pm 0.7$ | $\underline{18.8 \pm 0.4}$ | $\underline{45.3 \pm 0.4}$ | $11.9 \pm 0.8$ | $\underline{59.5 \pm 1.0}$ | $47.9 \pm 0.9$ |
| DARE | $\mathbf{85.6 \pm 0.6}$ | $80.5 \pm 1.0$ | $96.6 \pm 0.4$ | $76.7 \pm 2.0$ | DARE | $\mathbf{61.7 \pm 0.2}$ | $\underline{19.3 \pm 0.1}$ | $\underline{47.4 \pm 0.5}$ | $\mathbf{13.1 \pm 0.7}$ | $\underline{61.2 \pm 1.3}$ | $\mathbf{51.4 \pm 0.3}$ |

Table 1: Performance of *linear* predictors on top of fixed features learned via ERM. Each letter is a domain. Because all algorithms use the same set of features for each trial, results are not independent. Therefore, **bold** indicates highest mean according to one-sided paired t-tests at $p = 0.1$ significance. If not the overall highest, underline indicates higher mean than ERM under the same test.

Most algorithms implemented in the DOMAINBED benchmark are only applicable to deep networks; many apply complex regularization to either the learned features or the network itself. We instead compare to three popular algorithms which work for linear classifiers: ERM (Vapnik, 1999), IRM (Arjovsky et al., 2019), and GroupDRO (Sagawa et al., 2020). We evaluate all approaches on four datasets: Office-Home (Venkateswara et al., 2017), PACS (Li et al., 2017), VLCS (Fang et al., 2013), and DomainNet (Peng et al., 2019). We find that DARE consistently matches or surpasses prior methods. A detailed description of the evaluation and comparison methodology can be found in Appendix D and evaluations of other end-to-end methods in Appendix E.

**Prior methods now consistently outperform ERM.** Interestingly, the previously observed gap between ERM and alternatives disappears in this setting with a linear predictor. Gulrajani & Lopez-Paz (2021) report IRM and GroupDRO performing much worse on DomainNet (5-10% lower accuracy on some domains) but they surpass ERM when using frozen features. This suggests they are more difficult to optimize over a deep network and when they do work, it is most likely due to learning a better linear classifier. This further motivates work on methods for learning simpler robust predictors.

# 7 RELATED WORK

A popular approach to domain generalization matches the domains in feature space, either by aligning moments (Sun & Saenko, 2016) or with an adversarial loss (Ganin et al., 2016), though these methods are known to be inadequate in general (Zhao et al., 2019). DARE differs from these approaches in that the constraint requires only that the *feature mean projection onto the vector $\beta$* be invariant. Domain-invariant projections (Baktashmotlagh et al., 2013) were recently analyzed by Chen & Bühlmann (2021), though notably under fully observed features and only for domain adaptation.

There has been intense recent focus on invariant prediction, based on ideas from causality (Peters et al., 2016) and catalyzed by IRM (Arjovsky et al., 2019). Though the goal of such methods is minimax-optimality under major distribution shift, later work identifies critical failure modes of this approach (Rosenfeld et al., 2021; Kamath et al., 2021). These methods eliminate features whose information is not invariant, which is often overly conservative. DARE instead allows for *limited* new variation by aligning the non-invariant subspaces, enabling stronger theoretical guarantees.

Some prior works "normalize" each domain by learning separate batchnorm parameters but sharing the rest of the network. This was initially suggested for UDA (Li et al., 2016; Bousmalis et al., 2016; Chang et al., 2019), which is not directly comparable to DARE since it requires unlabeled test data. This idea has also been applied to domain generalization (Seo et al., 2019; Segù et al., 2020) but in an ad-hoc manner. Because of the difficulty in training the network end-to-end, there is no consistent method for optimizing or validating the objective—in particular, all deep domain generalization methods were recently called into question when Gulrajani & Lopez-Paz (2021) gave convincing evidence that nothing beats ERM when evaluated fairly. Nevertheless, our analysis provides an initial justification for these methods, suggesting that this idea is worth exploring further.

LIMITATIONS

Our experiments demonstrate that traditional training already learns excellent features for generalizing to new distributions. This suggests that we should focus on methods learning to use these features, but it also means that these features are likely to contain just as much spurious or sensitive information as the original inputs. Thus, even if a method which uses frozen features performs better under shift (such as on minority subpopulations), care must still be taken to account for possible biases in the resulting classifier.

REPRODUCIBILITY STATEMENT

A major advantage to evaluating only a linear predictor is that we can release the exact model weights which were used to extract the frozen features on which all algorithms were trained. Code for our experiments will be released along with these models.

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

## A  DISCUSSION

### A.1  CONNECTION TO ANCHOR REGRESSION

The DARE objective for linear regression is written

$$\min_{\beta} \sum_{e \in \mathcal{E}} \mathbb{E}_{p^e}[(\beta^T \Sigma_e^{-1/2}(x - \mu_e) - y)^2] \qquad \text{s.t.} \quad \beta^T \Sigma_e^{-1/2} \mu_e = 0. \ \forall e \in \mathcal{E}. \qquad (3)$$

The idea of adjusting for domain projections has similarities to Anchor Regression (Rothenhäusler et al., 2021), an objective which linearly regresses separately on the projection and rejection of the data onto the span of a set of *anchor variables*. These variables represent some (known) measure of variability across the data, and the resulting solution enjoys robustness to pointwise-additive shifts in the underlying SCM. If we define the anchor variable to be a one-hot vector indicating a sample's environment, the Anchor Regression objective minimizes

$$\frac{1}{|\mathcal{E}|} \sum_{e \in \mathcal{E}} \mathbb{E}_{p^e} \left[ \ell(\beta^T(x - \mu_e), \ y - \mu_{y,e}) + \gamma \ell(\beta^T \mu_e, \ \mu_{y,e})) \right],$$

where $\ell$ is the squared loss and $\mu_{y,e} = \mathbb{E}_{p^e}[y]$. Here we see that Anchor Regression is "adjusting" in a sense, by regressing on the residuals, though the objective also regresses the mean prediction onto the target mean. Unfortunately, this requires access to the target mean, which is unavailable in logistic regression due to the lack of a good estimator for $\mathbb{E}\left[\log \frac{p(y=1|x)}{p(y=-1|x)}\right]$. Nevertheless, if we (i) assume the feature covariance is fixed for all environments and (ii) assume the target mean is zero for all environments, we observe that the above objective becomes equivalent to the Lagrangian form of DARE for linear regression (3). Alternatively, we could imagine combining the two by keeping Anchor Regression's use of separate target means while adding DARE's feature covariance whitening which would give

$$\frac{1}{|\mathcal{E}|} \sum_{e \in \mathcal{E}} \mathbb{E}_{p^e} \left[ \ell(\beta^T \Sigma_e^{-1/2}(x - \mu_e), \ y - \mu_{y,e}) + \gamma \ell(\beta^T \Sigma_e^{-1/2} \mu_e, \ \mu_{y,e}) \right],$$

However, this still requires us to estimate the target mean, so it is unclear if or how this objective could be applied to the task of classification.

### A.2  ON THE CONDITIONS ASSUMED WITHOUT LOSS OF GENERALITY

It may not be immediately clear why it is reasonable to assume both $V = I$ and $\Sigma_\epsilon = I$ WLOG simultaneously, so we clarify this point here. It is crucial to observe here that $\beta^*$ and $A_e$ do not need to be directly identifiable, because we only care about the predictive distribution $\beta^{*T} \epsilon$. We only need $A_e$ to be identifiable from $x$ up to equivalence of this distribution. So if for example we recover some invertible transformation $\hat{A}_e = A_e M$, this is not at all a problem because we also learn the corresponding $\hat{\beta} = M^T \beta^*$ such that $\hat{\beta}^T \hat{A}_e^{-1} x = \beta^{*T} M M^{-1} A_e^{-1} x = \beta^{*T} \epsilon$.

In particular, suppose instead $V \neq I$ and $\mathbb{E}[\epsilon_0 \epsilon_0^T] = \Sigma_0$. Then we can simply reparameterize as $\epsilon_0 \rightarrow \Sigma_0^{-1/2} \epsilon_0$, $b_e \rightarrow \Sigma_0^{1/2} b_e$, $\beta^* \rightarrow \Sigma_0^{1/2} \beta^*$, $A_e \rightarrow A_e V^{-1} \Sigma_0^{1/2}$. It is easy to see this results in the same observed distribution over $(x, y)$, and further that learning $\beta^*$ to predict on $\epsilon_0$ is the same as learning $\Sigma_0^{1/2} \beta^*$ to predict on $\Sigma_0^{-1/2} \epsilon_0$. So now we've reduced this to a setting where $\mathbb{E}[\epsilon_0 \epsilon_0^T] = I$ but perhaps $V \neq I$. However, when $V \neq I$ it represents precisely the unidentifiable transformation $M$ above, which does not pose a problem for prediction because it will not change in future environments.

## B  PROOFS OF MAIN RESULTS

### B.1  NOTATION

We use capital letters to denote matrices and lowercase to denote vectors or scalars, where the latter should be clear from context. $\| \cdot \|$ refers to the usual vector norm, or spectral norm for matrices. For a matrix $M$, we use $\lambda_{\max}(M)$ to mean its maximum eigenvalue—the minimum is defined analogously.

We write the pseudo-inverse as $M^\dagger$. For a collection of samples $\{x_i\}_{i=1}^n$, we frequently make use of the sample mean, $\hat{\mu} := \frac{1}{n}\sum x_i$, and the sample covariance, $\hat{\Sigma} := \frac{1}{n}\sum(x_i - \bar{\mu})(x_i - \bar{\mu})^T$. The notation $\lesssim$ means less than or equal to up to constant factors.

## B.2 STATEMENT AND PROOF OF LEMMA B.1, AND DISCUSSION OF RELATED RESULTS

**Lemma B.1.** *Assume our data follows a logistic regression model with regression vector $\beta^*$ and covariates $z \sim \mathcal{N}(0, I)$: $\log \frac{p(y=1|z)}{p(y=-1|z)} = \beta^{*T} z$. Then the solution to the dimension-constrained logistic regression problem*

$$\arg\min_\beta \ \mathbb{E}_{z,y}[-\log \sigma(y\beta^T z)] \qquad \text{s.t. } \beta_{S^c} = \mathbf{0},$$

*where $S \subseteq [d]$ indexes a subset of the dimensions, is equal to $\alpha\beta_S^*$ for some $\alpha \in (0, 1]$.*

*Proof.* The logistic regression model can be rewritten:

$$y \mid z = \mathbf{1}\{\beta^{*T} z + \epsilon > 0\},$$

where $\epsilon$ is drawn from a standard logistic distribution. If we are restricted to not use $z_{S^c}$, we can see that these can be modeled simply as an additional noise term. Thus, our new model is

$$y \mid z = \mathbf{1}\{\beta_S^{*T} z_S + \epsilon + \tau > 0\},$$

where $\tau := \beta_{S^c}^{*T} z_{S^c} \sim p$ is symmetric zero-mean noise, independent of $z_S$. Because we are now modeling the other dimensions as noise, moving forward we will drop the $S$ subscript, writing simply $\beta^{*T} z$. Define $F, f$ as the CDF and PDF of the distribution of $\epsilon' := \epsilon + \tau$. Then the MLE population objective can be written

$$\mathcal{L}(\beta) = -\mathbb{E}_z[\mathbb{E}_{\epsilon' \sim f(\epsilon')}[\mathbf{1}\{\beta^{*T} z + \epsilon' > 0\}\log\sigma(\beta^T z) + \mathbf{1}\{-(\beta^{*T} z + \epsilon') > 0\}\log\sigma(-\beta^T z)]].$$

For a fixed $z$, note that $\mathbb{E}_{\epsilon'}[\mathbf{1}\{\beta^{*T} z + \epsilon' > 0\}] = \mathbb{P}(\epsilon' \geq -\beta^{*T} z) = F(\beta^{*T} z)$ (since $f$ is symmetric), and therefore taking the derivative of this objective we get

$$\nabla_\beta \mathcal{L}(\beta) = -\nabla_\beta \mathbb{E}_z[F(\beta^{*T} z)\log\sigma(\beta^T z) + F(-\beta^{*T} z)\log\sigma(-\beta^T z)]$$
$$= \mathbb{E}_z[z \cdot (F(-\beta^{*T} z)\sigma(\beta^T z) - F(\beta^{*T} z)\sigma(-\beta^T z))]$$

Because $f$ is symmetric, we have $F(z) = 1 - F(-z)$, giving

$$\nabla_\beta \mathcal{L}(\beta) = \mathbb{E}_z\left[z \cdot (\sigma(\beta^T z) - F(\beta^{*T} z))\right].$$

Consider the directional derivative of the loss in the direction $\beta^*$, at the point $\beta = \alpha\beta^*$:

$$\beta^{*T}\nabla_\beta \mathcal{L}(\alpha\beta^*) = \mathbb{E}_z\left[\beta^{*T} z \cdot (\sigma(\alpha\beta^{*T} z) - F(\beta^{*T} z))\right].$$

Because $F$ is the CDF of a logistic distribution convolved with $p$, by Fubini's theorem we have

$$F(z) = \int_{-\infty}^z f(z)\ dz$$
$$= \int_{-\infty}^z \left[\int_{-\infty}^\infty p(\tau)\sigma'(z - \tau)\ d\tau\right] dz$$
$$= \int_{-\infty}^\infty p(\tau)\left[\int_{-\infty}^{z-\tau} \sigma'(\omega)\ d\omega\right] d\tau$$
$$= \int_{-\infty}^\infty p(\tau)\sigma(z - \tau)$$
$$= \mathbb{E}_{\tau \sim p}[\sigma(z - \tau)].$$

Further, because $p$ is symmetric, this is equal to $\frac{1}{2}\left(\mathbb{E}_{\tau \sim p}[\sigma(z - \tau)] + \mathbb{E}_{\tau \sim p}[\sigma(z + \tau)]\right)$. Thus, we have

$$\beta^{*T}\nabla_\beta \mathcal{L}(\alpha\beta^*) = \mathbb{E}_z\left[\beta^{*T} z \cdot \mathbb{E}_{\tau \sim p}\left[\sigma(\alpha\beta^{*T} z) - \frac{1}{2}\left(\sigma(\beta^{*T} z - \tau) + \sigma(\beta^{*T} z + \tau)\right)\right]\right].$$

We first consider the case where $\alpha = 1$. When $\beta^{*T} z > 0$, the term inside the expectation is positive for all $\tau \neq 0$, and vice-versa when $\beta^{*T} z < 0$ (this can be verified by writing the difference as a function of $\beta^{*T} z, \tau$, and observing that all the terms are non-negative except for a factor of $e^{\beta^{*T} z} - 1$). It follows that at the point $\beta = \beta^*$, $-\beta^*$ is a descent direction. Furthermore, since the objective is continuous in $\alpha$, we can follow this direction by reducing $\alpha$ (that is, moving in the direction $-\beta^*$) until the directional derivative vanishes.

Next, consider $\alpha = 0$. Then the directional derivative is

$$\beta^{*T} \nabla_\beta \mathcal{L}(0) = \frac{1}{2} \mathbb{E}_z \left[ \beta^{*T} z \cdot \mathbb{E}_{\tau \sim p} \left[ 1 - (\sigma(\beta^{*T} z - \tau) + \sigma(\beta^{*T} z + \tau)) \right] \right].$$

Here, when $\beta^{*T} z > 0$ the inner term is negative, and vice-versa for $\beta^{*T} z < 0$, implying that the directional derivative is now negative. Because the objective is convex, it follows that the optimal choice for $\alpha$ lies in $(0, 1]$, being equal to 1 when $\tau = 0$ almost surely.

It remains to show that the optimal vector has no other component orthogonal to $\beta^*$—in other words, that the solution is precisely $\alpha\beta^*$. For isotropic Gaussian $z$, we have for any $\delta$ perpendicular to $\beta^*$ that $\mathbb{E}[\delta^T z | \beta^{*T} z] = 0$. Therefore, the gradient in the direction $\delta$ is

$$\mathbb{E}_z[\delta^T z \cdot (\sigma(\alpha\beta^{*T} z) - F(\beta^{*T} z))] = \mathbb{E}_{\beta^{*T} z}[\mathbb{E}_{\delta^T z | \beta^{*T} z}[\delta^T z](\sigma(\alpha\beta^{*T} z) - F(\beta^{*T} z))]$$
$$= 0.$$

Since $\beta^*$ and all orthogonal directions form a complete basis, it follows that $\nabla \mathcal{L}(\alpha\beta^*) = 0$ and therefore that $\alpha\beta^*$ is the optimal solution. □

Though we prove this lemma only for Gaussian $z$, we found empirically that the result approximately holds whenever $z$ is dimension-wise independent and symmetric about the origin. We believe this is a consequence of the Central Limit Theorem: our proof relies on the conditional expectation of inner products with $z$ which converge to Gaussian in distribution as the dimensionality of $z$ grows.

We observe that Li & Duan (1989) prove a much simpler result under a general "linear conditional expectation" condition which is similar to the property we exploit regarding zero-mean conditional expectation of orthogonal inner products with isotropic Gaussians. Their result is more general, but it allows for any value of $\alpha$, including negative. In this case, we would actually be recovering the *opposite* effects of the ground truth, which is clearly insufficient for test-time prediction. Heagerty & Zeger (2000) give an analytical closed-form for the solution under a probit model with Gaussian noise; since this is not a logistic model, the Gaussian noise represents significantly less model misspecification, which explains why the exact closed-form is recoverable.

### B.3 PROOF OF THEOREM 5.2

**Theorem 5.2.** (Closed-form solution to the DARE population objective). *Under model (2), the solution to the DARE population objective (3) for linear regression is $\hat{\Pi}\beta^*$. If $\epsilon$ is Gaussian, then the solution for logistic regression (1) is $\alpha\hat{\Pi}\beta^*$ for some $\alpha \in (0, 1]$.*

*Proof.* Observe that under the constraint, regressing on the centered observations is equivalent to regressing on the non-centered observations (since the mean must have no effect on the output), so the solutions to these two objectives must be the same and have the same minimizers. We therefore consider the solution to the DARE objective but on non-centered observations.

It is immediate that the unconstrained solution to the DARE population objective on non-centered data is $\beta^*$ for both linear and logistic regression. For linear regression, we observe that because the adjusted covariates in each environment have identity covariance, the excess training risk of a predictor $\beta$ is exactly $\|\beta - \beta^*\|^2$. Therefore, the solution can be rewritten

$$\min_\beta \quad \|\beta - \beta^*\|^2$$

$$\text{s.t.} \quad \beta^T \Sigma_e^{-1/2} \mu_e = 0. \quad \forall e \in \mathcal{E}.$$

Recalling that $\Sigma_e^{-1/2} \mu_e = b_e$, the constraint can be written in matrix form as $\mathbb{B}^T \beta = \mathbf{0}$, and thus we see that the solution is the $\ell_2$-norm projection of $\beta^*$ onto the nullspace of $\mathbb{B}^T$ (i.e., the intersection of the nullspaces of the $b_e$). By definition, this is given by $(I - \mathbb{B}\mathbb{B}^\dagger)\beta^* = \hat{\Pi}\beta^*$.

To derive the closed-form for logistic regression, write the spectral decomposition $\mathbb{B} = UDV^T$, and consider regressing on $U^T\epsilon$ instead of $\epsilon$. As the predictor only affects the objective through its linear projection, the solution to this objective will be $U^T$ times the solution to the original objective (that is, for all vectors $v$, $(U^T\beta)^T U^T v = \beta^T v$). We will denote parameters for the rotated objective with a tilde, e.g. $\tilde{v} := U^T v$.

The constraint in Equation (1) is equivalent to requiring that the mean projection is a constant vector $c\mathbf{1}$ and, with the inclusion of a bias term, we can WLOG consider $c = 0$. Thus, the constraint can be written $\tilde{\mathbb{B}}^T\tilde{\beta} = VD\tilde{\beta} = \mathbf{0} \iff D\tilde{\beta} = \mathbf{0}$. We can therefore see that this constraint is the same as requiring that the dimensions of $\tilde{\beta}$ corresponding to the non-zero dimensions of $D$ are 0.

Noting that $U^T\epsilon \sim \mathcal{N}(0, I)$, we now apply Lemma B.1 to see that the solution will be $\tilde{\beta} = \alpha(I - DD^\dagger)\tilde{\beta}^*$ for some $\alpha \in (0, 1]$. Finally, as argued above we can recover the solution to the original objective by rotating back, giving the solution $\beta = U\tilde{\beta} = \alpha U(I - DD^\dagger)U^T\beta^* = \alpha\hat{\Pi}\beta^*$. $\qquad\square$

### B.4 PROOF OF THEOREM 5.6

**Theorem B.2** (Theorem 5.6, restated). *For any $\rho \geq 0$, denote the set of possible test environments $\mathcal{A}_\rho$ which contains all parameters $(A_{e'}, b_{e'})$ subject to Assumptions 5.3 and 5.5 and a bound on the mean: $\|b_{e'}\| \leq \rho$. For logistic or linear regression, let $\hat{\beta}$ be the minimizer of the corresponding DARE objective (1) or (3). Then,*

$$\sup_{(A_{e'}, b_{e'}) \in \mathcal{A}_\rho} \mathcal{R}_{e'}(\hat{\beta}) = (1 + \rho^2)(\|\beta^*\|^2 + 2B\|\beta^*_{\hat{\Pi}}\|\|\beta^*_{I-\hat{\Pi}}\|).$$

*Furthermore, the DARE solution is minimax:*

$$\hat{\beta} \in \arg\min_{\beta \in \mathbb{R}^d} \sup_{(A_{e'}, b_{e'}) \in \mathcal{A}_\rho} \mathcal{R}_{e'}(\beta).$$

*Proof.* Recall that in an environment $e$, $\mathbb{E}_e[y \mid x] = \beta^{*T}\Sigma_e^{-1/2}x$. So, for an environment $e'$ and predictor $\hat{\beta}$, we have the following excess risk decomposition:

$$\mathcal{R}_{e'}(\hat{\beta}) = \mathbb{E}_{e'}[(\hat{\beta}^T\bar{\Sigma}^{-1/2}x' - \beta^{*T}\Sigma_{e'}^{-1/2}x')^2]$$

$$= \underbrace{\mathbb{E}_{e'}[(\hat{\beta}^T\bar{\Sigma}^{-1/2}(x' - \mu_{e'}) - \beta^{*T}\Sigma_{e'}^{-1/2}(x' - \mu_{e'}))^2]}_{T_1} + \underbrace{\mathbb{E}_{e'}[(\hat{\beta}^T\bar{\Sigma}^{-1/2}\mu_{e'} - \beta^{*T}\Sigma_{e'}^{-1/2}\mu_{e'})^2]}_{T_2}.$$

Observe that term $T_1$ does not depend on the mean $b_{e'}$.

Term $T_2$ simplifies to

$$\mathbb{E}_{e'}[(\hat{\beta}^T\bar{\Sigma}^{-1/2}\mu_{e'} - \beta^{*T}\Sigma_{e'}^{-1/2}\mu_{e'})^2] = \left(\overbrace{(\Sigma_{e'}^{1/2}\bar{\Sigma}^{-1/2}\hat{\beta} - \beta^*)}^{v}{}^T b_{e'}\right)^2,$$

and so we can write a supremum over $T_2$ as

$$\sup_{\mathcal{A}_\rho} T_2 = \sup_{\mathcal{A}_\rho} (v^T b_{e'})^2$$

$$= \rho^2 \sup_{\mathcal{A}_\rho} \|v\|^2.$$

Next, observe that $T_1$ simplifies to

$$\hat{\beta}^T\bar{\Sigma}^{-1/2}\Sigma_{e'}\bar{\Sigma}^{-1/2}\hat{\beta} + \|\beta^*\|^2 - 2\hat{\beta}^T\bar{\Sigma}^{-1/2}\Sigma_{e'}^{1/2}\beta^* = \|\Sigma_{e'}^{1/2}\bar{\Sigma}^{-1/2}\hat{\beta} - \beta^*\|^2$$

$$= \|v\|^2.$$

So, returning to the full loss and recalling that $T_1$ is independent of $b_{e'}$, we have

$$\sup_{\mathcal{A}_\rho} \mathcal{R}_{e'}(\hat{\beta}) = \sup_{\mathcal{A}_\rho} T_1 + T_2$$

$$= (1 + \rho^2) \sup_{\mathcal{A}_\rho} \|v\|^2.$$

Of course, the ideal would be for a given environment $e'$ to set $\hat{\beta} := \bar{\Sigma}^{1/2}\Sigma_{e'}^{-1/2}\beta^* \implies v = 0$, but we have to choose a single $\hat{\beta}$ for all possible environments $e'$ parameterized by $(A_{e'}, b_{e'}) \in \mathcal{A}_\rho$. We will show that the choice of $\hat{\beta} := \alpha\hat{\Pi}\beta^*$ is minimax-optimal under this set for any $\alpha \in (0, 1]$.

Leaving the supremum over adversary choices implicit, we can rewrite the squared norm of $v$ as

$$v^T v = ((\Delta + I)\hat{\beta} - \beta^*)^T((\Delta + I)\hat{\beta} - \beta^*)$$
$$= \hat{\beta}^T \Delta^T \Delta \hat{\beta} + \|\hat{\beta} - \beta^*\|^2 + 2\hat{\beta}^T \Delta^T(\hat{\beta} - \beta^*).$$

By Lemma C.4, we can define an environment by defining the block terms of $U_{\hat{\Pi}}\Delta U_{\hat{\Pi}}^T$ directly. Consider the choice of $\Delta_1 = \lambda\frac{\hat{\beta}_{I-\hat{\Pi}}\hat{\beta}_{I-\hat{\Pi}}^T}{\|\hat{\beta}_{I-\hat{\Pi}}\|^2}, \Delta_2 = \Delta_{12} = \Delta_{21} = \mathbf{0}$. This choice is in $\mathcal{A}_\rho$ for any $\lambda \in \mathbb{R}$ since it is block-diagonal and $\|\Delta_2\| = 0$. Now we can write

$$v^T v = \lambda^2\|\hat{\beta}_{I-\hat{\Pi}}\|^2 + \|\hat{\beta} - \beta^*\|^2 + 2\lambda\hat{\beta}_{I-\hat{\Pi}}^T(\hat{\beta}_{I-\hat{\Pi}} - \beta^*_{I-\hat{\Pi}})$$
$$\geq \lambda^2\|\hat{\beta}_{I-\hat{\Pi}}\|^2 - 2\lambda\|\hat{\beta}_{I-\hat{\Pi}}\|\|\hat{\beta} - \beta^*\|),$$

via Cauchy-Schwarz and the triangle inequality. So, taking $\lambda \to \infty$ means that the minimax risk is unbounded unless $\hat{\beta}_{I-\hat{\Pi}} = 0 \iff \hat{\beta} = \hat{\beta}_{\hat{\Pi}}$. For the remainder of the proof we consider only this case.

With this restriction, we have

$$\|v\|^2 = \|(\Delta + I)\hat{\beta}_{\hat{\Pi}} - \beta^*\|^2$$
$$= \|(\Delta + I)\hat{\beta}_{\hat{\Pi}} - \beta^*_{\hat{\Pi}} - \beta^*_{I-\hat{\Pi}}\|^2$$
$$= \|(\Delta + I)\hat{\beta}_{\hat{\Pi}} - \beta^*_{\hat{\Pi}}\|^2 + \|\beta^*_{I-\hat{\Pi}}\|^2 - 2\beta^{*T}_{I-\hat{\Pi}}\Delta\hat{\beta}_{\hat{\Pi}}.$$

Assumption 5.3 implies that

$$|\beta^{*T}_{I-\hat{\Pi}}\Delta\hat{\beta}_{\hat{\Pi}}| = |\beta^{*T}U_{\hat{\Pi}}(I - S_{\hat{\Pi}})U_{\hat{\Pi}}^T \Delta U_{\hat{\Pi}}S_{\hat{\Pi}}U_{\hat{\Pi}}^T\hat{\beta}|$$
$$= \left|\beta^{*T}U_{\hat{\Pi}}(I - S_{\hat{\Pi}})\begin{bmatrix}\Delta_1 & \Delta_{12} \\ \Delta_{21} & \Delta_2\end{bmatrix}S_{\hat{\Pi}}U_{\hat{\Pi}}^T\hat{\beta}\right|$$
$$\leq B\|\beta^*_{\hat{\Pi}}\|\|\beta^*_{I-\hat{\Pi}}\|,$$

Consider the DARE solution $\hat{\beta} = \alpha\beta^*_{\hat{\Pi}}$ for $\alpha \in (0, 1]$. Then using the equivalent supremized set from Assumption 5.5 and Lemma C.3, the worst-case risk of this choice is

$$\sup_{\mathcal{A}_\rho} \mathcal{R}_{e'}(\hat{\beta}) = (1 + \rho^2) \sup_{\|\Delta\beta^*_{\hat{\Pi}}\|^2 < \|\beta^*_{\hat{\Pi}}\|^2}(\|(\alpha\Delta + (\alpha - 1)I)\beta^*_{\hat{\Pi}}\|^2 + \|\beta^*_{I-\hat{\Pi}}\|^2 + 2B\|\beta^*_{\hat{\Pi}}\|\|\beta^*_{I-\hat{\Pi}}\|)$$

$$\tag{4}$$

$$= (1 + \rho^2)(\|\beta^*_{\hat{\Pi}}\|^2 + \|\beta^*_{I-\hat{\Pi}}\|^2 + 2B\|\beta^*_{\hat{\Pi}}\|\|\beta^*_{I-\hat{\Pi}}\|).$$

It remains to show that any other choice of $\hat{\beta}$ results in greater risk. Observe that the second two terms of (4) are constant with respect to $\Delta$, so we focus on the first term. That is, we show that any other choice results in $\sup_{\|\Delta\beta^*_{\hat{\Pi}}\|^2 < \|\beta^*_{\hat{\Pi}}\|^2} \|(\Delta + I)\hat{\beta}_{\hat{\Pi}} - \beta^*_{\hat{\Pi}}\|^2 > \|\beta^*_{\hat{\Pi}}\|^2$ (except for $\hat{\beta} = \mathbf{0}$, which we address at the end).

For any choice of $\hat{\beta}_{\hat{\Pi}}$, decompose the vector into its projection and rejection onto $\beta^*_{\hat{\Pi}}$ as $\hat{\beta}_{\hat{\Pi}} = \alpha\beta^*_{\hat{\Pi}} + \delta$, with $\delta^T\beta^*_{\hat{\Pi}} = 0$. The adversary can choose $\Delta_2 = \lambda\delta\delta^T$, which lies in $\mathcal{A}_\rho$ for any $\lambda$. Then taking $\lambda \to \infty$ causes risk to grow without bound—it follows that we must have $\delta = 0$.

Next, consider any choice $\alpha \notin (0, 1]$. If $\alpha > 2$ or $\alpha < 0$, choosing $\Delta_2 = 0$ makes this term $(\alpha - 1)^2\|\beta^*_{\hat{\Pi}}\|^2 > \|\beta^*_{\hat{\Pi}}\|^2$. If $2 \geq \alpha > 1$, choosing $\Delta_2 = \frac{1}{\alpha}I$ makes this term $\alpha^2\|\beta^*_{\hat{\Pi}}\|^2 > \|\beta^*_{\hat{\Pi}}\|^2$.

Finally, if $\alpha = 0$ then this term is equal to $\|\beta^*_{\hat{\Pi}}\|^2$—however, this value is the *supremum* of the adversarial risk of the DARE solution and it cannot actually be attained. $\qquad\square$

## B.5 PROOF OF THEOREM 5.7

**Theorem B.3** (Theorem 5.7, restated). *Fix test environment parameters $A_{e'}, b_{e'}$ and our guess $\bar{\Sigma}$. Suppose we minimize the* DARE *regression objective (3) on environments whose means $b_e$ are Gaussian vectors with covariance $\Sigma_b$, with span$(\Sigma_b)$ = span$(I - \Pi)$. After seeing $E$ training domains:*

*1. If $E \geq \text{rank}(\Sigma_b)$ then* DARE *recovers the minimax-optimal predictor almost surely: $\hat{\beta} = \beta_{\Pi}^*$.*

*2. Otherwise, if $E \geq r(\Sigma_b)$ then with probability $\geq 1 - \delta$,*

$$\mathcal{R}_{e'}(\hat{\beta}) \leq \mathcal{R}_{e'}(\beta_{\Pi}^*) + \mathcal{O}\left(\frac{\|\Sigma_b\|}{\xi(\Sigma_b)}\left(\sqrt{\frac{r(\Sigma_b)}{E}} + \max\left\{\sqrt{\frac{\log 1/\delta}{E}}, \frac{\log 1/\delta}{E}\right\}\right)\right),$$

*where $\mathcal{O}(\cdot)$ hides dependence on $\|\Delta\|$.*

*Proof.* Define $\hat{\Pi}_E$ as the projection onto the nullspace of $\mathbb{B}^T$ after having seen $E$ environments. Item 1 is immediate, since as soon as we observe $E = \text{rank}(\Sigma_b)$ linearly independent $b_e$ we have that span$(\mathbb{B})$ = span$(\Sigma_b)$ and therefore $\hat{\Pi}_E = \Pi$ (this occurs almost surely for any absolutely continuous distribution). To prove item 2, we will write the solution learned after seeing $E$ environments as $\hat{\beta}_E := \hat{\Pi}_E \beta^*$. We can write the excess risk of the ground-truth minimax predictor $\beta_{\Pi}^*$ as

$$\mathcal{R}_{e'}(\beta_{\Pi}^*) = \mathbb{E}[(\beta_{\Pi}^{*T}\bar{\Sigma}^{-1/2}\Sigma_{e'}^{1/2}\epsilon - \beta^{*T}\epsilon)^2]$$
$$= \mathbb{E}[(\beta^{*T}\Pi\bar{\Sigma}^{-1/2}\Sigma_{e'}^{1/2}\epsilon - \beta^{*T}\epsilon)^2],$$

and likewise we have

$$\mathcal{R}_{e'}(\hat{\beta}_E) = \mathbb{E}[(\hat{\beta}_E^T\bar{\Sigma}^{-1/2}\Sigma_{e'}^{1/2}\epsilon - \beta^{*T}\epsilon)^2]$$
$$= \mathbb{E}[(\beta^{*T}\hat{\Pi}_E\bar{\Sigma}^{-1/2}\Sigma_{e'}^{1/2}\epsilon - \beta^{*T}\epsilon)^2].$$

Taking the difference,

$$\mathcal{R}(\hat{\beta}_E) - \mathcal{R}(\beta_{\Pi}^*) = \mathbb{E}[(\beta^{*T}\hat{\Pi}_E\overbrace{\bar{\Sigma}^{-1/2}\Sigma_{e'}^{1/2}\epsilon}^{:=v} - \beta^{*T}\epsilon)^2 - (\beta^{*T}\Pi\bar{\Sigma}^{-1/2}\Sigma_{e'}^{1/2}\epsilon - \beta^{*T}\epsilon)^2]$$
$$\leq \|(\Pi - \hat{\Pi}_E)\mathbb{E}[vv^T](2I - \Pi - \hat{\Pi}_E)\| + 2\|(\Pi - \hat{\Pi}_E)\mathbb{E}[v\epsilon^T]\|$$
$$\leq 2\left[\|(\Pi - \hat{\Pi}_E)\mathbb{E}[vv^T]\| + \|(\Pi - \hat{\Pi}_E)\mathbb{E}[v\epsilon^T]\|\right].$$

Now we note that

$$\mathbb{E}[vv^T] = \mathbb{E}[(\bar{\Sigma}^{-1/2}\Sigma_{e'}^{1/2}\epsilon)(\bar{\Sigma}^{-1/2}\Sigma_{e'}^{1/2}\epsilon)^T]$$
$$= \bar{\Sigma}^{-1/2}\Sigma_{e'}\bar{\Sigma}^{-1/2}$$

and

$$\mathbb{E}[v\epsilon^T] = \bar{\Sigma}^{-1/2}\Sigma_{e'}^{1/2}.$$

These matrices are bounded by $\|\Delta + I\|^2, \|\Delta + I\|$ respectively and are constant with respect to the training environments we sample. It follows that we can bound the risk difference as

$$\mathcal{R}(\hat{\beta}_E) - \mathcal{R}(\beta_{\Pi}^*) \lesssim \|\hat{\Pi}_E - \Pi\|,$$

and all that remains is to bound the term $\|\hat{\Pi}_E - \Pi\|$.

Combining Theorems 4 and 5 from Koltchinskii & Lounici (2017) with the triangle inequality, we have that when $r(\Sigma_b) \leq E$, with probability $\geq 1 - \delta$,

$$\|\bar{\Sigma} - \Sigma_b\| \lesssim \|\Sigma_b\|\left(\max\left\{\sqrt{\frac{\log 1/\delta}{E}}, \frac{\log 1/\delta}{E}\right\} + \max\left\{\sqrt{\frac{r(\Sigma_b)}{E}}, \frac{r(\Sigma_b)}{E}\right\}\right).$$

Since $r(\Sigma_b) \leq E$, the first term of the second max dominates. Further, Corrolary 3 of Yu et al. (2014), a variant of the Davis-Kahan theorem, gives us

$$\|\hat{\Pi}_E - \Pi\| \lesssim \frac{\|\bar{\Sigma} - \Sigma_b\|}{\xi(\Sigma_b)}.$$

Combining these facts gives the result. □

### B.6 PROOF OF THEOREM 5.9

**Theorem B.4** (Theorem 5.9, fully stated). *Assume the data follows model (2) with $\epsilon \sim \mathcal{N}(0, I)$. Observing $n_S$ samples from a source distribution $S$ with covariance $\Sigma_S$, we use half to estimate $\hat{\Sigma}_S$ and the other half to learn parameters $\beta$ which minimize the unconstrained ($\lambda = 0$), uncentered DARE regression objective. At test-time, given $n_T$ samples $\{x_i\}_{i=1}^{n_T}$ from the target distribution $T$ with mean and covariance $\mu_T, \Sigma_T$, we predict $f(x; \beta) = \beta^T \hat{\Sigma}_T^{-1/2} x$.*

*Define $m(\Sigma) := \frac{\lambda_{\max}(\Sigma)}{\lambda_{\min}^3(\Sigma)}$, and assume $n_S = \Omega(m(\Sigma_S) d^2)$, $n_T = \Omega(m(\Sigma_T) d^2)$. Then with probability at least $1 - 3d^{-1}$, the excess squared error of our predictor on the new environment is bounded as*

$$\mathcal{R}_T(f) = \mathcal{O}\left((1 + \|\mu_T\|^2)\left(\frac{\mathbb{E}[\eta^2]}{1 - \mathcal{O}\left(\sqrt{\frac{d}{n_S}}\right)}\frac{d}{n_S} + d^2\left(\frac{m(\Sigma_S)}{n_S} + \frac{m(\Sigma_T)}{n_T}\right)\right)\right).$$

*Proof.* We begin by bounding the error of our solution $\|\beta - \beta^*\|$. Observe that with our estimate $\hat{\Sigma}_S$ of the source environment moments, we are solving linear regression with targets $\beta^{*T}\epsilon_i + \eta_i$ and covariates $\hat{x}_i = \hat{\Sigma}_S^{-1/2} x_i = \hat{\Sigma}_S^{-1/2} \Sigma_S^{1/2} \epsilon_i$. Thus, if we had access to the true gradient of the modified least-squares objective (which is not the same as assuming $n_S \to \infty$, because in that case we would have $\hat{\Sigma}_S \to \Sigma_S$), the solution would be

$$\mathbb{E}[\hat{x}\hat{x}^T]^{-1}\mathbb{E}[\hat{x}y] = \left(\hat{\Sigma}_S^{-1/2}\Sigma_S^{1/2}(I + \mu_T\mu_T^T)\Sigma_S^{1/2}\hat{\Sigma}_S^{-1/2}\right)^{-1}\left(\hat{\Sigma}_S^{-1/2}\Sigma_S^{1/2}(I + \mu_T\mu_T^T)\beta^*\right)$$

$$= \hat{\Sigma}_S^{1/2}\Sigma_S^{-1/2}\beta^*.$$

Moving forward we denote this solution to the modified objective as $\hat{\beta} := \hat{\Sigma}_S^{1/2}\Sigma_S^{-1/2}\beta^*$, and further define $\Delta_S := \Sigma_S^{1/2}\hat{\Sigma}_S^{-1/2}$, with $\Delta_T$ defined analogously. A classical result tells us that the OLS solution is distributed as $\mathcal{N}\left(\hat{\beta}, \frac{\sigma_y^2}{n_S}M^{-1}\right)$, where $M$ is the modified covariance $\Delta_S^T\Delta_S$ and $\sigma_y^2 := \mathbb{E}[\eta^2]$. To show a rate of convergence to the OLS solution, we need to solve for the minimum eigenvalue $\lambda_{\min}(M)$—this will suffice since the above fact implies finite-sample convergence of the OLS estimator to the population solution. The well-known bound for sub-Gaussian random vectors tells us that with probability $\geq 1 - \delta_1$,

$$\|\beta - \hat{\beta}\| \lesssim \sqrt{\lambda_{\max}(M^{-1})\sigma_y^2}\left(\sqrt{\frac{d}{n_S}} + \sqrt{\frac{\log 1/\delta_1}{n_S}}\right).$$

and moving forward we condition on this event. Now let $\gamma_S := \sqrt{d/n_S}$, with $\gamma_T$ defined analogously. Since $M \succeq 0$, it follows that

$$\lambda_{\max}(M^{-1}) = \lambda_{\min}(\Delta_S^T\Delta_S)^{-1},$$

and further,

$$\lambda_{\min}(\Delta_S^T\Delta_S) \geq 1 - \|\Delta_S^T\Delta_S - I\|.$$

By Lemma C.1, we have that with probability $\geq 1 - d^{-1}$

$$\|\Delta_S^T\Delta_S - I\| \lesssim \gamma_S.$$

This implies that our solution's error can be bounded as

$$\|\beta - \beta^*\| = \|(\beta - \hat{\beta}) + (\hat{\beta} - \beta^*)\|$$

$$\leq \sqrt{\frac{\sigma_y^2}{1 - O(\gamma_S)}}\left(\sqrt{\frac{d}{n_S}} + \sqrt{\frac{\log 1/\delta_1}{n_S}}\right) + \|\hat{\Sigma}_S^{1/2}\Sigma_S^{-1/2} - I\|\|\beta^*\|$$

We assume $\|\beta^*\| = 1$ so we can avoid carrying it throughout the rest of the proof. Bounding the second term with Lemma C.2 gives

$$\|\beta - \beta^*\| \lesssim \sqrt{\frac{\sigma_y^2}{1 - O(\gamma_S)}} \left( \sqrt{\frac{d}{n_S}} + \sqrt{\frac{\log 1/\delta_1}{n_S}} \right) + \gamma_S \sqrt{d\, m(\Sigma_S)}.$$

On the target distribution, our excess risk with a predictor $\beta$ is

$$\mathcal{R}_{e'}(\beta) = \mathbb{E}[(\beta^T \hat{\Sigma}_T^{-1/2} x - \beta^{*T} \Sigma_T^{-1/2} x)^2]$$

$$= \mathbb{E} \left[ \left( \overbrace{(\Sigma_T^{1/2} \hat{\Sigma}_T^{-1/2}}^{\Delta_T} \beta - \beta^*)^T \epsilon \right)^2 \right]$$

$$= (\Delta_T \beta - \beta^*)^T (I + \mu_T \mu_T{}^T)(\Delta_T \beta - \beta^*)$$

$$\leq (1 + \|\mu_T\|^2)\|\Delta_T \beta - \beta^*\|^2$$

Now, we have

$$\|\Delta_T \beta - \beta^*\| = \|(\Delta_T \beta - \Delta_T \beta^*) + (\Delta_T \beta^* - \beta^*)\|$$

$$\leq \|\Delta_T(\beta - \beta^*)\| + \|(\Delta_T - I)\beta^*\|$$

$$\leq (1 + \|\Delta_T - I\|)\|\beta - \beta^*\| + \|\Delta_T - I\|.$$

Once again invoking Lemma C.2, with probability $\geq 1 - d^{-1}$,

$$\|\Delta_T - I\| \lesssim \gamma_T \sqrt{d\, m(\Sigma_T)},$$

and using this plus the previous result, the triangle inequality, and $(a + b)^2 \leq 2(a^2 + b^2)$ gives

$$\|\Delta_T \beta - \beta^*\|^2 \lesssim (1 + \|\Delta_T - I\|)^2 \|\beta - \beta^*\|^2 + \|\Delta_T - I\|^2$$

$$\lesssim (1 + \gamma_T \sqrt{d\, m(\Sigma_T)})^2 \|\beta - \beta^*\|^2 + (\|\Sigma^{1/2}\|\|\Sigma^{-1}\|^{3/2} \sqrt{d} \gamma_T)^2$$

$$\lesssim \|\beta - \beta^*\|^2 + \gamma_T^2 d\, m(\Sigma_T),$$

where the lower bound on $n_T$ enforces $\gamma_T \sqrt{d\, m(\Sigma_T)} \leq 1$. It follows that the excess risk can be bounded as

$$\mathcal{R}_{e'} \lesssim (1 + \|\mu_T\|^2)\|\Delta_T \beta - \beta^*\|^2$$

$$\lesssim (1 + \|\mu_T\|^2)\left(\|\beta - \beta^*\|^2 + \gamma_T^2 d\, m(\Sigma_T)\right).$$

Letting $\delta_1 = 1/d$, combining all of the above via union bound, and eliminating lower-order terms, we get

$$\mathcal{R}_{e'}(\beta) \lesssim (1 + \|\mu_T\|^2) \left( \frac{\sigma_y^2}{1 - \mathcal{O}(\gamma_S)} \frac{d}{n_S} + \gamma_S^2 d\, m(\Sigma_S) + \gamma_T^2 d\, m(\Sigma_T) \right)$$

$$= (1 + \|\mu_T\|^2) \left( \frac{\sigma_y^2}{1 - \mathcal{O}\left(\sqrt{\frac{d}{n_S}}\right)} \frac{d}{n_S} + d^2 \left( \frac{m(\Sigma_S)}{n_S} + \frac{m(\Sigma_T)}{n_T} \right) \right)$$

with probability $\geq 1 - 3d^{-1}$. $\qquad\square$

## C  TECHNICAL LEMMAS

**Lemma C.1.** *Suppose we observe $n \in \Omega(d + \log 1/\delta)$ samples $X \sim \mathcal{N}(\mu, \Sigma)$ with $\Sigma \succeq 0$ and estimate the inverse covariance matrix $\Sigma^{-1}$ with the inverse of the sample covariance matrix $\bar{\Sigma}^{-1}$. Then with probability $\geq 1 - \delta$, it holds that*

$$\|\bar{\Sigma}^{-1/2} \Sigma \bar{\Sigma}^{-1/2} - I\| \lesssim \sqrt{\frac{d + \log 1/\delta}{n}}.$$

*Proof.* As $\bar{\Sigma}^{-1/2}\Sigma\bar{\Sigma}^{-1/2}$ and $\Sigma^{1/2}\bar{\Sigma}^{-1}\Sigma^{1/2}$ have the same spectrum, it suffices to bound the latter. Observe that

$$\|\Sigma^{1/2}\bar{\Sigma}^{-1}\Sigma^{1/2} - I\| = \|\Sigma^{1/2}(\bar{\Sigma}^{-1} - \Sigma^{-1})\Sigma^{1/2}\|.$$

Now applying Theorem 10 of Kereta & Klock (2021) with $A = B = \Sigma^{1/2}$ yields the result. $\square$

**Lemma C.2.** *Assume the conditions of Lemma C.1. Then under the same event as Lemma C.1, it holds that*

$$\|\Sigma^{1/2}\bar{\Sigma}^{-1/2} - I\| \lesssim \|\Sigma^{1/2}\| \left( \|\Sigma^{-1}\|^{3/2}\sqrt{d}\gamma + \|\Sigma^{-1}\|^2\gamma^2 \right),$$

*where $\gamma = \sqrt{\frac{d+\log 1/\delta}{n}}$. In particular, if $\delta = d^{-1}$, then*

$$\|\Sigma^{1/2}\bar{\Sigma}^{-1/2} - I\| \lesssim \sqrt{\|\Sigma\|\|\Sigma^{-1}\|^3\frac{d^2}{n}}$$

*Proof.* We begin by deriving a bound for $\|\bar{\Sigma}^{-1/2} - \Sigma^{-1/2}\|$. Define $E = \bar{\Sigma}^{-1} - \Sigma^{-1}$. Observe that

$$\begin{aligned}
\|E\| &= \|\Sigma^{-1/2}\Sigma^{1/2}E\Sigma^{1/2}\Sigma^{-1/2}\| \\
&= \|\Sigma^{-1/2}(\Sigma^{1/2}\bar{\Sigma}^{-1}\Sigma^{1/2} - I)\Sigma^{-1/2}\| \\
&\leq \|\Sigma^{-1}\|\|\Sigma^{1/2}\bar{\Sigma}^{-1}\Sigma^{1/2} - I\|,
\end{aligned}$$

and now apply Lemma C.1 (since $\bar{\Sigma}^{-1/2}\Sigma\bar{\Sigma}^{-1/2}$ and $\Sigma^{1/2}\bar{\Sigma}^{-1}\Sigma^{1/2}$ have the same spectrum), giving

$$\|E\| \lesssim \|\Sigma^{-1}\|\gamma.$$

Let $UDU^T$ be the eigendecomposition of $\Sigma$, and define the matrix $[\sqrt{\cdot}, \alpha]_{i,j} = \frac{1}{\sqrt{D_{ii}}+\sqrt{D_{jj}}}$ as in Carlsson (2018). The Daleckii-Krein theorem (Daleckii & Krein, 1965) tells us that

$$\|\bar{\Sigma}^{-1/2} - \Sigma^{-1/2}\| = \|U([\sqrt{\cdot}, \alpha] \circ E)U^T\| + O(\|E\|^2).$$

Note that $\max_{i,j} |[\sqrt{\cdot}, \alpha]_{i,j}| = 1/2\sqrt{\lambda_{\min}(\Sigma)} \implies \|[\sqrt{\cdot}, \alpha]\| \leq \sqrt{d\|\Sigma^{-1}\|}$, and therefore by sub-multiplicativity of spectral norm under Hadamard product,

$$\begin{aligned}
\|\bar{\Sigma}^{-1/2} - \Sigma^{-1/2}\| &\lesssim \sqrt{d\|\Sigma^{-1}\|}\|E\| + \|E\|^2 \\
&\lesssim \|\Sigma^{-1}\|^{3/2}\sqrt{d}\gamma + \|\Sigma^{-1}\|^2\gamma^2.
\end{aligned}$$

Finally, noting that

$$\begin{aligned}
\|\Sigma^{1/2}\bar{\Sigma}^{-1/2} - I\| &= \|\Sigma^{1/2}(\bar{\Sigma}^{-1/2} - \Sigma^{-1/2})\| \\
&\leq \|\Sigma^{1/2}\|\|\bar{\Sigma}^{-1/2} - \Sigma^{-1/2}\|
\end{aligned}$$

completes the main proof. To see the second claim, note that $n \geq 2\|\Sigma^{-1}\|$ implies $\|\Sigma^{-1}\|^{3/2}\sqrt{d} \geq \|\Sigma^{-1}\|^2\gamma$, meaning the first of the two terms dominates. $\square$

**Lemma C.3.** *Let $\bar{\Sigma}$ be fixed. Then Assumption 5.5 is satisfied if and only if $\|\Delta\beta^*_{\hat{\Pi}}\|^2 < \|\beta^*_{\hat{\Pi}}\|^2$.*

*Proof.* The claim follows by rewriting the risk terms. Recall that $\Delta = \Sigma_{e'}^{1/2}\bar{\Sigma}^{-1/2} - I$. Writing out the excess risk of the ground truth $\beta^*$ in the subspace $\hat{\Pi}$,

$$\begin{aligned}
\mathcal{R}_{p_{e'}}^{\hat{\Pi}}(\beta^*) &= \mathbb{E}_{p_{e'}} \left[ (\beta^{*T}_{\hat{\Pi}}\bar{\Sigma}^{-1/2}x - \beta^{*T}_{\hat{\Pi}}\Sigma_{e'}^{-1/2}x)^2 \right] \\
&= \mathbb{E}_{p_{e'}} \left[ (\beta^{*T}\hat{\Pi}\bar{\Sigma}^{-1/2}\Sigma_{e'}^{1/2}\epsilon - \beta^{*T}\hat{\Pi}\epsilon)^2 \right] \\
&= \mathbb{E}_{p_{e'}} \left[ (\beta^{*T}\hat{\Pi}\Delta^T\epsilon)^2 \right] \\
&= \|\Delta\hat{\Pi}\beta^*\|^2.
\end{aligned}$$

Next, the excess risk of the null vector $\hat{\beta} = \mathbf{0}$ in the same subspace is

$$\begin{aligned}
\mathcal{R}_{p_{e'}}^{\hat{\Pi}}(\mathbf{0}) &= \mathbb{E}_{p_{e'}}[(\beta^{*T}_{\hat{\Pi}}\Sigma_{e'}^{-1/2}x)^2] \\
&= \mathbb{E}_{p_{e'}}[(\beta^{*T}\hat{\Pi}\epsilon)^2] \\
&= \|\hat{\Pi}\beta^*\|^2. \qquad\qquad \square
\end{aligned}$$

**Lemma C.4.** *For a fixed $\bar{\Sigma}$, choosing an environmental covariance $\Sigma_{e'}$ is equivalent to directly choosing the error terms $\Delta_1, \Delta_2, \Delta_{12}, \Delta_{21}$.*

*Proof.* For a fixed $\bar{\Sigma}$, due to the unique definition of square root as a result of Assumption 5.1, the map $\Sigma_{e'} \rightarrow \Sigma_{e'}^{1/2} \bar{\Sigma}^{-1/2} - I$ is one-to-one. Recall that we can write

$$U_{\hat{\Pi}}^T \Delta U_{\hat{\Pi}} = \begin{bmatrix} \Delta_1 & \Delta_{12} \\ \Delta_{21} & \Delta_2 \end{bmatrix}$$

which defines a one-to-one map from $\Delta$ to the error terms. Since the composition of bijective functions is bijective, the claim follows. $\square$

# D  IMPLEMENTATION DETAILS

## D.1  EVALUATION PIPELINE

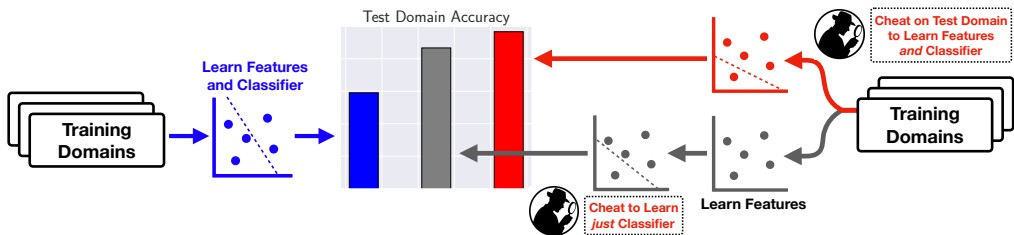

Figure 2: Depiction of evaluation pipeline. Standard training on train domains leads to poor performance. Cheating while training the full network (features and classifier) does substantially better. However, cheating on *just* the linear classifier does almost as well, implying that the features learned without cheating are already good enough for massive improvements over SOTA.

Figure 2 depicts the overall pipeline for evaluation of the different approaches we describe in Section 2. Our baselines are three distinct pipelines, each slightly different:

1. The first pipeline (dark blue in Figure 2) is the standard method of evaluating ERM on a new domain. We train the entire network (the combination of a feature embedder comprising all layers except the last linear layer, and the last layer linear classifier) end-to-end on the training domains, simultaneously learning the features and the linear classifier on the training domains via backpropagation. When evaluated on a new domain, this achieves state-of-the-art accuracy (Gulrajani & Lopez-Paz, 2021), but it still performs quite poorly.

2. The second pipeline (red) is very similar to the first, but we include the test domain among the domains on which the network is trained end-to-end. Because this means that the test distribution is one of the training domains, this simulates an "ideal" setting where no distribution shift occurs from train-time to test-time. As such, it is unsurprising that this approach performs substantially better (though it still leaves a bit to be desired—this raises a separate question about failures of in-distribution learning with multiple domains). As this approach requires cheating, it is completely unrealistic to expect current methods to even begin to approach this baseline, but it gives a good sense of what would be the best performance to hope for.

3. The final pipeline (grey) is our main experimental contribution. Here, the *features* (all but the last layer) are learned without cheating, as in the first pipeline. Next, we freeze the features and cheat only while learning a linear classifier on top of these features. Not only does this method do significantly better than the first baseline, it actually performs almost as well as—and sometimes better than—the second pipeline. Thus, we find that standard features learned via ERM without cheating are *already good enough for generalization* and that the main bottleneck to reaching the accuracy of the second baseline is in learning a good linear classifier *only*. This has several important advantages to current methods which attempt to change the entire end-to-end process, which we explicate in Section 2 in the main body.

## D.2  NUMERICAL COMPARISONS

The features used for each trial are the default hyperparameter sweep of DOMAINBED. For computational reasons, we used fewer random hyperparameter choices per trial, meaning the reported accuracies are not directly comparable. Nevertheless, our results are significant because they are consistently evaluated according to the same methodology.

All algorithms use the same set of features for each trial, so their performances are highly dependent. To account for this, we perform a one-sided paired t-test between algorithms to determine the best performers. The fact that DARE consistently bests ERM is somewhat surprising: it is intended to guard against a worst-case distribution shift and depends on the quality of our guess $\bar{\Sigma}$, so we would in general expect worse performance on some datasets.

### D.3 CODE DETAILS

All features were learned by finetuning a ResNet-50 using the default settings and hyperparameter sweeps of the DOMAINBED benchmark (Gulrajani & Lopez-Paz, 2021). We extracted features from 3 trials, with 5 random hyperparameter choices per trial, picking the one with the best training domain validation accuracy. We used the default random splits of 80% train / 20% test for each domain.

Using frozen features, the cheating linear classifier was trained by minimizing the multinomial logistic loss with full-batch SGD with momentum for 3000 steps. We did not use a validation set.

For the main experiments, all algorithms were trained using full-batch L-BFGS (Liu & Nocedal, 1989). We used the exact same optimization hyperparameters for all methods; the default learning rate of 1 was unstable when optimizing IRM, frequently diverging, so we lowered the learning rate until IRM consistently converged (which occurred at a learning rate of 0.2). Since the IRM penalty only makes sense with both a feature embedder *and* a classification vector, we used an additional linear layer for IRM, making the objective non-convex. Presumably due to their convexity for linear classifiers, all other methods were unaffected by this change. For all methods, we halted optimization once 20 epochs occurred with no increase in training domain validation accuracy; the maximum validation accuracy typically occurred within the first 5 epochs.

For stability when whitening (and because the number of samples per domain was often less than the feature dimension), in estimating $\hat{\Sigma}_e$ for each environment we shrank the sample covariance towards the identity. Specifically, we define $\hat{\Sigma}_e = (1-\rho)\frac{1}{n}\sum_{i=1}^{n} x_i x_i^T + \rho I$, with $\rho = 0.1$, and $\rho = 0.01$ for DomainNet due to its much larger size. We found that increasing $\lambda$ beyond $\sim 1$ had little-to-no effect on the accuracy, loss, *or* penalty of the DARE solution (see Appendix E for ablations). However, we did observe that choosing a very large value for $\lambda$ (e.g., $10^5$ or higher) could result in poor conditioning of the objective, with the result being that the L-BFGS optimizer took several epochs for the loss to begin going down.

## E   ADDITIONAL EXPERIMENTS

Here we present additional experimental results. All reported accuracies represent the mean of three trials, and all error bars (where present) display 90% confidence intervals calculated as $\pm 1.645 \frac{\hat{\sigma}}{\sqrt{n}}$. Note though that **the results are not independent across methods, so simply checking if the error bars overlap is overly conservative in identifying methods which perform better.**

### E.1   ACCURACY OF THE TEST COVARIANCE WHITENER

As we do not have access to the true test-time covariance, we estimate the adjustment with the average of the train-time adjustments. Table 2 reports the normalized squared Frobenius norm of our estimate's error to the sample covariance adjustment, defined as $\frac{\|\hat{W}-W\|_F^2}{\|W\|_F^2}$, where $W := \hat{\Sigma}_{e'}^{-1/2}$ is the sample covariance adjustment and $\hat{W} := \frac{1}{|\mathcal{E}|}\sum_{e\in\mathcal{E}} \Sigma_e^{-1/2}$ is our averaging estimate. We find that this normalized error is consistently small, meaning our estimated adjustment is reasonably close to the "true" adjustment, relative to its spectrum (we put "true" in quotation marks because the best we can do is estimate it via the sample covariance). This helps explain why our averaging adjustment performs so well in practice, though we expect future methods could improve on this estimate (particularly on the last domain of PACS).

### E.2   ALIGNMENT OF DOMAIN-SPECIFIC OPTIMAL CLASSIFIERS

As discussed in Section 3, DARE does not project out varying subspaces but rather aligns them such that the adjusted domains have similar optimal classifiers simultaneously. To verify that this is actually happening, we learn the optimal linear classifier for each domain individually and evaluate the inter-domain cosine similarity for these vectors for each class. We see that without adjustment, different domains' optimal linear decision boundaries have normals with small alignment on average, which explains why trying to learn a single linear classifier which does well on all domains simultaneously performs poorly in most cases. After alignment, the individually optimal classifiers are more aligned, which allows a single classifier to perform better on all domains. Furthermore, this is done without

| Dataset | Normalized Error |
|---|---|
| **Office-Home** | |
| A | 0.033 |
| C | 0.040 |
| P | 0.022 |
| R | 0.019 |
| **PACS** | |
| A | 0.052 |
| C | 0.016 |
| P | 0.025 |
| S | 0.217 |
| **VLCS** | |
| C | 0.094 |
| L | 0.039 |
| S | 0.051 |
| V | 0.029 |

Table 2: Mean normalized adjustment estimation error for each dataset and each train-domain/test-domain split.

throwing away the varying component (as would be done by invariant prediction (Peters et al., 2016)), allowing DARE to use more information and resulting in higher test accuracy.

| Dataset | With Adjustement | Without Adjustement |
|---|---|---|
| **Office-Home** | | |
| A | 0.693 | 0.596 |
| C | 0.710 | 0.691 |
| P | 0.695 | 0.636 |
| R | 0.642 | 0.590 |
| **PACS** | | |
| A | 0.903 | 0.863 |
| C | 0.896 | 0.776 |
| P | 0.905 | 0.827 |
| S | 0.903 | 0.791 |
| **VLCS** | | |
| C | 0.598 | 0.200 |
| L | 0.723 | 0.436 |
| S | 0.674 | 0.231 |
| V | 0.591 | 0.210 |

Table 3: Mean cosine similarity between linear classification vectors which are individually optimal for their respective domains (learned via logistic regression). For each dataset and each train-domain/test-domain split we report the average similarity across all class normal vectors and all domain pairs.

### E.3 RESULTS ON ADDITIONAL END-TO-END METHODS

Though ERM represents approximate state-of-the-art accuracy, for completeness we include a few additional end-to-end methods to confirm that this fact still holds under our experimental conditions.

We also include additional results on ColoredMNIST:

| Dataset / Algorithm | Mean Accuracy by Domain ($\pm$ 90% CI) | | | |
|---|---|---|---|---|
| Office-Home | A | C | P | R |
| ERM | $61.4 \pm 1.9$ | $52.1 \pm 1.6$ | $74.6 \pm 0.7$ | $75.8 \pm 1.9$ |
| Fish | $61.2 \pm 1.0$ | $50.1 \pm 1.7$ | $75.1 \pm 1.0$ | $76.5 \pm 0.2$ |
| CORAL | $61.1 \pm 0.7$ | $52.4 \pm 1.1$ | $75.2 \pm 0.7$ | $77.0 \pm 0.3$ |
| PACS | A | C | P | S |
| ERM | $84.3 \pm 1.5$ | $79.5 \pm 1.9$ | $96.7 \pm 0.5$ | $74.9 \pm 3.7$ |
| Fish | $83.8 \pm 2.6$ | $76.4 \pm 0.6$ | $96.6 \pm 1.2$ | $75.6 \pm 3.8$ |
| CORAL | $82.4 \pm 1.0$ | $78.6 \pm 1.3$ | $96.7 \pm 0.4$ | $73.0 \pm 2.9$ |
| VLCS | C | L | S | V |
| ERM | $97.6 \pm 0.7$ | $66.1 \pm 0.6$ | $71.9 \pm 1.9$ | $76.1 \pm 2.3$ |
| Fish | $98.8 \pm 0.4$ | $63.6 \pm 0.6$ | $72.0 \pm 1.6$ | $78.6 \pm 1.6$ |
| CORAL | $97.7 \pm 0.3$ | $64.0 \pm 2.0$ | $72.2 \pm 1.8$ | $74.0 \pm 0.7$ |

Table 4: Performance of other algorithms under our experimental conditions. Our results confirm that ERM remains approximate state of the art among end-to-end methods.

| Dataset / Algorithm | Mean Accuracy by Domain ($\pm$ 90% CI) | | |
|---|---|---|---|
| ColoredMNIST | 90% | 80% | 10% |
| ERM | $71.2 \pm 0.8$ | $72.8 \pm 0.2$ | $10.1 \pm 0.2$ |
| IRM | $71.3 \pm 0.6$ | $72.6 \pm 0.3$ | $10.1 \pm 0.1$ |
| GroupDRO | $71.2 \pm 0.5$ | $72.6 \pm 0.3$ | $10.1 \pm 0.1$ |
| DARE | $71.4 \pm 0.3$ | $72.9 \pm 0.1$ | $10.1 \pm 0.1$ |

Table 5: Performance on ColoredMNIST.

## E.4 ABLATIONS

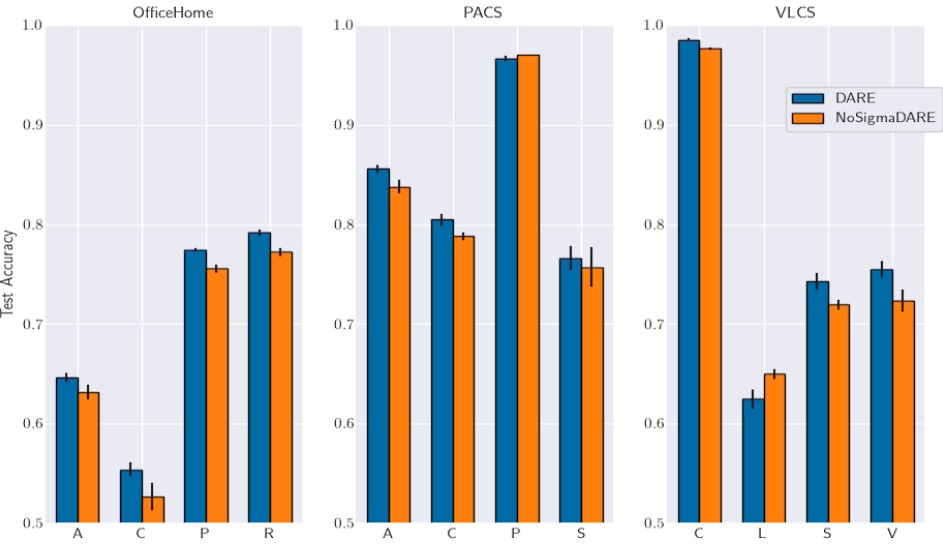

Figure 3: Demonstration of the effect of whitening. NoSigmaDARE is the exact same algorithm as DARE but with no covariance whitening. In almost all cases, covariance whitening + guessing at test-time results in better performance. We expect under much larger distribution shift that this pattern may reverse.

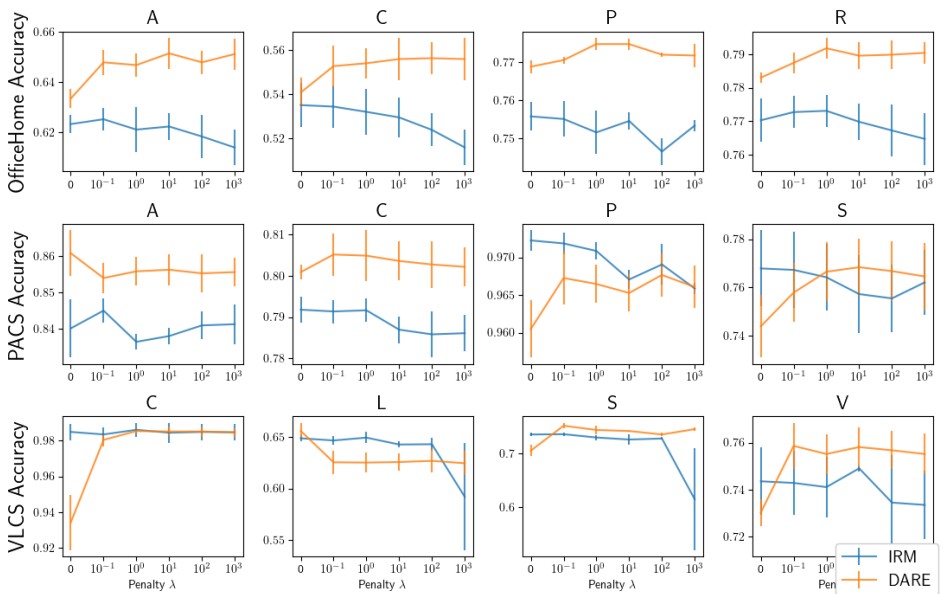

Figure 4: Effect of penalty term $\lambda$ on the two algorithms which use it. $\lambda = 0$ corresponds to no constraint, and the lower performance demonstrates that this invariance requirement is essential to the quality of the learned classifier. For $\lambda \neq 0$, DARE accuracy is extremely robust, effectively constant for all $\lambda \geq 1$; in practice we also found the penalty term itself to always be $\sim 0$. In contrast, IRM accuracy appears to *decrease* with increasing $\lambda$, implying that the observed benefit of IRM primarily comes from the domain reweighting as in our GroupERM method.

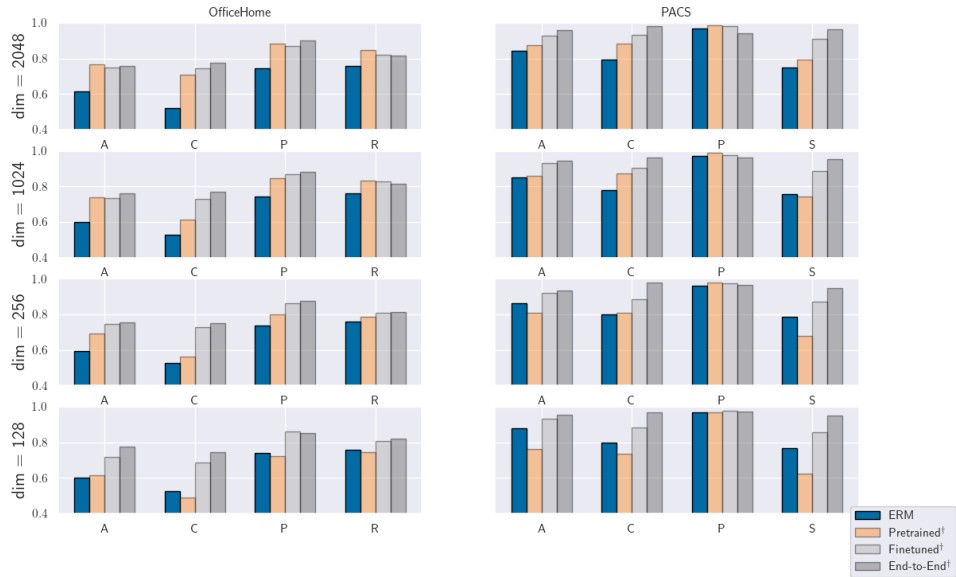

Figure 5: Effect of final feature bottleneck dimensionality on cheating accuracy. Reducing the dimensionality reduces accuracy of all methods to varying degrees, though in some cases it actually *increases* test accuracy. We observe that the main pattern persists, though the gap between cheating on finetuned features and traditional ERM shrinks as the dimensionality is reduced substantially. To reduce dimensionality of the pretrained features we use a random projection; unsurprisingly, the quality dramatically falls as the number of dimensions is reduced.

