# OpenReview forum: "Domain-Adjusted Regression or: ERM May Already Learn Features Sufficient for Out-of-Distribution Generalization"
_ICLR.cc/2023/Conference — Submitted to ICLR 2023_

### Official Review · Reviewer_aW4c · 2022-10-23

**Confidence:** 2
**Correctness:** 3
**Technical Novelty And Significance:** 2
**Empirical Novelty And Significance:** 2
**Recommendation:** 6

**Clarity, Quality, Novelty And Reproducibility:**

### Major concern on Motivation

The paper proposes a conjecture regarding feature learning and robust regression, which is clearly interesting and novel. However, a more comprehensive experiment or theoretical analysis is required, to fully validate this proposal. My previous experience is that whether ERM could learn sufficient information for generalization really depends. I was not fully convinced by the present experimental results, so I tried a quick experiment on the CMNIST dataset from the IRM paper. In particular, I followed the exact setting and the released codes of the IRM paper, and the results below clearly gives a counter-example to the motivation of this submission: ERM1 is the same setup as in the submission, but it does not learn well (acc in the cheating setup is lower than random guessing.); however, if feature layers are enabled to learn while the linear predictor layer is fixed, ERM2 can give the desired cheating result.

- ERM0: the released model has three layers, so I can treat the first two layers as **feature**, and the last linear layer as a **linear** predictor; train: ($p^e$=0.1, $p^e$=0.2) , test ($p^e$=0.9);

| step | train nll | train acc | train penalty | test acc |
| ---- | --------- | --------- | ------------- | -------- |
| 0    | 0.39170   | 0.83478   | 0.00749       | 0.18000  |
| 100  | 0.36110   | 0.85028   | 0.00374       | 0.11720  |
| 200  | 0.35049   | 0.85202   | 0.00362       | 0.11980  |
| 300  | 0.34190   | 0.85446   | 0.00356       | 0.12590  |
| 400  | 0.33326   | 0.85778   | 0.00353       | 0.13090  |
| 500  | 0.32384   | 0.86180   | 0.00355       | 0.13960  |
| 600  | 0.31265   | 0.86748   | 0.00367       | 0.15090  |

* ERM1: following the cheating setup in the submission, set train:($p^e$=0.1, $p^e$=0.9) , test ($p^e$=0.9); load ERM0 trained model, and only last linear layer is re-trained (the training steps is set to be large to make sure convergence, and only some intermediate results are listed hered.)

| Restart 0 | train nll | train acc | train penalty | test acc |
| --------- | --------- | --------- | ------------- | -------- |
| step      | train nll | train acc | train penalty | test acc |
| 0         | 1.00958   | 0.48872   | 0.60302       | 0.17940  |
| 100       | 0.81980   | 0.48718   | 0.20277       | 0.17520  |
| 500       | 0.68100   | 0.50126   | 0.00677       | 0.20390  |
| 1000      | 0.67899   | 0.51680   | 0.00583       | 0.23730  |
| 1500      | 0.67737   | 0.53232   | 0.00542       | 0.27350  |
| 2000      | 0.67627   | 0.54408   | 0.00514       | 0.30320  |
| 2500      | 0.67562   | 0.55112   | 0.00498       | 0.32080  |
| 3000      | 0.67528   | 0.55402   | 0.00490       | 0.32860  |
| 3500      | 0.67511   | 0.55556   | 0.00486       | 0.33170  |
| 4300      | 0.67506   | 0.55604   | 0.00485       | 0.33230  |
| 4400      | 0.67505   | 0.55608   | 0.00484       | 0.33230  |
| 4500      | 0.67505   | 0.55606   | 0.00484       | 0.33220  |
| 5000      | 0.67505   | 0.55610   | 0.00484       | 0.33220  |

- ERM2: following the cheating setup in the submission, set train:($p^e$=0.1, $p^e$=0.9) , test ($p^e$=0.9); load ERM0 trained model, and only the feature layers are re-trained.


| step | train nll | train acc | train penalty | test acc |
| ---- | --------- | --------- | ------------- | -------- |
| 0    | 1.00681   | 0.48788   | 0.59460       | 0.17580  |
| 100  | 0.56662   | 0.73568   | 0.00181       | 0.74280  |
| 200  | 0.54682   | 0.74500   | 0.00204       | 0.74440  |
| 300  | 0.52516   | 0.75480   | 0.00252       | 0.74430  |
| 400  | 0.50260   | 0.76696   | 0.00331       | 0.74310  |
| 500  | 0.48133   | 0.77864   | 0.00481       | 0.74090  |
| 600  | 0.46495   | 0.78962   | 0.00564       | 0.73570  |



## Other questions:

- authors wrote “The difference is that DARE does not learn a single featurizer which aligns the moments, but rather aligns the moments of already learned features” and the proposed method in (2): the method first whitens the learned features, so they still yield  the same moments. One can, for instance, add a new layer (or several layers) before the linear layer, so that they output the same moments,  In this sense, the proposed method is not particularly new.
- the proposed method assume that the mean is a multiple of all-one vector and in the test domain simply using the mean covariance matrix in train domains is less than justified. Again, I feel it not very hard to find a counter-example like above.
- regarding the proposed optimality compared invariant learning: I think it beneficial to mention the setting *earlier*, where the distribution shift may not be arbitrary, like in the anchor regression paper. Otherwise, it has been shown the causal parents could yield the minimax optimal predictor.  Also, I feel supervised that there is no discussion on anchor regression in the main text?
- Experiments:  many baselines are missing. Since you already use Resnet as the feature extractor, i did not see why only compare method with linear predictor.


**Strength And Weaknesses:**

Pros:
- an interesting observation, which could provide more insights
- the proposed method is easy to solve compared with other approaches
- a new model regarding data generations process for OOD generalization

Cons:
- the observation that ERM method have learnt sufficient features is not sufficiently justified. (I run an commonly-used dataset, which gives a counter-example). This makes the followed method, model, and analysis less convincing and useful.
- the proposed method is quite similar to other methods.
- benchmarked methods are really lacking.
- writing could be improved.

**Summary Of The Paper:**

The paper considers OOD generalization problem. In four benchmark datasets, authors observe that ERM may already learn sufficient features and then conclude that the current bottleneck is robust regression rather than feature learning. A method called domain-adjusted regression (DARE) is then proposed, which is easy to optimize based on its convexity nature (as it is a linear regression over learned, fixed features). Also motivated by this observation, authors consider a mathematical model of data generation and shows that DARE achieves a minimax optimal solution. Some experiments also validate usefulness of the proposed method.

**Summary Of The Review:**

To summarize,  the foundation and motivation of this paper are not necessarily true and are questionable. The followed results are therefore not very meaningful. Thus, I have to recommend REJ at this round. If authors can provide further evidence or show when such a conclusion can hold, I would like to raise my score.

=========after rebuttal=====
I thank the authors for detailed reponse and clarifications.

Let me resummarize a key point of the paper: that "ERM already learn sufficient features for generalization" is interesting, inspiring, and is likely to hold with modern DL artchitecture and training, but is not always guranteed and hard to check in practice. I suggest authors to discuss this point in more details in future versions, e.g., how to validate or falsify this claim in certain cases and the present limitations/chanllenges.

I'm now between weak rej and weak acc, and decide to increase my score to WA but with a lower confidence.

---

> ### Author Response · Authors · 2022-11-07
> **Addressing your experimental results**
>
> **To address your experimental results**
>
> We reran your experiment ourselves and found that our results do hold (see our experimental setup and numerical results below). **Relatedly, there are several issues with your experimental setup.** First we list our concerns with your experiment and conclusions:
>
> * The original IRM code is *not valid*. The hyperparameters were chosen by *evaluating on the test set*. Therefore, **any experiment which uses the IRM code or hyperparameters is flawed.** For a valid evaluation, we encourage you to check out DomainBed (https://github.com/facebookresearch/DomainBed), which implements IRM correctly (this is what we used).
>
> * Colored MNIST is an extremely unrealistic dataset which does not represent real-world tasks. As evidence of this, note that simply reducing the amount of label noise from 25% to 15% allows ERM to succeed with no difficulty. In other words, **CMNIST is more of a test of robustness to *label noise* than robustness to *distribution shift*.**
>
> * CMNIST uses a small MLP and is linearly separable, which does not capture the behavior of modern deep neural networks on realistic data.  In contrast, our experiments are on popular domain generalization benchmarks which consist of realistic tasks involving real-world images and modern architectures. Further, **work published after ours has made similar findings under related but different experimental setups, further supporting our claims [2, 3].**
>
> To reiterate: (i) the IRM code is fundamentally flawed, and even ignoring this, (ii) *our results **do** hold on CMNIST* (see below). Furthermore, (iii) several later works have provided additional support for our observation. Based on these points, it is clear that our experimental observations are quite robust and well supported by our work and others’.
>
> You’ve written that “If authors can provide further evidence or show when such a conclusion can hold, I would like to raise my score.” **We sincerely hope that you meant this, and that you will increase your score in light of our response. If not, can you explain what you still don’t believe, and suggest a concrete experiment that would constitute strong evidence for you?**
>
> [1] In Search of Lost Domain Generalization. Gulrajani & Lopez-Paz, 2021.
>
> [2] Last layer re-training is sufficient for robustness to spurious correlations. Kirichenko et al. 2022
>
> [3] On Feature Learning in the Presence of Spurious Correlations. Izmailov et al. 2022
>
>
> **Experimental results**
>
> Our experiment requires only a minor modification of DomainBed code and the sklearn python package. We encourage you to reproduce our results!
>
> Using the default DomainBed architecture and hyperparameters for a single run, we trained an MLP on only the first two environments of CMNIST ($p_e = 0.1$ and $p_e = 0.2$). This achieved a best training-domain validation accuracy of 85.15% on iteration 1700, with OOD accuracy 9.77%. We then saved the corresponding features and labels. Next, we concatenated the in-split features for all 3 domains (the 80% of each domain that the features were trained on, plus the 80% in-split from the unseen test domain).
>
> We trained a model with sklearn.linear_model.LogisticRegression (https://scikit-learn.org/stable/modules/generated/sklearn.linear_model.LogisticRegression.html) on these features using all default settings. This achieved a training accuracy of 73.22%. Then we evaluated on just the 20% out-split from the unseen domain ($p_e = .9$), and got an accuracy of 67.01%. Keep in mind that the best possible accuracy is 75%, and also that IRM or ERM don’t reach this observed 67% accuracy *even with oracle validation*.

---

> > ### Comment · Reviewer_aW4c · 2022-11-09
> > **follow-up on the motivation**
> >
> > Thanks for response and clarification. The former response addresses my other questions. I still want to have follow-up dicussion on the motivation.
> >
> > First, let me clarify: the experiments I did were using **ERM** which did not have cheating as in the IRM part.
> >
> > Regarding the third point: ''work published after ours has made similar findings under related but different experimental setups, further supporting our claims [2, 3].'' --- I checked the mentioned papers. (Note that I am reviewing this submission, not other papers that have been accepted somewhere)
> >
> > Still, I have questions on this motivation. Let me list the points you have mentioned:
> >
> > - ``CMNIST uses a small MLP and is linearly separable, which does not capture the behavior of modern deep neural networks''
> > - ``Using the default DomainBed architecture and hyperparameters for a single run''
> >
> > When making a conclusion (or conjecture) like" existing features learned via ERM may be “good enough” for out-of-distribution generalization in deep learning" in the paper, I still believe more care should be taken. In particular, this does not hold in all cases---my way of training an ERM model was questioned. I would like authors to clarify:  **under what conditions this statement is expected to hold?** e.g., does one have to work on image data or whatever datasets (like tabular or time series data?) do I need to use some augmentation techniques (like ERM in domainbed)? do I have to use some pretrained models for initialization purpose? Perhaps, more importantly, **how can one check if the ERM trained model would follow this statement in real settings?** In practice, we may only have some training domains, or even a big dataset consisting of data collected from different domains without domain indexes.

---

> > > ### Author Response · Authors · 2022-11-09
> > > **Response**
> > >
> > > Thanks for your continued engagement! **We are very glad to hear that we’ve adequately addressed all but this last remaining point.** With continued discussion we are confident we can resolve this last point, and we hope you will update your review accordingly.
> > >
> > > * **ERM vs. IRM**
> > >
> > > We did understand that you ran your experiment with ERM, not IRM. Our emphasis was on the fact that the codebase for the IRM paper is **flawed**. As a few more examples, *in addition* to the hyperparameter tuning, the IRM codebase (i) does not use a validation set, and (ii) does not evenly sample the MNIST dataset to produce the environments. That is, the ERM features are learned without validating on an in-domain holdout set, and the test environment ($p_e=0.9$) is much smaller than the other two domains—the third environment was constructed from the MNIST validation set whereas the first two were formed by splitting the training set in half. We recommend avoiding this codebase in general and using DomainBed instead.
> > >
> > > * **”my way of training an ERM model was questioned.” / “under what conditions this statement is expected to hold?”**
> > >
> > > Hopefully the previous paragraph clarifies that we are not questioning *all possible* implementations of ERM, but rather ones which are *known to be flawed*. There will of course always exist ways to train a model and construct a dataset such that our findings do not hold. For example, if someone trained a small 2-layer MLP on ImageNet and said “look, this network with ERM didn’t learn good enough features”, we hope you agree that this would not refute our experimental findings. Or, one could generate random labels for the OOD dataset, which no features could possibly be useful for. We are not suggesting that literally every implementation of ERM works for every OOD task. The point we are making is that **training a *modern architecture* with ERM and *currently accepted training/regularization practices* seems to be good enough for the benchmarks the community is currently focusing on. We have added this clarification to the paper.**
> > >
> > > To be more precise: deep learning research can be roughly split into work on improving methods “in-distribution” (ID) and “out-of-distribution” (OOD). ID research has made *enormous* progress and excels at a great variety of tasks. OOD research remains a very difficult problem, and the prior consensus was that “the methods we’ve developed for learning features ID don’t work at all for OOD, so we need to do something totally different.” Hence all the proposed objectives for training a network end-to-end for better OOD performance.
> > >
> > > In this work, we give evidence (for the first time) that **the methods we’ve *already developed* for ID feature learning could be good enough for OOD!** This exciting finding gives the research community a great deal of information on how to approach the OOD problem moving forward. Furthermore, it implies that *future improvements to ID feature learning could be leveraged for better OOD performance,* a fact which was not previously considered necessarily true.
> > >
> > > * **”how can one check if the ERM trained model would follow this statement in real settings?”**
> > >
> > > Unfortunately, **this kind of check simply cannot exist with our current (lack of) understanding of deep learning**. In the entire field of OOD generalization, there are roughly two ways to say anything at all about when something will work OOD: either we prove a formal result, in a precise mathematical model that will necessarily be somewhat simplistic (like our theorems), or we evaluate the method on some popular benchmarks with the *hope* that similar things will happen on similar benchmarks (like our experiments). There is (currently) no way to state with any reasonable precision a result of the form “method X will work for all image datasets that contain pictures with a camera of resolution at least Y”—nor is there any way to "check" what kind of datasets a network will be good for ahead of time.
> > >
> > > We want to emphasize that **such a statement would be an unreasonable ask for any deep learning method in general.** As far as we are aware, there are no such guarantees or checks for *any* deep learning applications. When a practitioner wants to know if a network will work on a task, the only options are to (a) analyze its behavior in a simplified model, or (b) test it on relevant data and see how it performs. **In this work, we do both!** For option (a), we present explicit, formal conditions when the features will be “good enough”, under a model that is quite general. For option (b), we present experiments on a variety of accepted benchmarks.
> > >
> > > We hope this discussion is clear. It is simply **not possible** to answer this question with our limited understanding of deep learning.
> > >
> > > We look forward to your response.

---

> ### Author Response · Authors · 2022-11-07
> **Author response**
>
> Thanks for your feedback! We appreciate that you took the time to try out our experiment yourself. **However, there are several issues with your experimental setup. We reran your experiment and confirmed that our conclusions *do hold* on Colored MNIST.**
>
> We’ll first address your other comments, and then in another comment discuss the experiment you ran.
>
> * We are not sure we understand your first concern. Did you intend for “(2)” to be a reference to prior work here? You wrote that the proposed method is “not particularly new”, **but you did not include any references, so we do not know what prior work you may be referring to.**
>
> Alternatively, it sounds like you could be saying that we could append a linear layer to the end of the feature embedder instead of “adjusting”. **This is not the case.** The problem with this idea is that no single linear layer can align *all environments simultaneously*. Prior work solves this by learning one function which is constrained to output aligned moments on all environments. But as we point out, this means that information must be discarded in order to satisfy the constraint. Instead, DARE first learns the unaligned features and then applies a *separate adjustment to each domain*. This approach **cannot** be implemented as an additional layer before the classifier. Example 1 perfectly demonstrates this fact: there is no single transformation that one could apply to $x$ that results in aligned distributions without throwing away the subspace $I-\Pi$.
>
> * **”the proposed method assume that the mean is a multiple of all-one vector”**
>
> This is not correct. **Enforcing a constraint is not the same as making an assumption on the test distribution**. We make *no assumptions* on the test environment mean except the stated bound $||b_e|| \leq \rho$.
>
> * **“I think it beneficial to mention the setting earlier, where the distribution shift may not be arbitrary, like in the anchor regression paper.”**
>
> In the first paragraph of the Section 3, we wrote: “While invariance is a powerful framework, there are some clear drawbacks, such as the need to throw away possibly informative features.” This was meant to convey the same idea, but we agree that it would be beneficial to explicitly mention “bounded distribution shifts” here. We have updated the draft. Thank you for the suggestion!
>
> Regarding the connection to Anchor Regression, **we do explicitly mention this connection at the end of Section 3** with reference to more discussion in the Appendix.
>
> * **“Since you already use Resnet as the feature extractor, i did not see why only compare method with linear predictor.”**
>
> Other methods that train a ResNet end-to-end *change the features that are learned*. Any objective defined over the entire network would not be directly comparable. The only methods directly comparable with frozen features are methods which can be run on *only* the last linear layer, such as GroupDRO. Regarding other end-to-end baselines, it is well established that **no existing end-to-end method beats ERM [1],** which means that comparing to ERM is effectively as good as comparing to all prior methods. Furthermore **we *do* compare to several other end-to-end baselines in the Appendix.** Are there other specific baselines you had in mind?
>
> * **”simply using the mean covariance matrix in train domains is less than justified.”**
>
> We agree that it would be ideal to have a better method for making this adjustment. Unfortunately, the entire difficulty of domain generalization is that we have *no knowledge* of the test domain; without additional information we cannot expect to be able to do better than an informed guess. Our averaging represents precisely this informed guess, and we show in Tables 2+3 in the Appendix that it is actually quite close to the truth. If our guess is very off, this implies that the test environment is *very* different from the train environment—at which point it is unrealistic to expect to be able to generalize at all. We’ve given a precise model and set of adversarial constraints under which our algorithm is optimal. Of course one can always come up with settings in which these assumptions don't hold, but in order to say *anything* in domain generalization we fundamentally require some bound on how the test distribution can change.

---

### Official Review · Reviewer_o4xo · 2022-10-30

**Confidence:** 3
**Clarity, Quality, Novelty And Reproducibility:** The paper is well-written and produce…
**Correctness:** 4
**Technical Novelty And Significance:** 4
**Empirical Novelty And Significance:** 4
**Recommendation:** 6

**Strength And Weaknesses:**

The experiments (from table 1) and the theoretical results (showing minimax optimality of DARE and the convergence as a function of environments) are both useful contributions. The linear correction has a simple implementation and there is further theoretical evidence that  the proposed idea also applies to other kinds of desirable invariances (proposition 3.1)


A few thoughts/questions to consider:
1. The random environments assumption in theorem 5.7 is peculiar. Can the authors justify why this would be true in practice?
2. It was a little weird to see that the object $\mathcal{R}_{e^\prime}$ hides the dependence of $\bar{\Sigma}$. The way I understand it is that $\bar{\Sigma}$ specifies a set of environments for which the DARE predictor does not too terribly. Stating early on that in general, environments can have "opposite" $A_e$ such that any one you can optimize for will be terrible on the other. Then identifying assumption 5.3 and showing it's a "large" set is useful could be a better way to present the ideas.
3. The sup in theorem 5.6 is not useful as it stands because it does no tell me whether DARE does better than prediction by chance. Can the authors comment on a quick way to bound 0-1 loss, for example, using $\mathcal{R}_{e^\prime}$?

**Summary Of The Paper:**

The paper makes points out that ERM-representation can contain sufficient information about the label that is linearly decodable to predict well. They follow this intuition to build an objective called DARE that is able to project away the non-invariant parts of the representations when predicting with a linear function. The key in the paper is to have a per-environment whitening of the representations and then making a mean restriction to then guarantee performance on every test distribution.

**Summary Of The Review:**

I like the paper but need clarifications about certain aspects of the theoretical work.

---

> ### Author Response · Authors · 2022-11-11
> **Author Response (1/2)**
>
> Thanks for your feedback! We are glad to hear you liked the paper and we are happy to provide clarification on the technical aspects. **We hope with your continued engagement we will be able to incorporate these clarifications and convince you to improve your recommendation.**
>
> * **”The random environments assumption in theorem 5.7 is peculiar. Can the authors justify why this would be true in practice?”**
>
> This is a great question! First, we want to emphasize that **the first of the two statements of Theorem 5.7 doesn’t require random environments**—DARE will recover the minimax-predictor so long we reach a “threshold” of at least $E$ linearly independent environmental means. This is exactly the same requirement as previous domain generalization (DG) methods such as IRM (actually, IRM and most other methods require $E+1$ domains). Such a result is really the best we can do in a “worst-case” setting, where an adversary chooses the test domain.
>
> But what if the test domain is not truly “worst-case”? When the domains are just a bit better behaved, the second part of Theorem 5.7 provides an *additional* guarantee which is better than all previous results. **This bound is just an added bonus:** if it doesn’t apply, our bound is the same as prior works; if it does, then DARE enjoys additional guarantees that other methods do not. This bound represents the first *finite-sample* environment complexity guarantee—the reason it doesn’t make sense to write as a deterministic bound is because our model, like other models for DG [1, 2], allows for *arbitrary* variation in the test domain (subject to the model structure). This means that **if we do not succeed in recovering an invariant predictor, the worst-case loss is unbounded.** And until we observe $E$ environments, the only predictor which we know will be invariant is a constant predictor. So you can see why a non-probabilistic model of environments does not allow us to say anything beyond the usual “threshold” result.
>
> Note that our proof technique does not require that the distribution be Gaussian, only that it is sufficiently “concentrated”. Of course, one could object to this on the grounds that we primarily care about generalization when the prior is *not* concentrated—this is precisely what the first, deterministic statement of Theorem 5.7 covers. The second statement is just saying that *if* the domains are a bit better behaved, DARE can do substantially better. And as we mentioned above, if the domains are *not* better behaved, the risk cannot be bounded anyways until the deterministic condition kicks in.
>
> Please let us know if any of the above is unclear!

---

> > ### Author Response · Authors · 2022-11-11
> > **Author Response (2/2)**
> >
> > * **”It was a little weird to see that the object $\mathcal{R}_{e’}$ hides the dependence of $\bar\Sigma$.”**
> >
> > Your understanding is correct in the sense that the “uncertainty set” for which our guarantees apply is determined once we commit to a $\bar\Sigma$. However, it is difficult to characterize when this set would be “large” or “small” as a general function of $\bar\Sigma$, or how to even quantify what those terms mean. Ultimately, the assumptions define test environments that are *very different from the training domains, but not **too** different in specific ways.* We tried to give some intuition for which types of variation specifically they allow for with remarks after the assumptions, e.g. Assumption 5.3 is strictly weaker than the “linear environment threshold”, and Assumption 5.5 is satisfied if the data has disjoint “varying” and “invariant” subspaces. **Please let us know if you have any specific suggestions for how this could be made clearer.**
> >
> > * **”The sup in theorem 5.6 is not useful as it stands because it does no tell me whether DARE does better than prediction by chance.”**
> >
> > Bounding the square loss is a common technique for analyzing logistic regression where directly evaluating the cross-entropy loss would be intractable. Note that the square loss is *classification calibrated* [3]. With an assumption on the entropy $H(y|x)$, a bound on the excess square loss of the log-odds can be used to bound square loss on the probabilities, which  directly translates to a bound on the excess 0-1 error. See Theorem 5 here: https://people.cs.umass.edu/~akshay/courses/cs690m/files/lec12.pdf
> >
> > As a very simple (loose) example in the binary case: a bound on the entropy for a binomial distribution gives a lower bound on the maximum probability over all outcomes. So suppose $\max_{c\in\{0,1\}} p(y=c|x) \geq \alpha,$ where $\alpha$ is somewhat close to 1 (indeed, for real-world data we often expect $\alpha \approx 1$ most of the time). Then
> > $$\log\frac{p(y=c|x)}{p(y \neq c|x)} = \beta^T x \geq \log \frac{\alpha}{1-\alpha}.$$ This means our prediction will be correct on a given $x$ if (but not only if!) our squared error is less than $\log^2 \frac{\alpha}{1-\alpha}$.
> > So, imagine that
> > $$R_{e’}(\hat\beta) \leq \frac{1}{C} \log^2 \frac{\alpha}{1-\alpha}$$
> > for some $C > 1,$ then Markov’s inequality tells us that the overall 0-1 error of our predictor is at most $\frac{1}{C}$. Note that when $\alpha$ is close to 1, $\log^2 \frac{\alpha}{1-\alpha}$ is large, so even when the bound in Theorem 5.6 is much greater than 1 this can still give a much better bound than random chance on the 0-1 loss, i.e.  $C \gg 2$.
> >
> > [1] Causal inference using invariant prediction: identification and confidence intervals. Peters et al. 2015
> >
> > [2] The Risks of Invariant Risk Minimization. Rosenfeld et al. 2020
> >
> > [3] Convexity, Classification, and Risk Bounds. Bartlett et al. 2003

---

> > > ### Comment · Reviewer_o4xo · 2022-11-18
> > > **Thanks for your response**
> > >
> > > This clarified a bunch of my concerns. I was also glad to see the responses to the other reviewers and the discussion there was also useful!
> > >
> > > I'm updating my score.

---

### Official Review · Reviewer_MFvo · 2022-10-31

**Confidence:** 3
**Correctness:** 3
**Technical Novelty And Significance:** 3
**Empirical Novelty And Significance:** 3
**Recommendation:** 6

**Clarity, Quality, Novelty And Reproducibility:**

Some parts on theoretical analysis can be improved. For example, Eq. (3) is in appendix, but seems important in the analysis. Overally, this paper contributes a few new insights and also proposes a new generalization method. The method can be easily implemented.

**Strength And Weaknesses:**

Strength:
1. Instead of learning invariant features or aligning the condition distribution of label given features, this paper analyzed the problem of OOD generalization from a new perspective by just learn a robust predictor, which is easy to implement and to be used in practice. The new design is promising and also contributes a few new insights such as the modular architecture.

2. It is good to see authors also build the connection between DARE with respect to IRM, which makes sense to me.

3. The new method also has strong theoretical guarantee.


Weakness:
1. My largest concern comes from the estimation of test covariance. Authors simply say that they just average the training domain adjusgments and empirically on a few datasets that the average works. However, I doubt about the rigorous of this strategy, especailly when the test data is out-of-control.

2. It seems that most of analysis focuses on Eq. (3) on a regression case, whereas the empirical study is on classification tasks. Is there a mismatch between theoretical analysis and empirical justification?

Questions for authors:
1. Can DARE be applied to scenario in which the training environments are coming sequentially, i.e., domain 1 comes first, followed by domain 2?

**Summary Of The Paper:**

This paper develops a new domain generalization method by learning a linear robust predictor on top of finetuned features extracted by a deep network (trained with ERM). Authors first performed a simulated study suggesting that the ERM produces features that are informative enough and the failure of deep networks to out-of-distribution (OOD) data is mainly due to the inadequate robust prediction. Based on this observations, authors developed a convex objective to learn a robust predictor that can be immediately applied to new test data. Authors theoretically analyzed the final solution and also established the connection between their method and anchor regression and IRM.


**Summary Of The Review:**

An interesting paper with clear motivation, new method and strong theoretical analysis. Some parts can be improved (see weakness points).

---

> ### Author Response · Authors · 2022-11-11
> **Author Response**
>
> Thanks for your feedback! We are very happy to hear that you appreciate our contribution and its ease of implementation.
>
> We want to begin our response by clarifying that DARE is **not** primarily focused on regression tasks—our use of the word “regression” in the name refers to “logistic regression” (i.e., classification). Hopefully this clarifies several of your points, namely: (i) the empirical study is on classification because we are primarily focused on classification, with the regression objective being included for completeness’s sake; (ii) Equation (3) is in the Appendix because our primary focus is on the classification objective in Equation (1). **Please note that all of our theorems apply simultaneously to regression and classification.**
>
> We want to add one more point in case the confusion stems from the fact that we study the square loss of the predictor: the logistic regression solution provided in Theorem 5.2 is the minimizer of the standard *cross-entropy loss*. We use square loss only for *analysis of the classifier’s error* in predicting the log-odds, which is a common technique for analyzing logistic regression where directly evaluating the cross-entropy loss would be intractable. One can derive a bound on the excess 0-1 risk from a bound on the excess square loss using the fact that the square loss is *classification calibrated* [1]. For example, see Theorem 5 here: https://people.cs.umass.edu/~akshay/courses/cs690m/files/lec12.pdf
> (see also our response to Reviewer o4xo for an explicit example of such a bound)
>
> * **To your point on the (lack of) rigor in “guessing” the test-time adjustment:**
>
> We agree that it would be ideal to have a better method for making this adjustment. Unfortunately, the major difficulty of domain generalization is that we have *no knowledge* of the test domain; without additional information we have to make some strong assumptions. We agree that expecting the average adjustment to work well is a strong assumption, but as we show in Tables 2+3 in the Appendix, it is actually quite close to the truth. We believe deriving better methods for choosing the adjustment is a promising future direction.
>
> In particular, the JIT-UDA setting offers a nice relaxation which may allow us to bridge the gap. This setting requires only unlabeled data at test-time, and can be performed truly “just-in-time” (i.e., on the fly as data is passed through the network). This might allow us to derive better estimates of the correct adjustment.
>
> * **”Can DARE be applied to scenario in which the training environments are coming sequentially, i.e., domain 1 comes first, followed by domain 2?**
>
> Absolutely! Each adjustment is domain-specific, so we only need access to the domain mean and covariance immediately before evaluating the loss (i.e., we don’t worry about adjusting the data until *after* it has been passed through the feature embedder). It would be sensible to optimize DARE in a “follow-the-leader” fashion, where each time we see a new domain we adjust the data appropriately and then find the new minimizer. In fact, this is effectively what JIT-UDA would be doing, but without updating the model in response to the new domain.
>
> **We hope the above addresses your concerns and convinces you to improve your score. Please let us know if anything else is unclear!**
>
> [1] Convexity, Classification, and Risk Bounds. Bartlett et al. 2003

---

### Official Review · Reviewer_4nvT · 2022-11-01

**Confidence:** 4
**Correctness:** 4
**Technical Novelty And Significance:** 3
**Empirical Novelty And Significance:** 3
**Recommendation:** 5

**Clarity, Quality, Novelty And Reproducibility:**

The claims about ERM in the paper are compelling and quite clear. The claims about DARE either need to be clarified or fixed.

The proposed method and the set of experiments seem to be novel.

There seems to be enough detail to reproduce the experiments in the appendix.

**Strength And Weaknesses:**

Strengths:
 + The first set of claims about the usefulness of ERM features is very compelling. Showing that retraining the last layer is sufficient for matching in-domain end-to-end performance is nice evidence that, among the problems represented in current domain generalization benchmarks, adaptation could be accomplished without re-learning representations.
 + The idea of applying per-environment transformations to features is quite interesting.
 + The proposed model of distribution shift---where both labels $y$ and inputs $x$ are driven by latent variables as a generalization of covariate- and label-shift assumptions---is interesting, and does fit a wider range of practical problems.

Weaknesses:
 - General statements in the first set of claims are stated as general truths and are not appropriately precise. For example, "We are not yet in a regime..." does not specify how "our" regime is defined. Is this specifically referring to DomainBed, and architectures trained on datasets of comparable size? Do these results say more about the particular set of problems in current benchmarks (i.e., the specific shifts that are represented), or about ERM in general? The phrasing of the statements seems to suggest the latter, but I'm not sure that is substantiated.
 - The framing of the DARE constraint as making the unknown centering of test data irrelevant seems strange to me. That framing makes it seem like if we could only whiten the test data, we wouldn't care about the DARE constraint at all. But in other parts of the paper, it seems that the DARE constraint actually plays a central role in the invariance claims.
 - In the same vein, the framing of the DARE constraint in terms of whitening seems to imply that if we could whiten the data in test domains (e.g., the JIT-UDA setting), we wouldn't need the DARE constraint at all. Is this true?
 - Despite making claims about the JIT-UDA setting, I don't see any empirical experiments about it. It would seem that, based on the claims in the paper, JIT-UDA should match oracle "cheating" performance.
 - In the theory, there is a lot of effort put into showing that the DARE solution for linear regression and logistic regression correspond to a post-hoc projection of the coefficients learned from the unconstrained learning problem (specifically, their projection onto the null space of the matrix of environment mean predictions). Why isn't DARE implemented this way? It seems like it would be straightforward, and close some gaps between the algorithm and the theory. At the very least, this would be a useful baseline since it is extremely simple.
- A synthetic experiment showing the difference between IRM and DARE in terms of dependence on invariant / variant information would be useful, especially when the means between environments vary.
 - There seem to be to be several technical issues in the discussion of DARE. Listing these below.
 - IMO it would be useful to present the model of distribution shift (including all related assumptions) first to motivate the method, and so that general claims can be interpreted in context. We know that no method will work for every kind of distribution shift, so specifying the assumptions up front would be helpful to understand exactly which problem is being solved.
 - The empirical performance of DARE seems rather underwhelming considering the extensive set of theoretical claims made about the method. I would have expected a step change in performance, rather than an incremental change over ERM.
 - It would be useful to have some discussion of the settings where the distribution shift model is likely to hold, and thus where we would expect DARE to perform well. For example, isolating particular tasks where it seems like the shift in mean prediction is the primary driver of poor performance would be useful, rather than averaging across a large set of tasks.

**Technical issues / questions**
 - Is Proposition 3.1 correct? How is it possible to make a statement about minimax risk across environments in terms of a minimizer of average risk across environments without putting constraints on the set of environments? I don't see a formal version of this proposition or the proof anywhere.
 - **Biggest issue for me:** The DARE constraint doesn't make sense to me from a whitening perspective. Even if the mean representation does not affect predictions, the centering still matters for estimating the covariance of $x$ for each environment $\hat \Sigma_e$. Specifically, without centering, the covariance $\hat \Sigma_e$ would include a rank-1 component corresponding to the mean of the data; i.e. $\hat \Sigma_e \approx E_e[X X^\top] = \Sigma_e + \mu \mu^\top$. This is a major issue, since the proofs of the main theorems take as a given that regression on $\hat \Sigma_e^{-1/2} x \approx A_e^{-1} x = \epsilon$. In cases where the DARE objective does something non-trivial, this is not the case.
 - What is the impact of the DARE constraint on the distribution predictions in each environment? It seems to imply that the predicted distribution of labels needs to be close to uniform? How does this constraint interact with variation in the label distribution between environments?
 - What is the interpretation of the assumption about shared right singular vectors? There is a remark about the assumption saying that *some* assumption is necessary, but it doesn't give a sense of *when* this assumption would be likely to hold and when it wouldn't. Alternatively, is this an empirically testable assumption? If so, proposing a direct test might be more compelling than claiming that the constraint helps empirically (since it depends on the very specific empirics).
 - Is there a proof that the given constraint on shared right singular vectors identifies $\beta^*$? It is not clear to me that one can recover $A_e$ from the observed covariance of $x$ under this assumption alone.
 - The assumption $E[\epsilon_0 \epsilon_0^\top]$ does not appear to be WLOG. In particular, this assumption seems to imply that one can recover $A_e$ from $\Sigma_e$, which is not trivial unless one assumes that the covariance of $\epsilon_0$ is known. In particular, this assumption seems to justify the whitening operation as opposed to some other transformation to a shared covariance.
 - It is not clear that the assumptions $V=I$ and $E[\epsilon_0 \epsilon_0^\top] = I$ are together WLOG. For example, this seems to imply that dimensions that are uncorrelated in $\epsilon_0$ must remain uncorrelated in the distribution of $x$?
 - The claim in the second paragraph of the proof of Theorem 5.2 claims that we immediately obtain $\beta^*$ from the unconstrained DARE objective. Then why do we incorporate an objective in DARE in the first place? Is this not the ideal parameter for a Bayes optimal predictor?
 - The paragraph after Theorem 5.2 seems to argue that nothing is projected out (i.e., $\Pi$ has full rank), but this seems to contradict (or is at least irrelevant to) the argument above that the dimensions containing the environment-wise means are being projected out, which would result in $E$ rank $d-E$ (unless there are collinear means). I guess this is because the environments all have the same mean in Example 1? This is a bit confusing, especially because it seems like the mean constraint is a central part of the DARE method.
 - Assumption 5.3 seems vacuous since the projection matrices $\hat \Pi$ and $I-\hat \Pi$ are by definition orthogonal.
 - The Lemma about logisitc regression is a known result, although there isn't a good citation for it that I could find. See this stackexchange: https://stats.stackexchange.com/questions/113766/omitted-variable-bias-in-logistic-regression-vs-omitted-variable-bias-in-ordina. It might be useful for streamlining the proof.

**Summary Of The Paper:**

The paper makes two arguments about domain generalization: first, there is an empirical observation that taking a neural network that was trained on training environments and retraining its last linear layer on data from test environments results in near-oracle performance on test environments. Thus, this is an argument that the features learned by ERM are sufficient, and all that is necessary for domain generalization is an adjustment to the final decision made on the basis of these features.

The second argument is in favor of a domain generalization method called DARE that specifies a new objective for a linear layer when one does not have access to test environments for fitting the last linear layer. The DARE objective specifies an average classification or regression loss with respect to features that have been decorrelated within environments (by multiplying by the inverse square root of each environment's feature covariance matrix), with an additional constraint that the average prediction within each environment be equal to a particular vector. Several claims are made about the minimax optimality of models learned with this objective under a particular model of distribution shift.

Empirically, it is shown that the DARE objective does as well or better than other invariance-encouraging objectives (and ERM) when implemented on top of ERM features.

**Summary Of The Review:**

I like the first set of claims about ERM, but the motivation and theoretical claims about DARE are unnecessarily confusing, and the empirical results for DARE are somewhat underwhelming.

===

After author discussion, I raised my rating to borderline accept, pending discussion with the AC and other reviewers about issues with clarity that I had with the paper. After discussion with the AC and reviewers, it became clear that many shared my concerns about clarity and the scoping of certain assertions, so I am lowering to borderline reject.

---

> ### Author Response · Authors · 2022-11-07
> **Author Response**
>
> Thanks for your detailed feedback! It’s clear that you’ve given a lot of thought to this review and we appreciate your efforts. **The technical concerns you’ve raised suggest to us that you might have misunderstood several key aspects of our theory and misread some of our notation**. We have worked very hard in this rebuttal to carefully detail what we think are the major points of confusion.
>
> **We hope you will continue to engage during the discussion period so that we can identify the narrow points of concern and hopefully convince you to improve your score.**
>
> We address your earlier comments in order, and then follow up with a separate comment on the technical points:
>
> * We feel that we have only claimed as much generality as is supported by our experiments and made clear that any additional applicability is conjecture. **We use very intentional language to convey this, including in the title of the paper (i.e. “may”).** By “regime” we refer to the architectures and methods that currently achieve SOTA on a popular DG benchmark. Our experiments are relevant insofar as the setting we study is precisely what the community currently cares about (e.g., if DARE got +20% accuracy on these benchmarks, we would not expect a large number of people to complain that we didn’t test on others). Incidentally, we note that work published after ours has made similar findings under related, but slightly different experimental setups, providing further support [1, 2].
>
> * You are correct that if we could identify the correct whitening for each environment, the invariance constraint would be unnecessary. But *we do not have access to data from the test distribution in domain generalization*. Instead, we rely on the assumption that the shared representation space would have “varying” and “invariant” subspaces, as in Example 1. Since we *cannot* know the correct whitening for new environments, we instead project out the varying subspace to avoid catastrophic error.
>
> * *If the data follows our model* then yes, JIT-UDA would not require any constraint at all. We make a similar remark in the paragraph after Theorem 5.2. An earlier draft of this work also discussed experimental results, but unfortunately we had to edit for length and JIT-UDA felt less essential to the general story of this work. In particular, we had written that **DARE does not outperform UDA methods on deep networks with unlabeled test data** (“all models are wrong”), but we feel this remains a promising direction which we are actively pursuing. We agree with you that some mention of the empirical performance of JIT-UDA is appropriate, so we have added this back in.
>
> * Along the same vein, and as a response to several of the concerns you’ve raised in your review: **our theoretical results are precise and formal claims under a simple data model, made with the goal of motivating and understanding the DARE objective** when learning from multiple heterogeneous data sources. Real world data of course does not perfectly follow this model, so it would be unrealistic to expect our method to improve performance by a large margin in all settings. **We emphasize that *no known method* produces a “step change in performance” over ERM [6] (or even any sort of robust improvement across benchmarks), so even an incremental change (especially using frozen features from a network trained with ERM) is a substantial contribution**.
>
> [1] Last layer re-training is sufficient for robustness to spurious correlations. Kirichenko et al. 2022
>
> [2] On Feature Learning in the Presence of Spurious Correlations. Izmailov et al. 2022
>
> [3] The Risks of Invariant Risk Minimization. Rosenfeld et al. 2020
>
> [4]  On calibration and out-of-domain generalization. Wald et al. 2021
>
> [5] Iterative feature matching: Toward provable domain generalization with logarithmic environments. Chen et al. 2021
>
> [6] In Search of Lost Domain Generalization. Gulrajani & Lopez-Paz, 2021.

---

> > ### Comment · Reviewer_4nvT · 2022-11-15
> > **Thanks for clarifications; some follow-ups**
> >
> > Thanks for these clarifications, and for your willingness to engage. Given your responses, I revisited the paper, and saw that there were several things I misread. I'll take most of the responsibility for this, although I think the presentation could be clearer. I agree that the idea of transforming inputs to have similar distributions and trying to project out differences in their means could work well in some contexts. It was just difficult for me to extract this simple insight from how the paper is currently written.
> >
> > I have a few additional questions about the DARE objective, now that I've spent more time with the paper. Sorry to bring these up later in the review process.
> >
> >  - Is the formulation of the DARE objective for classification in (1) actually convex as claimed? As I now understand it, the constraint is fixing a part of the softmax objective that is fundamentally ambiguous, but I am not sure that the constraints as stated actually resolve all of the ambiguity. For example, when $k=2$ as in the logistic regression case, $\beta$ is a $d \cross 2$ matrix, even though the problem is fully characterized by a $d \cross 1$ vector, so there seems to be $d - E$ degrees of freedom left in $\beta$ even after applying the constraint. In the standard logistic regression formulation, this is resolved by setting the first column $\beta_0=\boldmath 0$, and my understanding is that this is also the case for multinomial logistic regression, but this is not compatible with the stated objective.
> >  - Relatedly, how is the DARE objective translated to the logistic regression setting in (2) and the theorems? In those problems, $\beta$ is a vector not a matrix, so I am not sure how the constraint is expressed there, or what the "solution to the DARE objective" means.
> >  -  In the experiments where the constraint was omitted, was there some other heuristic used to resolve this ambiguity (having $\beta$ be a $d \cross 2$ matrix) or was the optimizer trusted to choose a solution from this equivalence class? In practice, I have seen some of these ambiguities lead to models failing to train, which could explain the poor performance of classifiers at the $\labmda=0$ setting in Figure 4, while any nonzero $\lambda$ seems to lead to better stability. Were the ERM baselines also implemented in this way?
> >  - In the technical proofs, bouncing between the regression problem and the logistic regression problem is very confusing, since the objectives are written differently (at least I assume so, although I'm still not sure how this is written for standard logistic regression).
> >
> > Also, I wanted to reiterate this question / comment:
> >
> >  - In the theory, there is a lot of effort put into showing that the DARE solution for linear regression and logistic regression correspond to a post-hoc projection of the coefficients learned from the unconstrained learning problem (specifically, their projection onto the null space of the matrix of environment mean predictions). Why isn't DARE implemented this way? It seems like it would be straightforward, and close some gaps between the algorithm and the theory. At the very least, this would be a useful baseline since it is extremely simple.
> >
> > Finally, re: the question of a step change in performance, I should clarify: I agree that we can't expect any single perform fantastically across all OOD benchmarks, since these benchmarks are often measuring fundamentally different problems. However, to motivate the particular approach being used here, it would be useful to understand when we expect it to work and when we expect it to fail. Selecting particular domain generalization tasks where you think the assumptions are particularly close to the true story, and showing that the method works well there, would be compelling. Likewise, potentially selecting some tasks where the assumptions are far from being met, and showing that the method is worse there, would also be useful. At the very least, describing how the assumptions seems like they're failing in the real examples would be useful.

---

> > > ### Author Response · Authors · 2022-11-16
> > > **Addressing the follow-ups (1/2)**
> > >
> > > Thanks for your updated remarks and for continuing to engage with us! We are happy to clarify the writing and we are certain this discussion will lead to an improved manuscript. **We hope that our previous response plus addressing these remaining points will convince you that the contributions presented in this work (both experiments and theoretical analysis) are of interest to the community and are worth presenting at the conference.**
> > >
> > > * **”In particular, your explanation of the WLOG assumptions in this forum was useful but not obvious, because it was not clear to me what the goal was.”**
> > >
> > > We are very glad to hear that this explanation was helpful! We have added a sentence to this effect in the main body, with a reference to a longer discussion in the appendix. We appreciate your help in identifying this point of confusion.
> > >
> > > * **”IIUC, the claim here is that under the proposed model, if you were to transform all of the observed environment X's so that they all had the same centered covariance (even if that covariance were not $I$), you would end up with the same results. Is this correct?”**
> > >
> > > Exactly! We choose the identity as the “canonical” covariance for convenience. We could choose any fixed covariance and the same arguments would apply. The key is that the domains start out with different covariances, and so we require different adjustments to align them.
> > >
> > > * **”I am still confused about what the implications of the "right singular vectors preserved" assumption are”**
> > >
> > > Intuitively, you could think of this assumption as saying that the *way* in which the covariance changes from $\epsilon$ to $x$ is identifiable, up to a transformation which won’t affect the recovered predictive distribution. We only require that the *unidentifiable component* of the transformation (i.e., the component that cannot be “adjusted for”) is fixed—otherwise we cannot give any guarantees, as demonstrated by the simple example we gave previously.
> > >
> > > Geometrically, the left and right singular vectors of $A_e$ identify a basis for the latent space ($\epsilon$) and a basis for the observed space ($x$) such that $A_e$ is diagonal in these bases. Our assumption is that the bases for the latent space do not change between domains.
> > >
> > > **We hope this helps with your intuition and we welcome your feedback/suggestions on how to further clarify.**
> > >
> > > * **”Also, it seems like this assumption is only used in Theorem 5.2 in the discussion of the closed form of the logistic regression solution”**
> > >
> > > It looks this way because of our WLOG assumption that $V=I$. As with logistic regression, the “true” solution would be $V^T \beta^*$. So, the domain-adjusted classifiers would be different for different domains if $V$ is not fixed.
> > >
> > > * **”Is the formulation of the DARE objective for classification in (1) actually convex as claimed?”**
> > >
> > > The objective is standard logistic regression which is certainly convex. Likewise, the feasible set is convex. We omit the bias vector $\beta_0$ for brevity, but it is included in both the objective and the constraint and does not change the convexity of either. Perhaps the omission of $\beta_0$ is the source of confusion? We’ve added a sentence about this to the draft, does this help clarify the objective? (Also see the next point)
> > >
> > > * **”how is the DARE objective translated to the logistic regression setting in (2) and the theorems?”**
> > >
> > > The DARE objective is defined the same for binary or multinomial logistic regression, with $\beta\in\mathbb{R}^{d\times k}, \beta_0\in\mathbb{R}^k$. For binary classification, we allow $\beta$ to be a vector—the constraint would then be $\beta^Tx = 0$ (i.e., $\sigma(\beta^T x) = \frac{1}{2}$), because this would induce a uniform distribution over the two classes. Note that **this is exactly the constraint for linear regression,** and this point should hopefully clarify the connection between the two. In both cases the idea is simply that *the representation mean should have no effect on the prediction.* We have added a sentence on this to the draft.
> > >
> > > * **”was there some other heuristic used to resolve this ambiguity”**
> > >
> > > As is standard, we added a small penalty on the squared $\ell_2$-norm of $\hat\beta$ to ensure a unique solution.
> > >
> > > * **”bouncing between the regression problem and the logistic regression problem is very confusing”**
> > >
> > > We note that our proofs only deal with the two cases separately in *one* instance, for the separate proofs of Theorem 5.2. All other theoretical arguments apply *simultaneously* to logistic and linear regression.

---

> > > > ### Author Response · Authors · 2022-11-16
> > > > **Addressing the follow-ups (2/2)**
> > > >
> > > > * **”Why isn't DARE implemented this way?”**
> > > >
> > > > We tried this method and the results are mostly similar, sometimes a bit worse (1-2%). Note that this approach suffers two distinct sources of finite-sample error: estimating $\beta^*$ and estimating $\Pi$. We present the objective as in the paper to expose more clearly the motivation for the constraint.
> > > >
> > > > * **”it would be useful to understand when we expect it to work and when we expect it to fail.”**
> > > >
> > > > We agree that it would be helpful to “qualitatively” characterize what kinds of settings we could expect DARE to work well. Unfortunately, it’s rarely possible to “translate” between qualitative properties of real-world data and a precise mathematical model which allows us to give formal guarantees.
> > > >
> > > > We could give a high level intuition of the type of datasets that would be favorable—for example, since we adjust the second moment but induce invariance to the first moment, DARE can be expected to perform well when, in the ERM feature space, the test domain is “locally shaped like the training domains”, but with a bounded change in the location of the cluster centers. However, **we advocate healthy skepticism when interpreting these kinds of “stories”,** since it’s difficult to map human-ascribed semantics to mathematical models of real-world data with any reasonable accuracy.

---

> > > > > ### Comment · Reviewer_4nvT · 2022-11-18
> > > > > **When things would fail**
> > > > >
> > > > > I don't think it's necessary to tell a qualitative story here, but rather to discuss classes of problems where it would be obvious that key assumptions are violated, so that practitioners would have reasonable expectations.
> > > > >
> > > > > E.g., in cases where there's strong label shift bw classes, given that this method constraints the mean prediction, would we expect this to eliminate a lot of important signal? In cases where the SVD assumption is violated, could the method lead you to a bad representation?

---

> > > > > > ### Author Response · Authors · 2022-11-18
> > > > > > **Not sure we understand what sort of statement you have in mind**
> > > > > >
> > > > > > Thank you for your continued engagement! We’re still not quite sure of the flavor of the statement you are interested in. Our theoretical statements apply to a precise data model, and that is the most “quantitative” result one could hope for—any statements about how much a real benchmark deviates from such a model would necessarily be “qualitative”.
> > > > > >
> > > > > > Our prior response gave an example of such a qualitative characterization: in datasets where the ERM features cluster in such a way that the shape of the clusters (i.e. the covariance) is roughly preserved, by the means of the clusters can shift (arbitrarily)—the algorithm could be reasonably expected to work.
> > > > > >
> > > > > > You asked specifically about “strong label shift”: *our model allows for arbitrary label shift between domains* due to the different distributions $p^e$ over $\epsilon$. This is since in our model, the labels are distributed as $y = \sigma((\beta^*)^T \epsilon)$, and $\epsilon$’s distribution can vary arbitrarily across domains (see Equation (2)). Moreover, on some of the benchmarks we evaluate on, the datasets *do* exhibit massive label shift, and DARE matches or exceeds SOTA.
> > > > > >
> > > > > > You asked what happens if the SVD assumption is violated. In the most general sense—i.e. if it’s violated to an *arbitrary degree*—there is fundamentally *nothing* that can be said about generalizing to new distributions. This should hopefully be clear from the example we gave above, which we’ve added to the paper. If you mean what would happen if it’s “somewhat violated”: Assumption 5.1 could be relaxed by assuming that all $V_e$ lie in a ball of some small radius, which would imply approximate alignment of the bases, which in turn would add an additional error term as a function of the radius of this ball.
> > > > > >
> > > > > > You’ve also asked for settings where it may be “obvious that key assumptions are violated, so that practitioners would have reasonable expectations.” As we’ve discussed, giving a qualitative condition which a practitioner may be aware of a priori (such as “my dataset exhibits severe label shift”), will rarely yield useful quantitative guarantees. However, we can try to give examples of specific checks that a practitioner could perform on their dataset which may give some indication of the likelihood of DARE being appropriate for their use case. While the experiments presented in Tables 2+3 in the appendix give evidence that the conditions (approximately) hold on the benchmarks we consider, **these experiments could also serve as checks of whether or not DARE is appropriate for a future task.** If one were to repeat these experiments with a new dataset and see opposite results, they could reasonably infer that the modeling assumptions may be strongly violated and that DARE might not be the correct choice for that setting. To be more explicit: two conditions when DARE might not be appropriate could be described as (i) when adjusting the domains does not result in the individual domain-optimal classifiers being more closely aligned with one another, and (ii) when the average covariance of some of the domains is very different from the covariance of another domain. Are these the sort of examples you had in mind?

---

> ### Author Response · Authors · 2022-11-07
> **Addressing your technical points (1/3)**
>
> * You are right, Proposition 3.1 indeed contains a typo. In the second condition, we meant to quantify $f’$ instead of $S’$. That is, it should read that for all $f’$ with lower risk than $f^*$ on any environment, $$\exists e, e’.\ S(f’, p_e) \neq S(f’, p_{e’}).$$
> Thanks for pointing this out, we have fixed it in the updated version!
>
> * We are not sure we understand what you mean by “without centering, the covariance would include a rank-1 component corresponding to the mean of the data”. Perhaps there is a notational confusion: $\Sigma_e$ **is** the centered covariance (and $\hat \Sigma_e$ is an estimate thereof). Your comment suggests that you think we are adjusting with the uncentered sample covariance, which is not the case. In the setting without access to unlabeled data, we adjust with a best guess which is the average of the training adjustments, which all use the *centered* sample covariance. We show in Table 2 in the Appendix that this guess is quite close to the true centered sample covariance.
>
> **Since you’ve indicated that this is your biggest concern, we are eager to make sure we have addressed it. If you think we’ve misunderstood what you meant here, please let us know.**
>
> * **“It seems to imply that the predicted distribution of labels needs to be close to uniform”**
>
> This is not correct. Note that the *softmax of the average of the logits* is not the same as the *average of the softmaxes of the logits*. The DARE constraint does not enforce any requirement on the distribution of label predictions.
>
> * **”What is the interpretation of the assumption about shared right singular vectors?”**
>
> This is a non-degeneracy assumption which is needed for (partial) identifiability. Such an assumption is not testable on real data (since real data will not even exactly follow our model), but it is fundamentally necessary when working with latent variables in the theory of OOD generalization. To see this, consider training on data from a linear regression model with latent variables $z\sim\mathcal{N}(0, I)$ where we observe $(x, y) = (Az,  \beta^T z)$. Now suppose we encounter a test distribution with $(x, y) = (A’z, \beta^T z)$, where $A$ has different right singular vectors from $A’$. the resulting prediction task would be *completely detached from the training data* and we could not possibly hope to generalize. In fact, we would argue it is surprising that our method *only* requires $V$ to be constant, and not the entire matrix $A$.
>
> On that note, we *do* discuss simple settings when this assumption could hold—this assumption includes (but is more general than!) additive interventions on a causal linear DAG, which is precisely the setup used for Anchor Regression [1]. To be clear: we make a *strictly weaker* assumption—**the corresponding assumption made by [1] would be that $A_e$ is constant for all environments**. We hope you agree that in comparison, our assumption is quite mild. We discuss this further in the next point:

---

> > ### Author Response · Authors · 2022-11-07
> > **Technical points (2/3)**
> >
> > * **”Is there a proof that the given constraint on shared right singular vectors identifies** $\beta^*$? **It is not clear to me that one can recover $A_e$ from the observed covariance of $x$ under this assumption alone.”**
> >
> > This comment, as well as the later ones on the assumptions we make without loss of generality, suggest that you think we are trying to recover $A_e, \beta^*$ directly, but **this is not the task we are solving**. You are correct that *if* we needed exactly these parameters, it would not be valid to make these WLOG assumptions jointly. But it is crucial to observe here that $\beta^*$ **and** $A_e$ **do not need to be directly identifiable**, because we only care about the predictive distribution $\beta^*$$^T\epsilon$. We only need $A_e$ to be identifiable from $x$ up to equivalence of this distribution. So if for example we recover some transformation $\hat A_e = A_e M$, this is not at all a problem because we also learn the corresponding $\hat\beta = M^T \beta^*$ such that $\hat\beta^T \hat A_e^{-1} x = \beta^*$$^TM M^{-1} A_e^{-1} x = \beta^*$$^T \epsilon$. This is why it is ok to make these WLOG assumptions jointly.
> >
> > More directly: suppose instead $V \neq I$ and $\mathbb{E}[\epsilon_0 \epsilon_0^T] = \Sigma_0$. Then we can simply reparameterize as $\epsilon_0 \to \Sigma_0^{-1/2} \epsilon_0$, $b_e \to \Sigma_0^{-1/2} b_e$, $\beta^* \to \Sigma_0^{1/2} \beta^*$, $A_e \to A_e V^{-1} \Sigma_0^{1/2} $. It is easy to see this results in the same observed distribution over $(x,y)$, and further that learning $\beta^*$ to predict on $\epsilon_0$ is the same as learning $\Sigma_0^{1/2} \beta^*$ to predict on $\Sigma_0^{-1/2} \epsilon_0$. So now we’ve reduced this to a setting where $\mathbb{E}[\epsilon_0 \epsilon_0^T] = I$ but perhaps $V \neq I$.
> >
> > This does *not* imply that “dimensions that are uncorrelated in $\epsilon$ must remain uncorrelated in the distribution of $x$”. **Assuming $V=I$ without loss of generality is not the same as assuming $V=I$.** The former allows us to make arguments *as if* $V=I$ which generalize to other settings where $V\neq I$ but is still shared across environments. Specifically, when $V \neq I$, it represents precisely the unidentifiable transformation $M$ in the above paragraph, which as we pointed out does not pose a problem for prediction because it will not change in future environments.
> >
> > Please let us know if any of this is still unclear.
> >
> > * **Then why do we incorporate an objective in DARE in the first place? Is this not the ideal parameter for a Bayes optimal predictor?**
> >
> > When we speak of the “Bayes optimal predictor $\beta^*$”, we are referring to the regression vector which *would* be optimal for any single environment **if we were able to invert the environment-specific transformation $A_e$**. That is, for any *fixed* environment $e$, we would like to adjust the data via $x’ = A_e^{-1} x$, and then predict with $\beta^*$$^Tx’$. The problem here is that this does not work *simultaneously for all allowable test environments*, because each one requires a *different, unknown adjustment*. Since we don’t see any data from the test distribution, we have *no knowledge* of what $A_e$ could be in the future. Thus, the goal of domain generalization is **not** to recover the Bayes optimal predictor, but to recover the *minimax-optimal* predictor, which explicitly ignores information which varies across environments. This is the reason for projecting out the varying subspace $I-\Pi$.
> >
> > * **The paragraph after Theorem 5.2 seems to argue that nothing is projected out…but this seems to contradict…the argument above that the dimensions containing the environment-wise means are being projected out.**
> >
> > There is a distinction here between the varying subspaces *before* and *after* the domain-specific adjustment. The point we are making is that *before* the adjustment, it could be the case (as in Example 1) that a subspace of the data *appears* to be varying across environments, and therefore an invariance-based solution such as IRM would project it out; but *after* adjustment, the subspace that previously appeared to be varying is now actually invariant, and therefore there is no need to remove it (hence $\hat\Pi$ is full rank). You are correct about this following from the mean being the same in all environments, but note that if we did not perform the adjustment, then even if $b_e$ were fixed, the observed mean $\mathbb{E}[x] = A_e b_e$ would vary (assuming $b_e\neq 0$), and so $\hat \Pi$ would be lower rank. The benefit of the adjustment is that now we can make use of the additional information which previously would have been removed, and **this is precisely the advantage of DARE over existing invariance based methods.** Note that the projection we perform is still necessary for any subspace that continues to vary *even after adjustment* (i.e., when the $b_e$ *do* vary). As we wrote in the referenced paragraph: “DARE therefore only removes what *cannot be aligned.*”

---

> > > ### Author Response · Authors · 2022-11-07
> > > **Technical points (3/3)**
> > >
> > >
> > > * **”Assumption 5.3 seems vacuous since the projection matrices $\hat\Pi$ and $I-\hat\Pi$ are by definition orthogonal.**
> > >
> > > This is not correct. Perhaps you missed the presence of $\Delta$ between $\hat\Pi$ and $I-\hat\Pi$ on the left-hand side?
> > >
> > > * **”The Lemma about logistic regression is a known result”**
> > >
> > > The link you’ve provided does not support this claim. Rather it shows that the result holds under a *probit* model (not logit), which says that $p(y|x) = \Phi(\beta^Tx)$, where $\Phi$ is the Gaussian CDF. We are aware of this result, and we discuss it and give a reference in the appendix [2]. Your link also discusses the idea that under the *logit* model, the solution is “biased towards 0”. This is *not* the same as what we prove. We prove that not only is the solution biased towards 0, but that it is an *exact scaling* of the true vector $\beta^*$. This fact is not stated in the link you’ve shared, nor in the paper that is referenced in the discussion on that page. In fact, this result is *not true* for general symmetric densities, and so it could not possibly be the claim being made in the linked post (at least, not without being wrong). We conducted an extensive search on this topic and were unable to find any existing work which proved the same result, which is why we set out to prove it ourselves.
> > >
> > > We hope our responses clarified the points of confusion and placed better in context the novelty and importance of our contributions. **We hope you will continue to engage and let us know if anything still remains unclear.**
> > >
> > > [1] Anchor regression: heterogeneous data meets causality. Rothenhäusler et al. 2018
> > >
> > > [2] Marginalized multilevel models and likelihood inference. Heagerty and Zeger, 2000

---

> > ### Comment · Reviewer_4nvT · 2022-11-15
> > **Thanks! Technical Points Follow-Up**
> >
> > Thanks for your thorough responses to my questions / comments, especially those assertions that I made that were wrong. I do think there could be greater clarity on these issues; when I revisited the paper, I had to work pretty hard to see your points from the text.
> >
> > I'll withdraw the following concerns:
> > - Non-centered covariance
> > - Label distribution
> > - The need for DARE if we can get a Bayes Optimal predictor in the training environments.
> > - Assumption 5.3 being vacuous. You're right that I missed $\Delta$.
> > - Logistic regression lemma
> >
> > My remaining concerns are still about the clarity in terms of articulating what the goal is in the theoretical arguments. The arguments are all framed in terms of (projections of) optimal parameter vectors $\beta^*$ but, as you point out to my questions, identifying $\beta^*$ is not actually the goal. I understand that the WLOG assumptions are being made here so that the problems can be formulated in terms of $\beta^*$ even though the goal is more general. However, I think that this needs to be made clearer and explicit in the text. In particular, your explanation of the WLOG assumptions in this forum was useful but not obvious, because it was not clear to me what the goal was.
> >
> > IIUC, the claim here is that under the proposed model, if you were to transform all of the observed environment X's so that they all had the same centered covariance (even if that covariance were not $I$), you would end up with the same results. Is this correct?

---

> > > ### Comment · Reviewer_4nvT · 2022-11-15
> > > **Additional Questions Re: Assumptions**
> > >
> > > I am still confused about what the implications of the "right singular vectors preserved" assumption are, because it would be useful to understand how this translates to substantive assumptions about the problem. I had suggested that this implies something like the correlation structure in $\epsilon$ being preserved in all downstream $X$'s, since they all undergo the same $V$ rotation before scaling by $D$ and rotation by $U$. Perhaps this is wrong, but it would be good to know what this assumption means, e.g., geometrically.

---

> ### Author Response · Authors · 2022-11-18
> **Gentle reminder**
>
> We want to sincerely thank you again for your involvement and thoughtful feedback! We hope
> that our most recent response addressed your additional questions.
>
> We would like to gently remind you that today is the end of discussion with author involvement. In light of your new understanding of DARE (and our updated manuscript to help others reach the same understanding), as well as your belief that our experiments regarding ERM are "compelling and quite clear", we hope you will reconsider your initial assessment. If you have any more thoughts, we are happy to continue to discuss until the deadline.

---

### Official Review · Reviewer_NbM4 · 2022-11-04

**Confidence:** 3
**Correctness:** 3
**Technical Novelty And Significance:** 2
**Empirical Novelty And Significance:** 3
**Recommendation:** 6

**Clarity, Quality, Novelty And Reproducibility:**

**Clarity:** Good.

**Quality:** Good.

**Novelty/originality:**
- Insight of ERM features being "good enough" is mostly novel (some previous works also found ERM features to perform well "in passing", e.g. [2]).
- The novelty of the normalization method DARE is somewhat unclear to me, since very related domain-specific normalization methods are not discussed in sufficient detail (see weakness above).


[2] Zhang, J., Lopez-Paz, D., \& Bottou, L. (2022). Rich Feature Construction for the Optimization-Generalization Dilemma. In _International Conference on Machine Learning_.

**Strength And Weaknesses:**

**Strengths:**
- _Interesting insights:_ ERM features being "good" enough.
- _Clear exposition:_ Lucid writing, precise notation, great Figure 1.
- _Solid theory:_ While many similar domain-adjustment-based methods exist, few provide such theoretical justification.

**Weaknesses:**
- _The domain specificity is questionable for the test domain:_ One of the major purported benefits of DARE over related alignment-based methods like CORAL is its domain-specificity. However, as the correct whitening matrix is not known for the test domain, the average training-domain adjustment is used as a "best guest" for the test domain. This means that the adjustment is not dependent on the test domain encountered, calling into question the purported benefits of domain specificity.
- _Difference with prior domain-specific normalization-methods is unclear_: The related work briefly discusses prior methods which normalize the features using a domain-specific mean and covariance. However, these works are not discussed/compared in detail, but rather written-off for being "ad-hoc".
  - A better discussion/comparison would improve the paper, putting the contribution in context.
  - Ideally, this would also include an empirical comparison, going beyond just ERM, IRM and GroupDRO to use more closely-related normalization methods.
- _Just-in-time UDA usually called test-time UDA_: The authors come up with a new name (Just-in-time UDA) for an existing task (test-time UDA [e.g. 1]). I think it would be best to keep the previous name to avoid confusion/overloading.



[1] Wang, D., Shelhamer, E., Liu, S., Olshausen, B., \& Darrell, T. (2020). Tent: Fully Test-Time Adaptation by Entropy Minimization. In _International Conference on Learning Representations_.

**Summary Of The Paper:**

This paper suggests that the features learned by ERM are "good enough" for out-of-distribution (OOD) generalization---they just need to be used in the "right way". This in turns suggest a shift in focus for OOD generalization, from feature learning to _robust regression_. Towards this end, a new objective called DARE is introduced which simply performs a domain-specific normalization of the feature-space mean and covariance. Theoretically, it is shown that DARE recovers the minimax-optimal predictor under a constrained set of test distributions. Empirically, DARE is shown to perform well on some DomainBed datasets.

**Summary Of The Review:**

Overall this paper presents interesting empirical insights followed by a method for exploiting these insights. While there are a few points that could be improved (see weaknesses above), I believe the value of this paper to the community is clear and thus recommended acceptance.

---

*Update*:

Following discussion with the other reviewers and a reading of their reviews, I still believe that the empirical insights about freezing ERM features is useful, but my concerns about the proposed method have grown. In particular, it is unclear: 1) when it is supposed to work and when not (e.g. on what types of shifts it succeeds/fails); 2) how exactly it relates and compares to existing whitening techniques (e.g. vs. test-time UDA methods which adjust/whiten based on a batch of test data; and vs. CORAL: unclear if it really helps to freeze the features as no comparison is made). As a result, I lower my recommendation to a weak/borderline accept.

---

> ### Author Response · Authors · 2022-11-08
> **Author response**
>
> Thank you very much for your positive review! We worked very hard to convey these ideas clearly, so we are glad that you consider the clarity and quality to be high. We appreciate your feedback and below we do our best to address your concerns. *We hope this response will increase your confidence in your rating.*
>
> First, we wanted to note that **our work actually predates the (Zhang et al.) reference you’ve provided.** (Unfortunately, providing arxiv links to prove this would violate anonymity.) Hopefully this further establishes the novelty of our experimental observations. Now to address your specific points:
>
> * We agree that the need to “guess” the test-time adjustment is not ideal. The difficulty of generalizing when we have truly *no information* about the test domain means that there is rarely anything better we can do without application-specific knowledge. We feel that DARE presents a promising approach because of its ability to naturally incorporate unlabeled data at test time, and to do so “just-in-time” (we address our use of this new terminology below). We also think it is encouraging that our best guess appears to be quite good! (see Tables 2 + 3) There is room for improvement here, and it is something we are actively working on.
>
> * Thanks for pointing this out—we have added more discussion on these methods in the updated draft, clarifying the following: though we point out that prior work is “ad-hoc”, this is not the main difference from DARE. The main difference is that these methods match statistics *while training the network end-to-end*. This makes optimization very finicky, in addition to the difficulty in understanding what this approach is actually doing and when we can expect it to work (hence the advantage of using frozen features). Further, such methods have consistently been shown to perform **no better than ERM when evaluated properly [1] (i.e., when hyperparameters are not chosen based on downstream performance)**, and so a comparison to ERM (and a few other end-to-end methods included in the appendix) is all that is needed to get a sense of how DARE stacks up against prior work.
>
> * We are familiar with the terms “test-time training” and “test-time adaptation” (and the reference you’ve provided), but we believe there is an important distinction here which warrants a new term. TENT and similar methods are applicable at test-time, but they are not truly “just-in-time”: given unlabeled test data, TENT needs to update (some of) the network parameters, which incurs the additional cost of several forward and backward passes. This computation scales with the size of the network and could not be done literally on the fly as new data is encountered. In contrast, DARE requires only estimating the mean and covariance of the network outputs, whose computation is trivial and scales only with the batch size. We devised the term JIT-UDA to capture this distinction, as we are unaware of prior work which can adjust so immediately (please let us know if we missed something!) We have clarified this point in the updated draft.
>
> [1] In Search of Lost Domain Generalization. Gulrajani & Lopez-Paz, 2021.

---

> > ### Comment · Reviewer_NbM4 · 2022-11-18
> > **Response to authors**
> >
> > - I agree that choosing the test time adjustment is difficult without knowledge about the test domain -- this is where interpretable hyperparameters become critical to the domain generalization problem. In this light, could there be other choices than the average training-domain adjustment, ideally with these choices having some interpretation (e.g. choosing the adjustment of the training domain on which the predictor performs worst?). In terms of justifying the current best guess, I'm not convinced by the argument that DARE is promising because it uses unlabeled test-domain data (so too do many other papers, e.g. those for source-free domain adaptation, particularly the optimization free ones like [1]).
> > - I think you'd need to justify your claims of optimization being finicky with some empirical comparisons, particularly if this is the major difference compared to your method. I don't find the (false) claim that they all perform worse than ERM on domainBed convincing/sufficient here: (1) CORAL actually outperforms ERM on DomainBed; and (2) other datasets exist and would serve the purpose of exhibiting the advantages of your less "finicky" method, if that is indeed a major problem.
> > - I take the point about test-time training Vs JIT updating. On this note, I think [1] is even more related than initially thought, since they too do a optimization-free classifier-adjustment for DG.
> >
> > [1] Iwasawa, Y., \& Matsuo, Y. (2021). Test-time classifier adjustment module for model-agnostic domain generalization. Advances in Neural Information Processing Systems, 34, 2427-2440.

---

### Comment · Reviewer_4nvT · 2022-11-18
**Objective convexity, connection to theory, and the claim that the mean has "no effect"**

I don't think the objective as written is convex. When I mentioned $\beta_0$ in the context of logistic regression, I meant the entire first column of the $\beta$ matrix (not a bias coefficient), since $\beta$ as presented in the objective (1) is parameterized by a $d \cross 2$ matrix, as opposed to a $d$-vector in the theory section (the $d$-vector version is convex). The objective in (1) is defined as constraining the 2-vector $\beta^\top \Sigma-1/2 \mu$ to be equal to (1/2, 1/2), which would have no effect on predictions (as explained in the text, bc softmax is invariant to this constraint). But in the theory section, according to the explanation here, the constraint enforces the mean logit to be zero, which does have a clear effect on predictions. Based on the explanation here, it seems that this happens bc in the beta-as-matrix representation, the theory effectively sets the first column of the $d\cross 2$ matrix $\beta$ to be zero, and softmax is not invariant to setting $\beta^\top \Sigma-1/2 \mu = (0, 1/2)$.

I'm confused by this inconsistency, specifically about:

(1) What it means about the applicability of the theory to the objective as stated. I am not sure whether the claims made apply to the case where the objective is made convex by an L2 penalty rather than setting one column of beta to a baseline value.

(2) What is the precise claim about how predictions are / are not affected by the DARE constraint.

---

> ### Author Response · Authors · 2022-11-18
> **The objective is convex**
>
> * **"The objective in (1) is defined as constraining the 2-vector $\beta^\top\Sigma^{-1/2}\mu$ to be equal to (1/2, 1/2)"**
>
> This is *not correct*. The DARE constraint in (1) is:  $$\textrm{softmax}(\beta^\top\Sigma^{-1/2}\mu) = \frac{1}{k}\mathbf{1}$$ You've dropped the crucial application of softmax in your above comment. Observe that
> $$\textrm{softmax}(\beta^\top\Sigma^{-1/2}\mu) = \frac{1}{k}\mathbf{1} \iff \exists c\in\mathbb{R} \textrm{ s.t. }\beta^\top\Sigma^{-1/2}\mu = c\mathbf{1}.$$
> That is, the average logit must be a *multiple of the all-ones vector* (we make this exact remark in the paragraph before (1)). Note that *we do not require $c=\frac{1}{2}$*. **In your notation, the constraint requires that the 2-vector $\beta^\top\Sigma^{-1/2}\mu$ be equal to $(c, c)$ for some scalar $c$.**
>
> In our theoretical analysis, we consider the setting where $c = 0$ **without loss of generality** because of the existence of a bias term $\beta_0$. Observe that for any feasible solution $\hat\beta$ such that $\hat\beta^\top\Sigma^{-1/2}\mu = c\mathbf{1}$, we can redefine the bias term as $\hat\beta_0^{new} = \hat\beta_0 - c\mathbf{1}$. The resulting expression parameterizes *exactly the same predictor* because this modification does not affect the output of the softmax, and now it also satisfies $\hat\beta^{new}$$^\top\Sigma^{-1/2}\mu = \mathbf{0}$. This is why we can assume $c=0$ WLOG.
>
> **Proof that the objective is convex:** We will formally prove the objective is convex. Consider any two solutions $\beta_1,\beta_2$ which satisfy the constraint. As we pointed out, this implies that there are scalars $c_1,c_2\in\mathbb{R}$ such that $\beta_1^\top\Sigma^{-1/2}\mu = c_1\mathbf{1}$ and $\beta_2^\top\Sigma^{-1/2}\mu = c_2\mathbf{1}$. So, any convex combination $\beta_\lambda = \lambda\beta_1 + (1-\lambda)\beta_2$ satisfies
> $$\beta_\lambda^\top\Sigma^{-1/2}\mu = \lambda\beta_1^\top\Sigma^{-1/2}\mu + (1-\lambda)\beta_2^\top\Sigma^{-1/2}\mu = \lambda c_1\mathbf{1} + (1-\lambda)c_2\mathbf{1} = \tilde c \mathbf{1},$$
> with $\tilde c = \lambda c_1 + (1-\lambda)c_2$. This in turn implies $$\textrm{softmax}(\beta_\lambda^\top\Sigma^{-1/2}\mu) = \textrm{softmax}(\tilde c \mathbf{1}) = \frac{1}{k}\mathbf{1}.$$
> So we've proven that a convex combination of any two elements of the feasible set remain in the feasible set, and therefore the feasible set is convex. As the DARE objective (1) is optimizing a convex objective subject to a convex constraint, it follows *the objective is convex*.
>
> Note that this convexity is independent of the application of an $\ell_2$ penalty. The penalty is simply to ensure a unique solution and improve the optimization.
>
> * **"What is the precise claim about how predictions are / are not affected by the DARE constraint."**
>
> The precise claim is that *the mean of the logits has no effect on the predictive distribution of the solution*. Formally: the predictions $\hat p(y) = \textrm{softmax}(\hat\beta^T \Sigma^{-1/2} x)$ are exactly the same with or without centering the adjusted $x$. This is in the definition of the constraint, which requires that the mean of the logits be a multiple of the all-ones vector, which softmax is invariant to. Note that this makes *no constraint* on the output distribution over labels. For any marginal distribution $p(y)$, there is a distribution over $x$ and a solution $\beta$ which matches $p(y)$ in its predictions while satisfying the DARE constraint.

---

> > ### Comment · Reviewer_4nvT · 2022-11-18
> > **Continuing discussion**
> >
> > I see that I was partially confused by the softmax / logits distinction here, although it makes my question about the distribution of predictions even more direct, I think?
> >
> > **Convexity**
> >
> > If you allow a bias term in the $d\times 2$ $\beta$ parameterization, can't arbitrarily set that bias term to take the same value for each column of $\beta$ and have the softmax be unchanged? It seems to me that the specific invariance in the softmax objective that's being voted here also implies that the loss is not convex. What's wrong with my reasoning here? And does the addition of a bias term complicate things?
> >
> > **Distribution on Prediction Distribution*"
> >
> > I don't understand the claim that you can constrain the mean of the predictions and not have an effect on the output distribution because softmax is invariant. For a fixed distribution over X, the distribution of predictions must change (just tautologically), right? I.e., in logistic regression, if you have a bias term then you would expect the mean prediction to match the mean label in the data, but under the DARE constraint the mean prediction will be 0.5.

---

> > > ### Author Response · Authors · 2022-11-18
> > > **Trying to understand the source of the error**
> > >
> > > * "can't arbitrarily set that bias term to take the same value for each column of  $\beta$ and have the softmax be unchanged?"
> > >
> > > Only if it is the case that the mean logit is *already* the same without the bias term. There is a difference between "setting the bias to be the same for each class" and "adding the same constant to the bias for each class". **Our constraint is not that $\hat\beta_0 = (c,c)$ for some $c$. It is that $\hat\beta^\top x + \hat\beta_0 = (c,c)$.** So for example, it could be the case that $\hat\beta^\top x = (a, b)$ for different values $a,b$, and then $\hat\beta_0 = (c-a, c-b)$. Note that $\hat\beta_0$ is different for each class, yet we have satisfied the constraint. We could then *additionally* add a constant, i.e. $\hat\beta_0^{new} = (c-a + d, c-b + d)$, giving $\hat\beta^\top x + \hat\beta_0 = (c+d,c+d),$ which still satisfies the constraint.
> > >
> > > **It is difficult to precisely identify where your reasoning is incorrect without a more precisely written statement.** Can you write out explicitly and formally why you think the objective is non-convex? In particular, do you agree that the unconstrained objective is convex? And do you follow our proof above that the constraint set is convex? If not, could you tell us what part don't you agree with?
> > >
> > > * "For a fixed distribution over X, the distribution of predictions must change (just tautologically), right?"
> > >
> > > The distribution of **logits** must change. Because softmax has an extra degree of freedom, it is not necessarily the case that the distribution of **predictions** (i.e., the probabilities after the softmax) changes. As an explicit example, if all values $X$ are shifted by some vector $v$ with $\beta^\top v = c\mathbf{1}$, then the **logits** will all increase by $c$, but the **predictions** will not change at all.
> > >
> > > * "in logistic regression, if you have a bias term then you would expect the mean prediction to match the mean label in the data, but under the DARE constraint the mean prediction will be 0.5."
> > >
> > > Both of these statements are not correct (*Edit: We have realized what you've said about the mean prediction would be consistent with the average of the *probabilities*.* Please see our response to this below). The DARE constraint does not require the mean prediction to be 0.5. As we pointed out in our earlier comment:
> > >
> > > *The softmax of the average of the logits is not the same as the average of the softmax of the logits*.
> > >
> > > The former is what we constrain, and the latter is the "mean prediction" to which you are referring. Softmax is nonlinear, so you cannot push the expectation through.
> > >
> > > Next, it is not the case that "if you have a bias term then you would expect the mean prediction to match the mean label". **This is the case in some very specific cases (such as Linear Discriminant Analysis), but not necessarily otherwise.** Your intuition is valid for linear regression, where the unbiasedness of the solution implies that $\mathbb{E}[\beta^\top x] = \mathbb{E}[y]$. This is not true in logistic regression, because of the nonlinearity of softmax.
> > >
> > > You can test this yourself! Here's a very simple program to prove this, where we fit a logistic regression model under perfect model specification:
> > >
> > > `import numpy as np`
> > >
> > > `from sklearn.linear_model import LogisticRegression`
> > >
> > > `from scipy.special import expit as sigmoid`
> > >
> > > `x = np.random.randn(500000, 1) ** 2  # p(x) follows a chi-square distribution`
> > >
> > > `beta, beta_0 = np.random.randn(1) * 2, np.random.randn(1) * 2`
> > >
> > > `y = np.random.binomial(1, sigmoid(beta * x + beta_0))[:, 0]`
> > >
> > > `clf = LogisticRegression().fit(x, y)`
> > >
> > > `print(f'parameters:\tbeta: {beta[0]:.5f}\tbias: {beta_0[0]:.5f}')`
> > >
> > > `print(f'estimates:\tbeta: {clf.coef_[0,0]:.5f}\tbias: {clf.intercept_[0]:.5f}')`
> > >
> > > `print(f'average prediction: {clf.predict(x).mean()}')`
> > >
> > > `print(f'average label: {y.mean()}')`
> > >
> > > **Edit: We have realized what you've said about the mean prediction would be consistent with the average of the *probabilities*.** This would be the case (under perfect model specification), but note that this would still not say anything about the distribution of *argmax class predictions*. So, under the correct model, you are right that this would imply that the average of the *probabilities* is (0.5, 0.5). But this could occur when half the points are predicted (1, 0) and the other half are predicted (0, 1), or it could similarly occur if 1/3 of the points are predicted (.9, .1) and 2/3 are predicted (.3, .7). So you can see that both produce an average probability of (0.5, 0.5), but with very different marginal distributions $p(y)$.

---

> > > > ### Comment · Reviewer_4nvT · 2022-11-18
> > > > **Clarification**
> > > >
> > > > Re: convexity, my claim is that the unconstrained objective as written is not convex, but it is convex in the case where you write it in terms of a vector $\beta$ instead of a matrix $\beta$.
> > > >
> > > > Re: label distribution, regardless of whether the mean prediction matches the mean label bc of nonlinearity, the larger question is how one can claim that the constraint doesn't affect the prediction distribution if the unconstrained predictions don't automatically satisfy the mean constraint. It seems that the constraint then changes the label distribution by definition if the original mean of the label distribution is not 0.5.

---

> > > > > ### Author Response · Authors · 2022-11-18
> > > > > **Still unclear**
> > > > >
> > > > > Can you give us a *precise, mathematical argument* for why the objective is no longer convex when $\beta$ is a vector? Keeping in mind when $\beta$ is a vector, the constraint becomes $\sigma(\beta^T\Sigma^{-1/2}\mu) = \frac{1}{2}$? Does our above proof not still apply, and if so, why not?
> > > > >
> > > > > You are correct that "implementing a constraint will have an effect on the label distribution." **We are not claiming that "the DARE constraint has no downstream effect on the predicted label distribution of the solution". We are claiming that "the DARE constraint does not *constrain* the predicted label distribution of the solution".**
> > > > >
> > > > > You were originally concerned because you thought that the constraint enforces the predictive distribution to be uniform over the labels, which may not actually be true of the test distribution. We are pointing out that the DARE constraint does not require the solution to fit a particular predictive marginal $\hat p(y)$ which does not match the ground truth, and so this is not a concern. For example, **$\ell_2$ regularization will *also* affect the predicted label distribution, but this is not a problem because it doesn't *constrain* the distribution.**

---

> > > > > > ### Comment · Reviewer_4nvT · 2022-11-18
> > > > > > **See above**
> > > > > >
> > > > > > To be clear, I'm saying it's not convex when beta is a **matrix**, which is how logistic regression is represented in (1), iiuc.
> > > > > >
> > > > > > Re: predictive distribution, I think we're getting somewhere. I will withdraw the label distribution question, but still have something to ask about the role of the constraint in logistic regression when beta is a vector, which I'll put in the thread under my other comment.

---

> > > > > > > ### Author Response · Authors · 2022-11-18
> > > > > > > **Response**
> > > > > > >
> > > > > > > Ok, thanks for the clarification. We'll consider the setting when $\beta$ is a matrix. To be clear, this implies the following:
> > > > > > >
> > > > > > > 1. $k > 2$, since we let $\beta$ be a vector for binary classification.
> > > > > > >
> > > > > > > 2. If $d$ is the number of feature dimensions (i.e., $x\in\mathbb{R}^d$), then we have $\beta\in\mathbb{R}^{d\times k}$ and $\beta_0 \in \mathbb{R}^k$. This is the standard parameterization for multinomial logistic regression.
> > > > > > >
> > > > > > > 3. When predicting on a point $x$, we get the logits as $\beta^T\Sigma_e^{-1/2}x + \beta_0$ (written as $\beta^T\Sigma_e^{-1/2}x$ in the paper because we omit the bias), which will be a vector in $\mathbb{R}^k$. We then take the softmax of these logits to get a probability distribution on the $k$-dimensional simplex, which defines our predicted probabilities for each of the $k$ classes.
> > > > > > >
> > > > > > > Now let's consider the objective (1). The objective is made up of two parts: (i) the loss, and (ii) the constraint.
> > > > > > >
> > > > > > > The loss is standard multinomial logistic regression. The covariates have been modified in a particular way, but not in any manner that uses the parameters, so this does not affect the convexity. That is, you can treat $\Sigma_e^{-1/2}x$ as some "new covariates" $\hat x$ and observe that we are doing multinomial logistic regression on these new covariates. Ignoring the constraint, would you agree that the multinomial logistic loss function is convex?
> > > > > > >
> > > > > > > Next, consider the constraint. As we proved above, the set of all vectors $\beta$ which satisfy this constraint is convex. Therefore, the feasible set is a convex set. Do you see anything wrong with our proof?
> > > > > > >
> > > > > > > If both of these points hold, then it must be the case that the objective is convex. If not, can you tell us which part of the above you do not agree with?

---

> > > > > > > > ### Comment · Reviewer_4nvT · 2022-11-18
> > > > > > > > **Convexity**
> > > > > > > >
> > > > > > > > I would not agree that the multinomial logistic regression objective is convex. To make it convex you need to effectively eliminate one column of beta. That's what I've been saying. That's why (I think) for the k=2 case you have kept reducing beta to a vector.

---

> > > > > > > > > ### Comment · Reviewer_4nvT · 2022-11-18
> > > > > > > > > **Reference**
> > > > > > > > >
> > > > > > > > > For example, from the Wiki page https://en.m.wikipedia.org/wiki/Multinomial_logistic_regression
> > > > > > > > >
> > > > > > > > > Note that not all of the {\displaystyle \beta _{k}}\beta _{k} vectors of coefficients are uniquely identifiable. This is due to the fact that all probabilities must sum to 1, making one of them completely determined once all the rest are known. As a result, there are only {\displaystyle k-1}k-1 separately specifiable probabilities, and hence {\displaystyle k-1}k-1 separately identifiable vectors of coefficients.

---

> > > > > > > > > > ### Author Response · Authors · 2022-11-18
> > > > > > > > > > **To clarify what you are claiming**
> > > > > > > > > >
> > > > > > > > > > We do not understand how the section from Wikipedia you've quoted is relevant to the convexity of the loss. To be clear, the way we understand your previous post is that you are claiming:
> > > > > > > > > >
> > > > > > > > > > "*When solving multinomial logistic regression with the parameterization $\beta\in\mathbb{R}^{d\times k}, \beta_0\in\mathbb{R}^k$ for $x\in\mathbb{R}^d$ and $k$ classes, the cross-entropy loss is *not* convex in the parameters $\beta,\beta_0$.*"
> > > > > > > > > >
> > > > > > > > > > **Is this what you are claiming?**

---

> > > > > > > > > > > ### Comment · Reviewer_4nvT · 2022-11-18
> > > > > > > > > > > **Correct**
> > > > > > > > > > >
> > > > > > > > > > > Yes, that is my claim. The wiki excerpt says that there are many maximizers of the objective. That's the "not identifiable" statement. The article goes on to describe conventional methods for resolving this non-convexity, including fixing one of the columns of beta to default values.
> > > > > > > > > > >
> > > > > > > > > > > IMO, this could be patched by specifying the general DARE objective by including one of these standard comvexifying constraints. However, I'm not sure that the DARE constraint would work as written. Instead, simply requiring that $\beta^\top \mu =0$ (taking my to be the transformed mean here) seems like it would work? That is what is assumed for all of the reformulations in the theory.

---

> > > > > > > > > > > > ### Author Response · Authors · 2022-11-18
> > > > > > > > > > > > **.**
> > > > > > > > > > > >
> > > > > > > > > > > > We wanted to thank you once more for taking the time for this discussion!
> > > > > > > > > > > >
> > > > > > > > > > > > In your last message, you’ve used “convexity” and “identifiability” interchangeably, but these two concepts are distinctly different. Namely, while it is true (as the Wikipedia page you linked says) that the objective (1) does not *uniquely* identify $\beta$ (precisely, there are multiple vectors $\beta$ that result in the same predictive distribution $\hat p(y \mid x)$), this does *not* mean the loss is not convex. In fact, the multinomial logistic regression loss parametrized as (1) *is* convex (see, e.g. https://trungvietvu.github.io/notes/2016/MLR for a full proof).
> > > > > > > > > > > >
> > > > > > > > > > > > Your suggestion of requiring $\beta^T \Sigma^{-1/2} \mu = 0$ is a special case of requiring $\beta^T \Sigma^{-1/2} \mu = c \mathbf{1}$—which is what we do via a regularizer. (See “Implementation in practice”, page 5 of our manuscript.)  Your suggestion could be seen as “tie breaking” the non-identifiability in the softmax by “preferring 0”. In our experiments, we “tie break” by adding a little $\ell_2$ regularization to $\beta$. In the theory this doesn’t matter, as all $\beta$ vectors that result in the same predictive distributions $\hat p(y|x)$ are equally good.

---

> > > > > ### Comment · Reviewer_4nvT · 2022-11-18
> > > > > **Addendum**
> > > > >
> > > > > To be clear, I'm asking about a tension bw the arguments that have been made here. There is an argument that the distribution of predictions does not change when the constraint is added, which would then imply that the loss does not change. This the unconstrained objective is not convex.
> > > > >
> > > > > Or, the unconstrained objective is convex, in which case the constraint must change the distribution of predictions.
> > > > >
> > > > > I think the tension here comes from the fact that beta is a matrix in (1) and a vector in the theory. If beta is a matrix, you get the invariance of the softmax . If beta is a vector, you're computing the sigmoid $(1 + \exp(-beta^\top X))^{-1}$, not softmax (although you can get this if you compute softmax setting the first column of beta to 0), so you don't.

---

> > > > > > ### Author Response · Authors · 2022-11-18
> > > > > > **Asking for some further clarification**
> > > > > >
> > > > > > "There is an argument that the distribution of predictions does not change when the constraint is added, which would then imply that the loss does not change. This the unconstrained objective is not convex."
> > > > > >
> > > > > > Please see our response below. We are not claiming that the distribution does not change when the constraint is added. As we noted, standard regularization *also* changes the distribution, but it does not *constrain* it. Can you formally explain why this causes the unconstrained objective to be non-convex?
> > > > > >
> > > > > > "If beta is a vector, you're computing the sigmoid, not softmax... so you don't."
> > > > > >
> > > > > > **Above, we wrote the following:**
> > > > > >
> > > > > > > The DARE objective is defined the same for binary or multinomial logistic regression, with $\beta\in\mathbb{R}^{d\times k}, \beta_0\in\mathbb{R}^k$. For binary classification, we allow $\beta$ to be a vector—the constraint would then be $\beta^T x = 0$ (i.e., $\sigma(\beta^T x) = \frac{1}{2}$), because this would induce a uniform distribution over the two classes. Note that this is exactly the constraint for linear regression, and this point should hopefully clarify the connection between the two.
> > > > > >
> > > > > > Does this help to clarify?

---

> > > > > > > ### Comment · Reviewer_4nvT · 2022-11-18
> > > > > > > **Very simple ask**
> > > > > > >
> > > > > > > Can you write down how you get the logistic regression DARE objective from the objective written in (1) when $d=2$?
> > > > > > >
> > > > > > > Also, is the primary claim here that maximizing the objective on centered or uncentered data subject to the DARE constraint is equivalent? If so it would be useful just to state that claim clearly in the text.
> > > > > > >
> > > > > > > And thanks for this discussion. If you can fulfill my ask here, I'll raise my score.

---

> > > > > > > > ### Author Response · Authors · 2022-11-18
> > > > > > > > **Hope this helps**
> > > > > > > >
> > > > > > > > Happy to! When you say $d=2$, do you mean $k=2$? We assume you are referring to binary classification, not setting the number of features to 2. We note that both binomial and multinomial classification are solved by "logistic regression", but we assume when you wrote "the logistic regression DARE objective" that you meant "the *binomial* logistic regression DARE objective". If this is not correct, let us know and we'll revise this comment.
> > > > > > > >
> > > > > > > > Under the assumption you meant $k=2$:
> > > > > > > >
> > > > > > > > The objective (1) for multiclass classification with multinomial logistic regression (copied exactly from the paper) is:
> > > > > > > >
> > > > > > > > $$\min_\beta \sum_{e\in\mathcal{E}} \mathbb{E}_{p^e}[\ell(\beta^T\Sigma_e^{-1/2}x, y)]\quad\textrm{subject to }\quad \textrm{softmax}(\beta^T\Sigma_e^{-1/2}\mu_e) = \frac{1}{k}\mathbf{1}.$$
> > > > > > > >
> > > > > > > > As we write in the sentence after Equation (1), for binary classification we let $\beta$ be a vector and replace the softmax with the logistic function. Note that because $\beta$ is a vector, the logit output will be 1-dimensional, and therefore $\mathbf{1}$ is a one-dimensional "vector" with just the value 1. Thus, $\frac{1}{k} \mathbf{1} = \frac{1}{k}$. Plugging these changes into Equation (1) gives the binomial logistic regression DARE objective:
> > > > > > > >
> > > > > > > > $$\min_\beta \sum_{e\in\mathcal{E}} \mathbb{E}_{p^e}[\ell(\beta^T\Sigma_e^{-1/2}x, y)]\quad\textrm{subject to }\quad \sigma(\beta^T\Sigma_e^{-1/2}\mu_e) = \frac{1}{2}.$$
> > > > > > > >
> > > > > > > > Because $\sigma$ is invertible, this is equivalent to requiring $\beta^T\Sigma_e^{-1/2}\mu_e = 0$, as we note in the paper.
> > > > > > > >
> > > > > > > > * **"Also, is the primary claim here that maximizing the objective on centered or uncentered data subject to the DARE constraint is equivalent? If so it would be useful just to state that claim clearly in the text."**
> > > > > > > >
> > > > > > > > This is not the *primary* claim. It is certainly a *consequence* of the constraint, but our primary motivation is to ensure that the mean representation does not affect the prediction, because we want our classifier to be invariant to certain structured shifts in the distribution mean at test time. This is achieved if $\beta^T\Sigma_e^{-1/2}\mu_e = c\mathbf{1}$ in the multiclass case, or $\beta^T\Sigma_e^{-1/2}\mu_e = 0$ in the binary case.
> > > > > > > >
> > > > > > > > Hopefully the above two points are clear and answer your question. If not, let us know. Thanks again for your continued efforts.

---

> > > > > > > > > ### Comment · Reviewer_4nvT · 2022-11-18
> > > > > > > > > **Thanks; Follow-up (sorry)**
> > > > > > > > >
> > > > > > > > > You're right I meant k=2.
> > > > > > > > >
> > > > > > > > > So what I don't understand here is why you get to let beta be a vector when k=2 as a special case, rather than sticking to the specification of the DARE objective in (1). What I wanted was for you to show that working thru the DARE objective (1) when k=2 is symbolically equivalent to eliminating one column of the beta matrix and letting beta be a vector (maybe with a bias term?). This matters bc there's a similar overpareterization issue when k>2, which induces the specific invariance in softmax that you note, and which I continue to insist makes the unconstrained objective nonconvex.
> > > > > > > > >
> > > > > > > > > It's not clear to me how the theory for the vector beta case (which specifies a convex objective) should generalize to the multinomial logistic regression case where beta is a d x k matrix, or how the claim about the invariance of softmax applies to the sigmoid, which doesn't have this invariance.
> > > > > > > > >
> > > > > > > > > This is why I'm asking whether the guiding principle here is to make the centered and uncentered formulations equivalent. Then at least all of the special cases seem to align. At the very least it seems like the various DARE objectives need to be specified separately.

---

> > > > > > > > > > ### Comment · Reviewer_4nvT · 2022-11-19
> > > > > > > > > > **Ping**
> > > > > > > > > >
> > > > > > > > > > Just want to make sure that you see this response here too. The non-identifiability issue (due to lack of *strict* convexity) we discussed in the other thread is salient here.

---

> > > > > > > > > > > ### Author Response · Authors · 2022-11-19
> > > > > > > > > > > **Thanks, hope this suffices**
> > > > > > > > > > >
> > > > > > > > > > > Thank you for the ping. We really believe the answer we wrote in the other thread about distinguishing “convexity” and “identifiability” should help clarify matters. Hopefully, given that subthread, we agree on the following two points:
> > > > > > > > > > >
> > > > > > > > > > > 1. The objective (1) *is* convex with either parametrization of the predictor. (The vector parametrization used in the theory, or the matrix parametrization in (1).). It just is not *strongly* convex with the matrix parametrization.
> > > > > > > > > > >
> > > > > > > > > > > 2. Due to lack of strong convexity, in the matrix parametrization, the optimal $\beta^*$ will not be unique. However, this is not a problem, because the optimal *predictive distribution* $\hat{p}(y|x)$ will be unique, and agree in both parametrizations.
> > > > > > > > > > >
> > > > > > > > > > > In this response, we understand that you are asking how these points “interact” with the theory in Sections 4 and 5.
> > > > > > > > > > > In these sections, since we are looking at the case $k=2$, it is more convenient for us to work with the vector parametrization, because it makes the optimum $\beta^*$ unique. If we trained instead (*in practice, for the experiments*) with the matrix parametrization, whatever predictor we converge to would give rise to the same predictive distribution as the vector parametrization. Thus, this is purely for convenience. *For the experiments* in which $k > 2$, we just train with objective (1). Since we are not proving anything about this case in Sections 4+5, there’s no reason to discuss/comment on the pros and cons of either parametrization.
> > > > > > > > > > >
> > > > > > > > > > > About making centered and uncentered formulations equivalent: yes, making these formulations equivalent is one way to think of (1). This is an informal way to express the constraint that the softmax of the mean logits should be a multiple of the all-ones vector. As you’ve noted, in the “$\beta$ is a vector” case, this principle would require $\beta^\top\mu = 0$, which induces the same invariance for sigmoid as the “multiple of all ones” requirement induces for softmax.

---

> > > > > > > > > > > > ### Comment · Reviewer_4nvT · 2022-11-19
> > > > > > > > > > > > **Fair enough**
> > > > > > > > > > > >
> > > > > > > > > > > > Thanks for talking through my questions. I do think that this could be greatly streamlined in the paper, as I think a lot of my questions revolve around whether the key idea is the centered/non-centered equivalence or the non-identifiability of the softmax. For me, a lot of this confusion came from not knowing how to interpret the phrase "the mean has no effect on predictions".
> > > > > > > > > > > >
> > > > > > > > > > > > I can change my rating to weak accept for now, but am still in the fence about clarity. I'll see how the other reviewers felt in discussions.

---

> > > > > > > > > > > > > ### Author Response · Authors · 2022-11-19
> > > > > > > > > > > > > **Thanks for the update!**
> > > > > > > > > > > > >
> > > > > > > > > > > > > We're very happy to hear that this has clarified things! We will continue to edit the manuscript and work to make the points raised in this discussion clearer for future readers.
> > > > > > > > > > > > >
> > > > > > > > > > > > > **We really do appreciate your efforts here.** We know that it would have been easier to just stop responding, and we are grateful that you took this time to engage so that we could figure out the source of confusion.

---

> > > > > > > > > > > > ### Comment · Reviewer_4nvT · 2022-11-19
> > > > > > > > > > > > **Something to check**
> > > > > > > > > > > >
> > > > > > > > > > > > This isn't necessary for this review process, but it would be worth checking whether the poor performance of the unconstrained objective in Figure 4 (when lambda = 0) would be improved by using one of the standard tie-breaking strategies described in the wiki article. As I've mentioned, this has been a practical issue in my experience.

---

### Decision · Program_Chairs · 2023-01-20

**Decision:**

Reject

**Justification For Why Not Higher Score:**

The paper needs another round of peer review.

**Justification For Why Not Lower Score:**

N/A

**Metareview: Summary, Strengths And Weaknesses:**

This paper studies the OOD generalization problem and made two important claims:

1) Based on empirical results on a few standard benchmark datasets, the paper claims that the empirical risk minimization (ERM) already learns features "sufficient" for out-of-distribution (OOD) generalization. The paper comes to this conclusion by comparing the performance of two models on DomainBed (Gulrajani & Lopez-Paz, 2021). The first model was trained on the test domains by adjusting the linear predictors on top of the frozen features obtained from finetuning with ERM on the training domains. The second model was trained end-to-end on the training domains and test domains. The authors observed that the performance gap between these two models is small, implying that the finetuned features with ERM already made up a significant performance gap in OOD generalization.

2) Inspired by the aforementioned observation, the paper proposes a new objective called domain-adjusted regression (DARE). DARE learns a linear robust predictor on top of the frozen feature extractor trained with ERM on all training domains. The predictor is learned by performing domain-specific normalization of the feature-space mean and covariance, i.e., whitening. Since the whitening matrices associated with the test domains are unknown, the paper considers averaging the whitening matrices over the training domains and using the average as an approximation for the test domains. The proposed method performs well on some DomainBed datasets.

While the hypothesis that ERM already learns features "sufficient" for OOD generalization is compelling, the scientific method used in testing this hypothesis is far from rigorous and is lacking in many aspects. Firstly, unlike IID generalization, OOD generalization is ill-defined in the sense that there are infinitely many ways in which the model can generalize to the unseen test domains. Hence, it remains unclear what "sufficient" means. It appears the paper adopts the "average-case" performance as an indicator of success. Furthermore, the experiments were conducted on a handful of benchmark datasets, i.e., mostly DomainBed, in which distribution shifts highly depend on the design of the benchmark datasets. Without experiments on datasets with real-world distributions such as WILDS, the conclusion remains at best at the level of benchmark datasets rather than in general. Based on the current setup,  the results may only highlight the limitation of benchmark datasets commonly used in the community and the ways used to construct them as ERM can already produce a competitive "baseline", which has also been observed in previous work. The current conclusion is therefore misleading.

If this hypothesis turns out to be true, a well-written paper with rigorous scientific methods and convincing results supporting this hypothesis alone would already be deemed publishable at the top-tier conference. Since the first part of this paper serves as a justification for the proposed method, it is also important that one puts more emphasis on it.

For the second part of the paper, the reviewers also raised several concerns. Firstly, it is unclear when the proposed method is supposed to work and when not, i.e., what types of shifts is the method expected to be robust against? Secondly, important comparisons with existing work are missing, e.g., existing whitening techniques, test-time UDA methods, and CORAL. Because of this, it remains unclear whether it is really helpful to freeze the features as no comparison is made.

To conclude, the paper investigates a compelling hypothesis, but due to unrigorous experimental methodology, the results remain inconclusive and the general conclusion made in this paper can be misleading. The paper will benefit from another round of peer review. Hence, I cannot recommend this paper for publication at ICLR2023.

**Summary Of Ac-Reviewer Meeting:**

This is a borderline paper that leans toward acceptance. After the AC-reviewer meeting, the paper remains at the borderline, but it becomes more apparent that there are several concerns with the paper that need to be addressed such as overclaiming the experimental results (the statement is too generic), lack of several baselines in the second part, conditions under which the method is expected to work or not to work, to name a few. No reviewer is willing to champion this paper.

It is therefore unclear whether the remaining issues can be addressed within the timeline of the camera-ready version as it requires a reframing of the general statement, providing additional discussion with respect to previous work, and conducting additional experiments. Based on the AC's judgment, it's therefore in the best interest of the paper, authors, and research community that this paper goes through another round of peer review after a major revision.